# ON THE IDENTIFIABILITY OF NONLINEAR REPRESENTATION LEARNING WITH GENERAL NOISE

## ABSTRACT

Noise is pervasive in real-world data, posing significant challenges to reliably uncovering latent generative processes. While evolution may have enabled the brain to solve such problems over millions of years, machine learning faces this task in just a few years. Most prior identifiability theories, even under restrictive assumptions like linear generating functions, are limited to handling only additive noise and fail to address nonparametric noise. In contrast, we study the problem of provably learning nonlinear representations in the presence of nonparametric noise. Specifically, we show that, under certain structural conditions between latent and observed variables, latent factors can be identified up to element-wise transformations, even when both the generative processes and noise are nonlinear and lack specific parametric forms. We further present extensions of the general framework, demonstrating trade-offs between different assumptions and the identifiability of latent variables in the presence of both noise and distortions. Moreover, we prove that the underlying directed acyclic graph can be recovered even with nonlinear measurement errors, offering independent insights into structure learning. Our theoretical results are validated on both synthetic and real-world datasets.

## 1 INTRODUCTION

Uncovering the underlying generative processes from observational data is a cornerstone of scientific discovery. While modern machine learning excels at capturing complex patterns in real-world data, it often lacks identifiability guarantees that the learned representations correspond to the true latent factors generating the data (Locatello et al., 2019). For many applications, the ability to reliably identify these latent factors is critical for unbiased analysis of complex data, such as in economics (Hu, 2008), psychology (Bollen, 2002), and biomedical research (Imbens & Rubin, 2015).

Classical methods for recovering the underlying data-generating process, with theoretical guarantees, have traditionally focused on linear relationships between latent and observed variables (Comon, 1994). Recent advances in nonlinear Independent Component Analysis (ICA) have extended this theory to nonlinear contexts (Hyvärinen & Pajunen, 1999; Hyvärinen et al., 2024), incorporating additional assumptions such as auxiliary variables (Hyvärinen & Morioka, 2016; Hyvärinen et al., 2019), time-series data (Hyvärinen & Morioka, 2017; Hälvä et al., 2021; Yao et al., 2021), structural conditions (Moran et al., 2021; Zheng et al., 2022), or specific functional forms (Taleb & Jutten, 1999; Buchholz et al., 2022). However, many of these approaches operate in deterministic settings, without accounting for noise, which limits their applicability in real-world scenarios where data is often affected by various forms of randomness.

While some studies have integrated noise into latent variable models, existing frameworks remain restrictive. Classical factor model literature, for instance, has typically employed additive noise under specific parametric assumptions related to the data-generating function, such as normality, linearity, or its reducibility to linear models (Reiersøl, 1950; Lawley & Maxwell, 1962; Kenny & Judd, 1984; Bekker & ten Berge, 1997; Ikeda & Toyama, 2000; Beckmann & Smith, 2004; Bonhomme & Robin, 2009). Recent works have expanded these frameworks to more general nonlinear settings with non-deterministic transformations, especially with advancements in nonlinear ICA (Khemakhem et al., 2020a; Sorrenson et al., 2020; Lachapelle et al., 2022; Hälvä et al., 2024). However, these methods continue to exhibit limitations, as they are primarily restricted to handling additive noise, even when further constraints, such as temporal structures or weak supervision, are incorporated.

Despite considerable advancements in establishing theoretical guarantees for latent variable models, the challenge of provably learning nonlinear representations in the presence of complex noise persists. This issue is particularly relevant in real-world applications, where data is frequently contaminated by various forms of nonparametric noise. In medical imaging, for example, accurately capturing detailed anatomical structures requires models that can disentangle meaningful signals from pervasive noise (Suetens, 2017). Similarly, in autonomous driving, sensor data from lidar and cameras must be interpreted with precision, despite environmental distortions and unpredictable noise, to ensure reliable perception and decision-making (Yurtsever et al., 2020). Large-scale foundation models, trained on extensive text corpora, also face the challenge of handling ambiguous or noisy data from diverse sources that often lack clear functional forms (Bommasani et al., 2021). Relying exclusively on additive noise models can introduce bias into representations, limiting the model's ability to veridically discover the process underlying the data. This brings us to a crucial, yet unresolved, question:

*Can machines reliably reveal the hidden world amid the chaos of noise?*

Towards addressing this open question, we establish a set of theoretical results for provably learning nonlinear representations in the presence of general noise. We demonstrate that, even when the underlying generative process is nonlinear and the noise lacks a specific parametric form, it is still possible to recover the underlying process with theoretical guarantees. Specifically, we prove that, under conditions on the hidden connective structure between latent and observed variables, the latent variables can be identified up to element-wise indeterminacies (Thm. 1). These guarantees hold in general settings without imposing restrictions on the distribution of the latent variables, the specific form of the generating function, or the parametric structure of the noise. To the best of our knowledge, this is one of the first results to achieve identifiability for nonlinear representation learning in an unsupervised setting with general noise.

Moreover, to illustrate the implications of our theoretical framework, we demonstrate that several challenging problems can be addressed under the umbrella of nonparametric identifiability with noise. First, we show that even weaker assumptions can be sufficient when leveraging the parametric form of the noise (Thm. 2), exploring the trade-off between different conditions. Next, we prove that the latent variables can be identified despite nonlinear distortions (i.e., element-wise unknown nonlinear transformations) combined with general noise (Cor. 1). Furthermore, we show that the hidden (causal) directed acyclic graph (DAG) among variables can also be uncovered in the general setting, even in the presence of nonlinear measurement error (Thm. 3, Prop. 2). Consequently, our identifiability theory offers broad applicability to a variety of existing problems, and the theorems may hold independent significance in fields such as generative modeling and causal discovery. We validate the theoretical results through experiments on synthetic and real-world datasets, but addressing practical challenges like finite sample errors remains a key open problem for future work to enable broader deployment of identifiability theory.

## 2 PRELIMINARIES

**Data-generating Process.** We consider a data-generating process where the observed variables $\mathbf{x} = (\mathbf{x}_1, \ldots, \mathbf{x}_m) \in \mathcal{X} \subseteq \mathbb{R}^m$ are generated from latent variables $\mathbf{z} = (\mathbf{z}_1, \ldots, \mathbf{z}_n) \in \mathcal{Z} \subseteq \mathbb{R}^n$ and independent noise variables with a nonparametric form $\boldsymbol{\epsilon} = (\boldsymbol{\epsilon}_1, \ldots, \boldsymbol{\epsilon}_{n_e}) \in \mathcal{E} \subseteq \mathbb{R}^{n_e}$ through a general function $f$, which is a $\mathcal{C}^2$-diffeomorphism onto its image $\mathcal{X} \subseteq \mathbb{R}^m$. Specifically, the process (Fig. 1) is defined as:

$$\mathbf{x} = f(\mathbf{z}, \boldsymbol{\epsilon}), \tag{1}$$

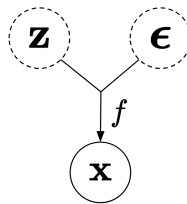

Figure 1: Visualization of Eq. (1).

where $f : \mathbb{R}^n \times \mathbb{R}^{n_e} \to \mathbb{R}^m$. Following the standard setting (Hyvärinen et al., 2024), all latent variables and noise variables possess positive and twice continuously differentiable probability density functions.

**Main Objective.** Given only the observational data $\mathbf{x}$, we aim to recover the underlying generating process related to latent variables $\mathbf{z}$, of which the main objective is defined as follows.

**Definition 1 (Element-wise Identifiability).** *The latent variables $\mathbf{z}$ are element-wise identifiable if there exists an invertible function $h$ and a permutation $\pi$ s.t. $\hat{\mathbf{z}}_i = h_i(\pi(\mathbf{z}_i))$.*

Element-wise identifiability guarantees that the estimated factors correspond to the true generating factors without any mixture or entanglement. Standard ambiguities such as permutations and rescaling may remain after identification, which are fundamental indeterminacies commonly noted in the literature (Hyvärinen & Pajunen, 1999; Khemakhem et al., 2020a; Sorrenson et al., 2020; Hälvä et al., 2021; Yao et al., 2021; Lachapelle et al., 2022; Buchholz et al., 2022; Zheng et al., 2022; Lachapelle et al., 2024; Hyvärinen et al., 2024) and represent the best achievable outcome without imposing further restrictive assumptions. Following the previous works (see e.g., a recent survey (Hyvärinen et al., 2024)), all of our results are in the asymptotic setting.

**Technical Notations.** To facilitate the discussion, we introduce some technical notations. For any vector $\mathbf{v} \in \mathbb{R}^d$ and a subset $\mathcal{S} \subseteq \{1, \ldots, d\}$, we define the subspace $\mathbb{R}^d_{\mathcal{S}} = \{\mathbf{v} \in \mathbb{R}^d \mid \mathbf{v}_i = 0 \text{ if } i \notin \mathcal{S}\}$; that is, vectors in this subspace have zeros in all components outside $\mathcal{S}$. The cardinality of a set $\mathcal{S}$ is denoted by $|\mathcal{S}|$. For a matrix $\mathbf{M} \in \mathbb{R}^{m \times n}$, we denote its $i$-th row by $\mathbf{M}_{i,:}$ and its $j$-th column by $\mathbf{M}_{:,j}$. The support of $\mathbf{M}$ is defined as $\text{supp}(\mathbf{M}) = \{(i,j) \mid \mathbf{M}_{i,j} \neq 0\}$. For matrix-valued functions $\mathbf{M}(\theta) : \Theta \to \mathbb{R}^{m \times n}$, where $\Theta$ is the parameter space, we define the support over $\Theta$ as $\text{supp}(\mathbf{M}(\Theta)) = \{(i,j) \mid \exists \theta \in \Theta \text{ s.t. } \mathbf{M}(\theta)_{i,j} \neq 0\}$. For any set of indices $\mathcal{S} \subseteq \{1, \ldots, m\} \times \{1, \ldots, n\}$, we define $\mathcal{S}_{i,:} = \{j \mid (i,j) \in \mathcal{S}\}$ and $\mathcal{S}_{:,j} = \{i \mid (i,j) \in \mathcal{S}\}$ to represent the column indices associated with row $i$ and the row indices associated with column $j$, respectively.

The Jacobian matrix of $f$ w.r.t. $\mathbf{z}$ is denoted by $D_{\mathbf{z}} f \in \mathbb{R}^{m \times n}$, has elements $(D_{\mathbf{z}} f)_{i,j} = \partial f_i / \partial \mathbf{z}_j$, and its support is defined as $\mathcal{F}_z = \text{supp}(D_{\mathbf{z}} f)$. Similar notations are used across different contexts, where the specific function and variables may vary accordingly. Estimated quantities are indicated with a hat symbol, such as $\hat{f}$ for an estimate of $f$ and $\hat{\mathbf{z}}$ for estimated latent variables. The estimated model $(\hat{f}, \hat{\mathbf{z}}, \hat{\epsilon})$ follows the data-generating process and matches the observed distributions, i.e., $p(\hat{\mathbf{x}}) = p(\mathbf{x})$ ($p(\hat{\mathbf{x}}|\mathbf{u}) = p(\mathbf{x}|\mathbf{u})$ if there exists a domain variable $\mathbf{u}$).

In the relation $D_{\hat{\mathbf{z}}} \hat{f}(\cdot) = D_{\mathbf{z}} f(\cdot) \, \mathbf{T}(\cdot)$, $\mathbf{T}(\cdot)$ is a matrix-valued function whose domain may vary depending on the context. We denote by $\boldsymbol{T}$ the set of matrices that share the same support as $\mathbf{T}(\cdot)$, i.e., $\boldsymbol{T} = \{\mathrm{T} \in \mathbb{R}^{n \times n} \mid \text{supp}(\mathrm{T}) = \text{supp}(\mathbf{T}(\cdot))\}$. We use $(\cdot)^{(\ell)}$ to denote a point with index $\ell$ (e.g., $(\mathbf{z}, \epsilon)^{(\ell)}$). A complete summary of notations can be found in Appx. A.

## 3 IDENTIFIABILITY WITH GENERAL NOISE

In real-world scenarios, where the underlying processes are unknown, it is essential to avoid assumptions about specific parametric forms of the generating process, latent variables, or noise. Therefore, we propose the following theorem to establish nonparametric identifiability in the general case.

**Theorem 1.** *Let the observed data be generated by a model defined in Eq. (1). Together with a $\ell_0$ regularization on $\hat{\mathcal{F}}_{\hat{z}}$ during estimation ($\|\hat{\mathcal{F}}_{\hat{z}}\|_0 \leq \|\mathcal{F}_z\|_0$), suppose the following assumptions:*

    *i. (Nondegeneracy) For all $i \in \{1, \ldots, n\}$, there exist points $\{(\mathbf{z}, \epsilon)^{(\ell)}\}_{\ell=1}^{|(\mathcal{F}_z)_{i,:}|}$ and a matrix $\mathrm{T} \in \boldsymbol{T}$ s.t. $\text{span}\{D_{\mathbf{z}} f((\mathbf{z}, \epsilon)^{(\ell)})_{i,:}\}_{\ell=1}^{|(\mathcal{F}_z)_{i,:}|} = \mathbb{R}^n_{(\mathcal{F}_z)_{i,:}}$ and $\left[D_{\mathbf{z}} f((\mathbf{z}, \epsilon)^{(\ell)}) \mathrm{T}\right]_{i,:} \in \mathbb{R}^n_{(\hat{\mathcal{F}}_{\hat{z}})_{i,:}}$.*

    *ii. (Domain Variability) For any set $A \subseteq \mathcal{Z} \times \mathcal{E}$ with non-zero probability measure that cannot be expressed as $B_\epsilon \times B_{\mathbf{z}}$ for any $B_\epsilon \subseteq \mathcal{E}$ and $B_{\mathbf{z}} = \mathcal{Z}$, there exist two domains $u_1$ and $u_2$ that are independent of $\epsilon$ s.t.*

$$\int_{(\mathbf{z}, \epsilon) \in A} [p(\mathbf{z}, \epsilon | u_1) - p(\mathbf{z}, \epsilon | u_2)] \, d\mathbf{z} \, d\epsilon \neq 0.$$

    *iii. (Structural Sparsity) For all $k \in \{1, \ldots, n\}$, there exists a set $\mathcal{C}_k$ s.t. $\bigcap_{i \in \mathcal{C}_k} (\mathcal{F}_z)_{i,:} = \{k\}$.*

*Then latent variables $\mathbf{z}$ are element-wise identifiable (Defn. 1).*

**Remark.** Since we are working with a nonparametric form of noise, rather than additive noise, the noise can alter the latent distribution in a rather arbitrary manner. As a result, traditional distributional assumptions offer limited insight for this general setting. Therefore, in Thm. 1, we leverage a structural view, focusing on the connective relations between latent and observed variables, which naturally generalize beyond specific functional forms or distributions.

**Proof Sketch.** We leverage distributional variability across two domains of the latent variables $\mathbf{z}$ to disentangle $\mathbf{z}$ and $\boldsymbol{\epsilon}$ into independent subspaces. To separate general noise from latent variables, we use the independence between $\mathbf{z}$ and $\boldsymbol{\epsilon}$ alongside the variability within $\mathbf{z}$. The structural sparsity condition is then employed to identify individual components of $\mathbf{z}$ in the nonlinear setting. Specifically, for each latent variable, the intersection of parental sets from a subset of observed variables uniquely specifies it. Since we only achieve relations among supports due to the nonparametric nature of the problem, an unresolved element-wise transformation remains. Consequently, we achieve element-wise identifiability for the latent variables $\mathbf{z}$ (Defn. 1).

**Insights and Implications.** Theorem 1 shows that, under appropriate conditions, the latent variables of a nonlinear data-generating process with nonparametric noise can be identified up to an element-wise invertible transformation and a permutation. This ensures that, no matter how complex the noise or mixing process, the underlying generative factors can still be provably recovered and disentangled. Such a result is particularly important for real-world applications, where noise often plays a disruptive role in biasing observations.

Furthermore, the assumption that noise is merely additive or follows a specific parametric form, as is common in many traditional frameworks, can also lead to misrepresentations of real-world complexity. For instance, if we assume $\mathbf{x} = \mathbf{z} + \boldsymbol{\epsilon}$, where $\boldsymbol{\epsilon} \sim \mathcal{N}(0, 0.1)$, we might overlook scenarios where noise interacts with latent variables in more complex ways, such as multiplicative noise $\mathbf{x} = \mathbf{z} \cdot (1 + \boldsymbol{\epsilon})$. Our theory ensures that even with nonparametric noise and nonlinear generating process, the latent variables of interest can be provably recovered, without being confounded by noise-induced misrepresentation.

**On the Assumptions.** Recovering latent variables from observational data in Eq. 1 is well-known to be impossible without additional assumptions, even when deterministic transformations are involved (Hyvärinen & Pajunen, 1999). The challenge becomes even more pronounced in the presence of nonparametric noise. Revealing the hidden generating process from the vast space of possible functions is inherently ill-posed. Therefore, to make the problem tractable and ensure sufficient information for recovery, we introduce specific conditions that eliminate these ill-posed scenarios.

**Assumption i (*Nondegenaracy*)** is crucial for linking the dependency structure of the latent variables to the Jacobian of the nonlinear mapping function, following the spirit of methodologies in (Lachapelle et al., 2022; Zheng et al., 2022). This assumption rules out unlikely cases where data samples originate from a highly restrictive subpopulation that spans only a degenerate subspace. The first part of the assumption ensures that there are enough data points such that the Jacobian matrix of the function spans its corresponding support—a condition typically satisfied as the sample size is not extremely small compared to the number of latent variables. The second part is generally mild because the derivative $D_{\hat{\mathbf{z}}}\hat{f} = [D_{\mathbf{z}}f\mathbf{T}]_{i,:}$ naturally resides within its support space $\mathbb{R}^n_{(\hat{\mathcal{F}}_{\hat{\mathbf{z}}})_{i,:}}$. Even in rare instances where the matrix does not align with the support due to specific combinations of values, the assumption remains valid asymptotically. This is because it only requires the existence of one matrix from the entire set $\mathcal{T}$ of matrices that share the support of $\mathbf{T}$. As a result, given the asymptotic nature of the theory, the assumption is almost always satisfied.

**Assumption ii (*Domain Variability*)** (Kong et al., 2022) requires a specific type of variability in the joint distribution of the latent variables $\mathbf{z}$ and the noise variables $\boldsymbol{\epsilon}$ across different domains $u_1$ and $u_2$. These two domains are realizations of a domain variable $\mathbf{u}$, which are observed and labeled. This variability is also independent of $\boldsymbol{\epsilon}$ to introduce the necessary distinction between the noise and latent variables. As verified in (Kong et al., 2022), this condition is typically satisfied in practice, as it is unlikely for the joint distributions under different domains to be very similar. The same as in (Kong et al., 2022), these two domains can differ for different values of A, providing great flexibility. To illustrate, let us consider the following example:

Furthermore, it might be worth noting that the variability required here is significantly less restrictive compared to existing results in the literature. Specifically, many identifiability theorems, even without accounting for general noise, typically require $2n + 1$ domains to identify $n$ latent variables (Hyvärinen et al., 2024). Differently, our theory does not put a hard constraint on requiring $O(n)$ domains, as long as the specific assumption of Domain Variability holds. However, since the conditions are different, the assumption of Domain Variability is not strictly weaker than the previous assumptions.

**Assumption iii (*Structural Sparsity*)** originates from prior work on the identifiability of ICA (without nonparametric noise) (Lachapelle et al., 2022; Zheng et al., 2022). In general, it necessitates the existence of a set of observed variables such that the intersection of their parents singles out itself. Since we do not have any assumptions on the functional type (e.g., post-nonlinear models) or the distributions of the latent variables (e.g., exponential distributions), we can only rely on the hidden structure between latent and observed variables. If certain latent variables are consistently entangled across the generation of all observed variables, identifying them individually becomes impossible without further constraints. Therefore, this assumption provides the necessary structural diversity for nonparametric identifiability. A specific example of when the assumption holds is as follows:

**Example 1.** *Suppose for a latent variable $\mathbf{z}_1$, there exists a set of observed variables, say $\{\mathbf{x}_1, \mathbf{x}_2\}$, s.t., $\mathbf{x}_1$ depends on $\{\mathbf{z}_1, \mathbf{z}_2\}$ and $\mathbf{x}_2$ depends on $\{\mathbf{z}_1, \mathbf{z}_3\}$. Alternatively, there exists a set $\{\mathbf{x}_1, \mathbf{x}_2\}$ s.t. the intersection of their parents singles out $\mathbf{z}_1$. Then the structural sparsity assumption satisfies for $\mathbf{z}_1$. Note that there could be an arbitrary number of observed variables. As long as there exists a subset of them satisfying the condition, the assumption holds for the target latent variable.*

Importantly, the structural sparsity condition only requires a subset of the observed variables—potentially as few as one or two—to satisfy the necessary conditions. This is particularly helpful because our theory allows for a larger number of observed variables compared to latent ones. This enables us to fulfill the required assumptions by incorporating additional observed variables (e.g., adding more microphones in an audio system). In practice, since the true underlying generative process is usually unknown, many assumptions in the literature cannot be directly tested. However, by augmenting the number of observed variables, we may often meet the structural sparsity condition without knowing the ground truth, which significantly increases the applicability.

It might be worth noting that the structural sparsity implies the independence among latent variables $\mathbf{z}$. Specifically, if two latent variables are dependent, it becomes impossible to disentangle one of them by the intersection of a set of observed variables that are influenced by these latent variables. Moreover, for real-world scenarios, it is extremely challenging to make sure that all conditions on the latent data generating process are perfectly satisfied and the distributions are perfectly matched after estimation. Bridging the gap requires a thorough study of the finite sample error and the robustness of the identification, which remains an open challenge in the literature.

## 4 NOISE, DISTORTION, AND STRUCTURE LEARNING

In this section, we present several theoretical developments grounded in the framework of nonlinear representation learning with general noise. First, we investigate the connections between various assumptions, demonstrating that variability can be bypassed by exploiting the parametric form of the noise (Sec. 4.1). Next, we show that even with nonlinear distortion in addition to general noise, the latent variables can still be identified up to element-wise indeterminacies (Sec. 4.2). Finally, we delve into the hidden structure of the data, proving that the causal DAG of a general nonlinear model remains identifiable, even when nonlinear measurement error is present (Sec. 4.3).

### 4.1 LEARNING WITHOUT DISTRIBUTIONAL VARIABILITY

Theorem 1 establishes that latent variables can be identified in the presence of general noise in a nonparametric manner, provided there exists variability in the distributions. While this variability is common in many real-world scenarios, it may not always be present. In some instances, data is generated under stable conditions with no variation between domains. For example, in short-term industrial monitoring systems or continuous physiological monitoring, external conditions may remain constant, eliminating the distributional variability typically required for identifiability.

To address this scenario, we extend our framework by removing the assumption of distributional variability. Specifically, we show that, even in the absence of any distributional variability, latent variables can still be identified. This broadens the applicability of our identifiability theory to environments where such variability is nonexistent. Of course, there is no free lunch. This extension introduces a trade-off: the assumption that noise is additive. This trade-off highlights an important insight—by restricting the form of the noise, as done in many prior works, the whole system becomes less obscure. The resulting data-generating process (Fig. 2) is as follows:

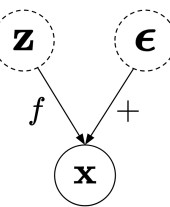

Figure 2: Visualization of Eq. (2).

$$\mathbf{x} = f(\mathbf{z}) + \boldsymbol{\epsilon}, \qquad (2)$$

where we reuse $f$ with a slight abuse of notation to denote a $\mathcal{C}^2$-diffeomorphism $\mathbb{R}^n \to \mathbb{R}^m$ onto its image. The element-wise identifiability is shown as follows with its proof in Appx. B.2.

**Theorem 2.** *Let the observed data be generated by a model defined in Eq. (2). Together with a $\ell_0$ regularization on $\hat{\mathcal{F}}_{\hat{z}}$ during estimation ($\|\hat{\mathcal{F}}_{\hat{z}}\|_0 \leq \|\mathcal{F}_z\|_0$), suppose the following assumptions:*

> *i. (Nondegeneracy) For all $i \in \{1, \ldots, n\}$, there exist points $\{\mathbf{z}^{(\ell)}\}_{\ell=1}^{|(\mathcal{F}_z)_{i,:}|}$ and a matrix $\mathrm{T} \in \boldsymbol{T}$ s.t. $\mathrm{span}\{D_{\mathbf{z}}f(\mathbf{z}^{(\ell)})_{i,:}\}_{\ell=1}^{|(\mathcal{F}_z)_{i,:}|} = \mathbb{R}^n_{(\mathcal{F}_z)_{i,:}}$ and $\left[D_{\mathbf{z}}f(\mathbf{z}^{(\ell)})\mathrm{T}\right]_{i,:} \in \mathbb{R}^n_{(\hat{\mathcal{F}}_{\hat{z}})_{i,:}}$.*

> *ii. (Structural Sparsity) For all $k \in \{1, \ldots, n\}$, there exists a set $\mathcal{C}_k$ s.t. $\bigcap_{i \in \mathcal{C}_k} (\mathcal{F}_z)_{i,:} = \{k\}$.*

*Then latent variables $\mathbf{z}$ are element-wise identifiable (Defn. 1).*

**Remark.** With the additional restriction that noise is additive, Thm. 2 shows that we can remove the requirement for distributional variability. This is natural because additive noise is more easily disentangled and does not influence the derivative of the observed variables with respect to the latent variables, which primarily reflects the structure in the nonlinear case.

**Insights and Implications.** Theorem 2 is particularly relevant for practical scenarios where distributional variability is absent. Moreover, the assumption of additive noise introduces a useful structure, making it easier to separate the noise from the underlying signals. This insight is consistent with prior theoretical work in areas like factor analysis and noisy ICA, emphasizing that constraining the noise form can lead to stronger identifiability results. Thus, Thm. 2 not only extends the applicability of identifiability to scenarios without variability but also serves as a bridge between our proposed theory and existing frameworks.

### 4.2 LEARNING WITH BOTH NOISE AND DISTORTION

Having established the nonparametric identifiability of nonlinear representation learning with general noise (Thm. 1), and further extending this to consider the setting without any variability (Thm. 2), we cover a significant portion of real-world scenarios. However, in many practical settings, noise is not the only challenge complicating data analysis. Data is often subject to additional nonlinear distortions during the measurement process, which apply unknown, element-wise, nonlinear transformations to each variable. For example, in financial markets, real-time price data can be affected by system latency or transaction delays, introducing nonlinear distortions alongside noisy observations.

Figure 3: Visualization of Eqs. (3) and (4).

To address such scenarios, we extend our framework to handle both noise and distortions simultaneously. Specifically, Cor. 1 demonstrates that latent variables can still be identified even when data is subject to both nonparametric noise and nonlinear distortions. The data-generating process (Fig. 3) in this case is as follows:

$$\mathbf{x}^* = f_1(\mathbf{z}, \boldsymbol{\epsilon}), \qquad (3)$$

$$\mathbf{x}_i = f_{2,i}(\mathbf{x}_i^*) + \boldsymbol{\eta}_i, \qquad (4)$$

where $\mathbf{x}^* = (\mathbf{x}_1^*, \ldots, \mathbf{x}_m^*) \in \mathcal{X}^* \subseteq \mathbb{R}^m$ and $\boldsymbol{\eta} = (\boldsymbol{\eta}_1, \ldots, \boldsymbol{\eta}_m) \in \mathcal{Q} \subseteq \mathbb{R}^m$ denote random vectors representing the generated variables before the distortion and another type of noise, respectively. While the noise $\eta$ allows for potential generalization, it is not the central focus here. The mixing function $f_1$ and the distortion function $f_2$ are $\mathcal{C}^2$-diffeomorphisms, and the observed data $\mathbf{x}_i$ is subject to both a nonlinear distortion $f_{2,i}$ and additional noise $\eta_i$. We reuse $f$ as the $\mathcal{C}^2$-diffeomorphism between $(\mathbf{z}, \boldsymbol{\epsilon}, \boldsymbol{\eta})$ and $\mathbf{x}$. The identifiability is provided in Cor. 1 with the proof in Appx. B.3.

**Corollary 1.** *Let the observed data be generated by a model defined in Eqs.* (3) *and* (4)*. Together with a $\ell_0$ regularization on $\hat{\mathcal{F}}_{\hat{z}}$ during estimation ($\|\hat{\mathcal{F}}_{\hat{z}}\|_0 \leq \|\mathcal{F}_z\|_0$), suppose the following assumptions:*

    *i. (Nondegeneracy) For all $i \in \{1, \ldots, n\}$, there exist points $\{(\mathbf{z}, \boldsymbol{\epsilon}, \boldsymbol{\eta})^{(\ell)}\}_{\ell=1}^{|(\mathcal{F}_z)_{i,:}|}$ and a matrix $\mathrm{T} \in \boldsymbol{T}$ s.t. $\mathrm{span}\{D_{\mathbf{z}}f((\mathbf{z}, \boldsymbol{\epsilon}, \boldsymbol{\eta})^{(\ell)})_{i,:}\}_{\ell=1}^{|(\mathcal{F}_z)_{i,:}|} = \mathbb{R}^n_{(\mathcal{F}_z)_{i,:}}$ and $\left[D_{\mathbf{z}}f((\mathbf{z}, \boldsymbol{\epsilon}, \boldsymbol{\eta})^{(\ell)})\mathrm{T}\right]_{i,:} \in \mathbb{R}^n_{(\hat{\mathcal{F}}_{\hat{z}})_{i,:}}$.*

    *ii. (Domain Variability) For any set $A \subseteq \mathcal{Z} \times \mathcal{E}$ with non-zero probability measure that cannot be expressed as $B_{\boldsymbol{\epsilon}} \times B_{\mathbf{z}}$ for any $B_{\boldsymbol{\epsilon}} \subseteq \mathcal{E}$ and $B_{\mathbf{z}} = \mathcal{Z}$, there exist two domains $u_1$ and $u_2$ that are independent of $\boldsymbol{\epsilon}$ s.t.*

$$\int_{(\mathbf{z},\boldsymbol{\epsilon}) \in A} [p(\mathbf{z}, \boldsymbol{\epsilon}|u_1) - p(\mathbf{z}, \boldsymbol{\epsilon}|u_2)] \, d\mathbf{z} \, d\boldsymbol{\epsilon} \neq 0.$$

    *iii. (Structural Sparsity) For all $k \in \{1, \ldots, n\}$, there exists a set $\mathcal{C}_k$ s.t. $\bigcap_{i \in \mathcal{C}_k} (\mathcal{F}_{1z})_{i,:} = \{k\}$.*

*Then latent variables $\mathbf{z}$ are element-wise identifiable (Defn. 1).*

**Insights and Implications.** While noise primarily introduces random fluctuations to the data, nonlinear distortions create systematic, element-wise transformations that alter observed variables in a more persistent and structural manner. Noise tends to be stochastic and, in many cases, can be averaged out over large samples. However, distortions are more systematic and can obscure underlying patterns if not properly disentangled, such as delays and biases. Traditional factor models, which focus solely on noise, often fail to recover the true generative factor due to the entanglement of these distortions with the signal. Corollary 1 addresses this by demonstrating that latent factors can still be identified even in the presence of both general noise and nonlinear distortions. This result may also offer valuable insights for tackling adversarial attacks, where crafted distortions are deliberately introduced to contaminate information and deceive models (Akhtar & Mian, 2018).

### 4.3 STRUCTURE LEARNING WITH NONLINEAR MEASUREMENT ERROR

In the previous sections, we have shown the identifiability of latent variables across various settings. Interestingly, under the umbrella of learning with noise, it is also possible for us to discover the hidden structure among variables even in the presence of general measurement error. We first introduce the data-generating process (Fig. 4) as follows:

$$\mathbf{z} = f_1(\boldsymbol{\xi}), \tag{5}$$

$$\mathbf{x}_i = f_{2,i}(\mathbf{z}_i) + \boldsymbol{\eta}_i, \tag{6}$$

where $\mathbf{z}_i$ represents the latent variables, $\boldsymbol{\xi} = (\boldsymbol{\xi}_1, \ldots, \boldsymbol{\xi}_{n_u}) \in \mathcal{U} \subseteq \mathbb{R}^{n_u}$ denotes noise, and $\boldsymbol{\eta}_i$ represents the nonlinear measurement errors. Functions $f_1$ and $f_2$ are $\mathcal{C}^2$-diffeomorphisms, and $\mathbf{x}_i$ is the observed variable generated from the latent variable and the nonlinear measurement error. Consistent with the previous theorems, we denote $\mathrm{G}_{f_1^{-1}}$ and $\mathrm{G}_{\hat{f}^{-1}}$ as the binary matrices with the same support as $\mathrm{supp}(D_z f_1^{-1})$ and $\mathrm{supp}(D_{\hat{x}} \hat{f}^{-1})$, respectively. We denote by $\boldsymbol{T}_\xi$ the set of matrices that share the same support as the matrix-valued function $\mathbf{T}_\xi(\cdot)$ in the equation $D_{\hat{\xi}} \hat{f}(\cdot) = D_{\boldsymbol{\xi}} f(\cdot) \mathbf{T}_\xi(\cdot)$, where $\mathbf{T}_\xi(\cdot)$ is a matrix-valued function. We reuse $f$ as the $\mathcal{C}^2$-diffeomorphism between $(\boldsymbol{\xi}, \boldsymbol{\eta})$ and $\mathbf{x}$.

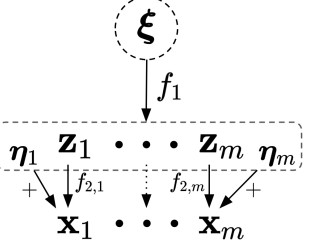

Figure 4: Visualization of Eqs. (5) and (6).

**Theorem 3.** *Let the observed data be generated by a model defined in Eqs.* (5) *and* (6). *Suppose for each* $i \in \{1, \ldots, m\}$, *there exist* $\{(\boldsymbol{\xi}, \boldsymbol{\eta})^{(\ell)}\}_{\ell=1}^{|\mathcal{F}_{\xi_{i,:}}|}$ *and a matrix* $\mathrm{T}_\xi \in \boldsymbol{T}_\xi$ *s.t.* $\mathrm{span}\{D_{\boldsymbol{\xi}} f((\boldsymbol{\xi}, \boldsymbol{\eta})^{(\ell)})_{i,:}\}_{\ell=1}^{|(\mathcal{F}_\xi)_{i,:}|} = \mathbb{R}^m_{(\mathcal{F}_\xi)_{i,:}}$ *and* $\left[ D_{\boldsymbol{\xi}} f((\boldsymbol{\xi}, \boldsymbol{\eta})^{(\ell)}) \mathrm{T}_\xi \right]_{i,:} \in \mathbb{R}^m_{(\hat{\mathcal{F}}_{\hat{\xi}})_{i,:}}$. *Then* $\mathrm{G}_{\hat{f}^{-1}} = P\mathrm{G}_{f_1^{-1}}$ *for a permutation matrix* $P$ *together with a $\ell_0$ regularization on $\hat{\mathcal{F}}_{\hat{z}}$ during estimation* ($\|\hat{\mathcal{F}}_{\hat{z}}\|_0 \leq \|\mathcal{F}_z\|_0$).

> **Remark.** Theorem 3 demonstrates that, under the nondegeneracy assumption—which prevents ill-posed cases where the samples fail to span its support space—the structure linking exogenous noises $\boldsymbol{\xi}$ and latent variables $\mathbf{z}$ remains identifiable up to permutation. This structural identification offers valuable insights into the mixing processes of existing factor models, such as ICA.

The identifiability of the mixing structure provides theoretical guarantees of discovering the hidden connection underlying the data-generating process. At the same time, similar to (Shimizu et al., 2006), the mixing structure also sheds light on the underlying causal graph under appropriate assumptions. If we assume that the noises $\boldsymbol{\xi}$ are independent and the dimensions of $\boldsymbol{\xi}$ and $\mathbf{z}$ are the same, i.e., $n_u = n$, we can transfer Eq. 5 as a Structural Causal Model (SCM) by considering $\boldsymbol{\xi}$ as the exogenous noise, which is equivalent to

$$\mathbf{z}_i = f_{1,i}(\mathbf{Pa}(\mathbf{z}_i), \boldsymbol{\xi}_i), \quad \forall i, \qquad (7)$$

where we denote the set of parents of $\mathbf{z}_i$ as $\mathbf{Pa}(\mathbf{z}_i) \subset \mathcal{Z}$. This results in a set of edges that forms a causal graph, which, under the acyclicity assumption, is a DAG (Fig. 5). The adjacency matrix of a causal DAG is defined as follows:

Figure 5: Visualization of Eqs. (6) and (7). The structure among $\mathbf{z}$ is a causal DAG for $\{\mathbf{z}_1, \mathbf{z}_2, \mathbf{z}_3\}$.

**Definition 2.** *The binary matrix $\mathcal{A}$ denotes the adjacency structure of a causal DAG, i.e., $\mathcal{A}_{i,j} = 0$ if and only if $\mathbf{z}_j \notin \textbf{Pa}(\mathbf{z}_i)$. In addition, the rows of $\mathcal{A}$ are ordered to make it strictly lower-triangular.*

Then we have the following results for the identifiability of the underlying directed acyclic graph (DAG) among variables $\mathbf{z}$.

**Assumption 1.** *(Structural Faithfulness (Reizinger et al., 2022)) The set of samples that induce additional zeroes (i.e., a sparser DAG) in the Jacobians $D_{\boldsymbol{\xi}} f_1$, $D_{\mathbf{z}} f_1^{-1}$ has zero measure, i.e., both Jacobians describe the sparsity structure of the underlying SCM with probability one. Alternatively, this structural independencies are reflected in a functional form via $D_{\boldsymbol{\xi}} f_1 / D_{\mathbf{z}} f_1^{-1}$.*

Loosely speaking, the faithfulness assumption ensures that no edges are accidentally canceled due to specific parameter combinations, a common condition in causal discovery (Zhang, 2013) and we include the version formalized in (Reizinger et al., 2022) here for the ease of reference. The identifiability results are as follows:

**Proposition 1.** *[Reizinger et al. (2022)] The matrix $\mathrm{G}_{f_1^{-1}}$ is structurally equivalent to $\mathbf{I}_n - \mathrm{A}$ for a structurally faithful SCM (Assump. 1), i.e., $\forall i, j, \left( \mathrm{G}_{f_1^{-1}} \right)_{ij} = 0 \Leftrightarrow (\mathbf{I}_n - \mathcal{A})_{ij} = 0$.*

**Proposition 2.** *Suppose the assumptions in Theorem 3 and Proposition 1 hold, then $\mathcal{A}$ in Eq. 7 is identifiable.*

> **Remark.** We demonstrate that the underlying causal structure of general SCMs can be identified despite nonlinear measurement distortions. The key intuition is that the structural equivalence between the mixing matrix and the causal graph, combined with the acyclicity of the DAG, eliminates the permutation indeterminacy of the mixing structure.

**Insights and Implications.** Causal discovery aims to find the causal structure underlying the data (Spirtes et al., 2000) based on pure observation. Traditional results on the identifiability of causal discovery usually make parametric assumptions such as post-nonlinear or additive noise models to identify the underlying causal DAG. Additionally, most previous methods assume that the observed

values directly correspond to the variables of interest. While some works have considered structure identification in the presence of measurement error (Zhang et al., 2018; Dai et al., 2022), these approaches impose parametric assumptions on both the SCM (linear non-Gaussian models) and the measurement error (linear distortions). These conditions are important given the challenges of learning causal structure without any interventional data, but still somehow limit the applicability in complicated real-world scenarios. As a result, the question of under what conditions we can identify the causal graph for general nonlinear models with nonparametric measurement error remains open.

Fortunately, the connection between the mixing matrix and the causal DAG brings us the opportunity to study the identifiability of this challenging problem. This is because all relations among causal variables (Eq. (7)) can be considered as how exogenous affect variables of interests if we take a view of the whole system (Eq. (5)). This connection has been firstly used in the seminal work of Shimizu et al. (2006) to identify linear non-Gaussian models based on linear ICA. The key insight is that, given the acyclicity constraints, the recovered mixing matrix—despite permutation indeterminacy—can be uniquely transferred to an adjacency matrix of a causal DAG. More recently, Reizinger et al. (2022) extend it to the nonlinear case by bridging the Jacobian of the mixing function $f_1$ in Eq. (7) and the causal structure (A). Thus, our results on the general factor model with nonlinear distortion and additive noise (Thm. 3) inherently lead to the identifiability of the underlying causal structure even in the presence of nonlinear measurement error (Prop. 2). This is exciting since causal discovery with general functional relations has long been an open problem, and the inclusion of nonlinear distortions adds further complexity. Naturally, this generalization requires an additional nondegeneracy condition on the latent factor model, much like how Reizinger et al. (2022) leverage identifiability conditions for nonlinear ICA in the context of nonlinear causal discovery. We believe that the revealed insight may suggest a different direction toward more general solutions for understanding the (causal) structure underlying the data.

## 5 EXPERIMENTS

To assess the identifiability of nonlinear representations in the presence of general noise, we perform experiments on both synthetic and real-world datasets. While numerous studies have demonstrated the empirical success of learning semantically meaningful representations from noisy data (e.g., through denoising techniques), our experiments aim to complement these findings by rigorously validating our theoretical framework under the specified conditions. For a broader range of applications, we refer the reader to the extensive body of prior empirical research (Tian et al., 2020).

**Setup.** The training process uses a General Incompressible-flow Network (GIN) (Sorrenson et al., 2020), a flow-based generative model, to optimize the objective function $\mathcal{L}(\theta)$, defined as: $\mathcal{L}(\theta) = \mathbb{E}_{(\mathbf{x},\mathbf{u})} \left[ \log p_{\hat{f}^{-1}}(\mathbf{x} \mid \mathbf{u}) \right] - \lambda \mathbf{R}$, where $\lambda$ is a regularization term and $\mathbf{R}$ represents the $\ell_1$-norm regularization applied to the Jacobian of $\hat{f}$. The dataset is denoted as $\mathcal{D} = \{(\mathbf{x}^{(1)}, \mathbf{u}^{(1)}), \ldots, (\mathbf{x}^{(N)}, \mathbf{u}^{(N)})\}$, with $N$ samples, where each data point $\mathbf{x}^{(i)}$ corresponds to a domain $\mathbf{u}^{(i)}$. During training, latent variables are drawn from two multivariate Gaussian distributions to satisfy the variability condition, while noise is also sampled from a separate multivariate Gaussian, with means sampled uniformly from the range $[-5, 5]$ and variances sampled uniformly from $[0.5, 2.5]$. The noise and latent variables are concatenated in the flow model, ensuring the nonparametric nature of the noise. In scenarios with two domains $\mathbf{u} = u_1$ and $\mathbf{u} = u_2$, the domain index is provided during the estimation process. We perform experiments across 10 independent trials, each initialized with a different random seed. Further experimental details are in Appx. D.1.

**Simulations.** We perform an ablation study to evaluate the necessity of the proposed assumptions. For the model grounded in our identifiability theory (*Ours*), all conditions required by Theorem 1 are satisfied in the data-generating process. In contrast, the baseline model (*Base*) violates key assumptions, particularly those related to structural sparsity (by a fully connected structure) and variability (by sampling from a single domain). In our experimental setup, half of the observed variables ($m/2$) correspond to latent variables, while the remaining half are noise variables. Datasets are generated according to these specifications, with further details provided in Appx. D.1. To assess model performance, we employ the mean correlation coefficient (MCC) between the true latent variables $\mathbf{z}$ and their estimates $\hat{\mathbf{z}}$, following the evaluation metrics used in prior works (Hyvärinen & Morioka, 2016; Lachapelle et al., 2022). We also extend our experiments to different numbers of

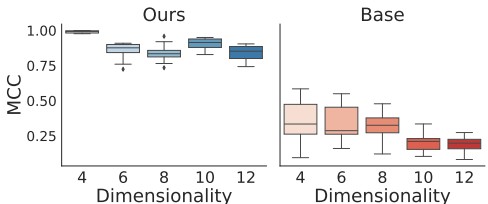
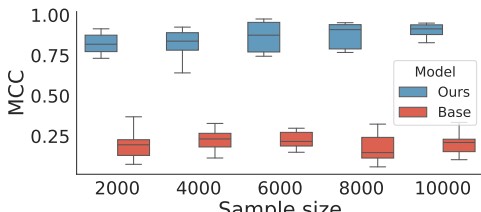

Figure 6: Identification of latent variables w.r.t. different $m$, where $n = m/2$.

Figure 7: Identification w.r.t. different sample sizes with the dimensions $m = 10, n = 5$.

variables to evaluate the model's scalability across various settings. Additionally, we test the models with varying sample sizes to study both the asymptotic behavior of our theory and its robustness to sample size variation.

The results for each model are presented in Figs. 6 and 7. It is evident that when the proposed assumptions are satisfied (*Ours*), the models consistently achieve higher MCC scores compared to the *Base* model. This confirms that latent variables can indeed be identified from observations generated by an unknown nonlinear process, even in the presence of general noise. Furthermore, our theory-based model shows stable performance across datasets with different numbers of variables, whereas the baseline model's performance degrades as scalability increases. Finally, as sample sizes grow, we observe a steady improvement in the model's performance, supporting the asymptotic properties of our theory.

**Real-world experiments.** In Appx. D.2, we conduct additional experiments to evaluate the practical applicability of our approach in real-world scenarios. These experiments are performed on two real-world image datasets: one featuring various types of clothing and another consisting of handwritten digits. Our findings indicate that even in these real-world settings, we can successfully identify semantically meaningful generative factors from the raw observational pixel data. These results further demonstrate the practical relevance and applicability of our theory. Importantly, several practical challenges persist. For example, human interpretations of latent factors are often guided by intuition, yet there is no guarantee that the true generative process aligns with these interpretations. Certain latent factors may inherently appear entangled or lack clear semantic meaning from a human perspective, even if they represent statistically independent components of the generative mechanism. Furthermore, practical constraints, such as finite sample errors, pose additional challenges to achieving perfect recovery of the hidden factors. Please refer to Figs. 9, 10 and 11 for details.

## 6 CONCLUSION

In this paper, we establish theoretical guarantees for nonlinear representation learning in the presence of general noise. Specifically, we prove that latent generating factors can be identified up to trivial indeterminacies, without imposing parametric constraints on either the generating process or the noise. Within this general framework, we explore the relationships between various conditions, highlighting the inherent trade-offs. Moreover, since real-world observations may involve not only noise but also nonlinear distortions, we extend the proposed nonparametric identifiability to account for both. Finally, we demonstrate that the underlying causal structure is also identifiable even with nonlinear measurement errors. Theoretical results are validated in both synthetic and real-world settings.

While we demonstrate nonparametric identifiability for learning with noise, several related questions remain open. One intriguing direction involves scenarios where the generating process includes more latent variables than observed ones, making the function non-injective. In such cases, some information is inevitably lost, raising the critical question of which part of the hidden world can still be recovered. Additionally, our theory focuses on asymptotic guarantees, leaving the finite-sample regime unexplored. Investigating sample complexity in this context, though distinct from our current focus, could be interesting as well. Many questions remain, but for now, we can confidently answer the question posed in the introduction:

*Yes, machines can reliably reveal the hidden world amid the chaos of noise.*

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

# Appendices

## Table of Contents

## A    NOTATION SUMMARY

In this section, we summarize the key notations used throughout the paper for clarity and reference.

**Variables and Spaces**

- $\mathbf{x} = (\mathbf{x}_1, \ldots, \mathbf{x}_m) \in \mathcal{X} \subseteq \mathbb{R}^m$: The observed data vector comprising $m$ observed variables.
- $\mathbf{z} = (\mathbf{z}_1, \ldots, \mathbf{z}_n) \in \mathcal{Z} \subseteq \mathbb{R}^n$: The latent variable vector comprising $n$ latent variables.
- $\boldsymbol{\epsilon} = (\boldsymbol{\epsilon}_1, \ldots, \boldsymbol{\epsilon}_{n_e}) \in \mathcal{E} \subseteq \mathbb{R}^{n_e}$: The independent noise vector comprising $n_e$ noise variables.
- $\boldsymbol{\eta} = (\boldsymbol{\eta}_1, \ldots, \boldsymbol{\eta}_m) \in \mathcal{Q} \subseteq \mathbb{R}^m$: Another type of noise vector associated with distortion or measurement error.
- $\boldsymbol{\xi} = (\boldsymbol{\xi}_1, \ldots, \boldsymbol{\xi}_{n_u}) \in \mathcal{U} \subseteq \mathbb{R}^{n_u}$: The independent noise vector used in the structural causal model (SCM) setting.
- $\hat{\mathbf{z}}, \hat{\boldsymbol{\epsilon}}, \hat{\boldsymbol{\eta}}, \hat{\boldsymbol{\xi}}$: Estimated versions of the variables, denoted with a hat to represent estimated quantities.

**Functions**

- **Theorem 1:** Data generating process:
$$\mathbf{x} = f(\mathbf{z}, \boldsymbol{\epsilon}),$$
where $f : (\mathbf{z}, \boldsymbol{\epsilon}) \to \mathbf{x}$ is the mixing function mapping latent variables and noise to the observed data.

- **Theorem 2:** Data generating process:
$$\mathbf{x} = f(\mathbf{z}) + \boldsymbol{\epsilon}.$$
where $f : \mathbf{z} \to \mathbf{x}$ is the mixing function mapping latent variables to the observed data with additive noise.

- **Corollary 1:** Data generating process:
$$\mathbf{x}^* = f_1(\mathbf{z}, \boldsymbol{\epsilon}),$$
$$\mathbf{x}_i = f_{2,i}(\mathbf{x}_i^*) + \boldsymbol{\eta}_i,$$
where $f_1 : (\mathbf{z}, \boldsymbol{\epsilon}) \to \mathbf{x}^*$ maps latent variables and noise to an intermediate variable $\mathbf{x}^*$, and $f_2 : \mathbf{x}^* \to \mathbf{x}$ is an element-wise transformation applied to $\mathbf{x}^*$ with additional noise $\boldsymbol{\eta}$.

- **Theorem 3, Proposition 2:** Data generating process:
$$\mathbf{z} = f_1(\boldsymbol{\xi}),$$
$$\mathbf{x}_i = f_{2,i}(\mathbf{z}_i) + \boldsymbol{\eta}_i,$$
where $f_1 : \boldsymbol{\xi} \to \mathbf{z}$ maps independent noises to latent variables, and $f_2 : \mathbf{z} \to \mathbf{x}$ is an element-wise transformation applied to each latent variable $\mathbf{z}_i$ with additional noise $\boldsymbol{\eta}_i$.

**Jacobians and Supports**

- $D_{\mathbf{z}}f$: The Jacobian matrix of the function $f$ with respect to $\mathbf{z}$.
- $D_{\boldsymbol{\xi}}f$: The Jacobian matrix of the function $f$ with respect to $\boldsymbol{\xi}$.
- $D_{\hat{\mathbf{z}}}\hat{f}$: The Jacobian matrix of the estimated function $\hat{f}$ with respect to $\hat{\mathbf{z}}$.
- $\mathrm{supp}(\mathbf{M})$: The support of a matrix $\mathbf{M} \in \mathbb{R}^{m \times n}$, defined as $\{(i,j) \mid \mathbf{M}_{i,j} \neq 0\}$.
- $\mathrm{supp}(\mathbf{M}(\Theta))$: The support of a matrix-valued function $\mathbf{M} : \Theta \to \mathbb{R}^{m \times n}$, defined as $\{(i,j) \mid \exists \theta \in \Theta, \mathbf{M}(\theta)_{i,j} \neq 0\}$.

**Index Sets and Subspaces**

- $\mathcal{S} \subseteq \{1, \ldots, d\}$: A subset of indices used to specify subspaces or supports.
- $\mathbb{R}_{\mathcal{S}}^d := \{\mathbf{v} \in \mathbb{R}^d \mid i \notin \mathcal{S} \implies \mathbf{v}_i = 0\}$: The subspace of $\mathbb{R}^d$ specified by index set $\mathcal{S}$.
- $|\mathcal{S}|$: The cardinality (number of elements) of the set $\mathcal{S}$.
- For a matrix $\mathbf{M} \in \mathbb{R}^{m \times n}$:
  - $\mathbf{M}_{i,:}$: The $i$-th row of $\mathbf{M}$.
  - $\mathbf{M}_{:,j}$: The $j$-th column of $\mathbf{M}$.
- For a set of indices $\mathcal{S} \subseteq \{1, \ldots, m\} \times \{1, \ldots, n\}$:
  - $\mathcal{S}_{i,:} := \{j \mid (i, j) \in \mathcal{S}\}$: The set of column indices associated with row $i$.
  - $\mathcal{S}_{:,j} := \{i \mid (i, j) \in \mathcal{S}\}$: The set of row indices associated with column $j$.

**Graphs and Matrices**

- $\mathcal{F}_z := \text{supp}(D_{\mathbf{z}} f)$: The support of the Jacobian of $f$ with respect to $\mathbf{z}$.
- $\mathcal{F}_\xi := \text{supp}(D_{\boldsymbol{\xi}} f)$: The support of the Jacobian of $f$ with respect to $\boldsymbol{\xi}$.
- $\mathcal{F}_{\hat{\xi}} := \text{supp}(D_{\hat{\boldsymbol{\xi}}} \hat{f})$: The support of the Jacobian of $\hat{f}$ with respect to $\hat{\boldsymbol{\xi}}$.
- $\boldsymbol{T}, \boldsymbol{T}_\xi$: Sets of matrices that share the same support as the matrix-valued functions $\mathbf{T}(\cdot)$ and $\mathbf{T}_\xi(\cdot)$, respectively, appearing in equations like $D_{\hat{\mathbf{z}}} \hat{f}(\cdot) = D_{\mathbf{z}} f(\cdot) \, \mathbf{T}(\cdot)$.
- $\mathbf{T}(\cdot), \mathbf{T}_\xi(\cdot)$: Matrix-valued functions whose supports define the sets $\boldsymbol{T}$ and $\boldsymbol{T}_\xi$.
- $\mathrm{T} \in \boldsymbol{T}$, $\mathrm{T}_\xi \in \boldsymbol{T}_\xi$: Specific matrices within these sets.
- $\mathrm{G}_{f_1^{-1}}$, $\mathrm{G}_{\hat{f}_1^{-1}}$: Binary matrices with the same support as $\text{supp}(D_{\mathbf{z}} f_1^{-1})$ and $\text{supp}(D_{\hat{\mathbf{z}}} \hat{f}_1^{-1})$, respectively.
- $\mathcal{A}$: The adjacency matrix of a directed acyclic graph (DAG) representing the causal structure among latent variables. It is defined as $\mathcal{A}_{i,j} = 0$ if and only if $\mathbf{z}_j \notin \mathbf{Pa}(\mathbf{z}_i)$, where $\mathbf{Pa}(\mathbf{z}_i)$ denotes the set of parents of $\mathbf{z}_i$.

**Permutations and Diagonal Matrices**

- $P$: A permutation matrix corresponding to a reordering of variables.
- $D_1, D_2$: Diagonal matrices, often used in element-wise transformations involving Jacobians.

**Domains and Probability Measures**

- $\Theta$: The parameter space for matrix-valued functions like $\mathbf{M}(\theta)$.
- $u_1, u_2$: Domains or conditions under which distributions are considered, particularly in variability assumptions.
- $p(\mathbf{z}, \boldsymbol{\epsilon} \mid u)$: The joint probability density function of $\mathbf{z}$ and $\boldsymbol{\epsilon}$ conditioned on domain $u$.

**Miscellaneous Notations**

- $\text{span}\{\cdot\}$: The linear span of a set of vectors.
- $\text{adj}(\mathbf{M})$: The adjugate (adjoint) of a square matrix $\mathbf{M}$.
- $\det(\mathbf{M})$: The determinant of the square matrix $\mathbf{M}$.
- $S_n$: The set of all permutations of $\{1, 2, \ldots, n\}$.
- $\text{sgn}(\sigma)$: The sign (parity) of the permutation $\sigma$, equal to $+1$ for even permutations and $-1$ for odd permutations.
- $\{\hat{\cdot}\}$: The hat symbol denotes estimated quantities, such as $\hat{f}$, $\hat{\mathbf{z}}$, and other estimated variables or functions.
- $\mathbf{I}_n$: The $n \times n$ identity matrix.

**Functions and Equations Specific to the SCM Setting**

- $f_{1,i}$: The function defining the $i$-th structural equation in the SCM, mapping from parents and noise to the latent variable $\mathbf{z}_i$.

- $\mathbf{z}_i = f_{1,i}(\mathbf{Pa}(\mathbf{z}_i), \boldsymbol{\xi}_i)$: The structural causal model equation for latent variable $\mathbf{z}_i$.

- $\mathbf{Pa}(\mathbf{z}_i)$: The set of parents of $\mathbf{z}_i$ in the causal graph.

- $\mathrm{G}_{f_1^{-1}}$ structurally equivalent to $\mathbf{I}_n - \mathcal{A}$: Indicates that the support of the inverse Jacobian of $\hat{f}_1$ reflects the structure of the DAG.

**Submatrices and Indexing**

- For matrices $\mathbf{M} \in \mathbb{R}^{n \times n}$ and indices $i$, $j$:

  - $\mathbf{M}_{[n]\setminus i, [n]\setminus j}$: The submatrix of $\mathbf{M}$ obtained by removing the $i$-th row and $j$-th column.

- $[n]$: Denotes the set $\{1, 2, \ldots, n\}$.

# B PROOFS

## B.1 PROOF OF THEOREM 1

Before the main proof, let us first introduce a lemma from (Kong et al., 2022). The proof of the lemma is directly based on steps 1, 2, and 3 in the proof of Theorem 4.2 in (Kong et al., 2022). We include its proof for the ease of reference.

**Lemma 1.** *(Kong et al., 2022) Let the observed data be a large enough sample generated by a model defined in Eq. (1). Suppose for any set $A \subseteq \mathcal{Z} \times \mathcal{E}$ with non-zero probability measure that cannot be expressed as $B_{\boldsymbol{\epsilon}} \times B_{\mathbf{z}}$ for any $B_{\boldsymbol{\epsilon}} \subseteq \mathcal{E}$ and $B_{\mathbf{z}} = \mathcal{Z}$, there exist two domains $u_1$ and $u_2$ that are independent of $\boldsymbol{\epsilon}$ s.t.*

$$\int_{(\mathbf{z},\boldsymbol{\epsilon}) \in A} [p(\mathbf{z}, \boldsymbol{\epsilon}|u_1) - p(\mathbf{z}, \boldsymbol{\epsilon}|u_2)]\, d\mathbf{z}\, d\boldsymbol{\epsilon} \neq 0.$$

*Then the partial derivative of $\boldsymbol{\epsilon}$ w.r.t. $\hat{\mathbf{z}}$ is zero.*

*Proof.* Please note that the proof is from steps 1, 2, and 3 in the proof of Theorem 4.2 in (Kong et al., 2022), and we just change the notation to be consistent in our setting. Because domains are independent of noise, for any $A_{\boldsymbol{\epsilon}} \subseteq \mathcal{E}$, we have the following relation for any $u_1, u_2 \in \mathcal{U}$ represents the domain variable.

$$\mathbb{P}\left[\hat{f}_{n+1:}^{-1}(\hat{\mathbf{x}}) \in A_{\boldsymbol{\epsilon}}|u_1\right] = \mathbb{P}\left[\hat{f}_{n+1:}^{-1}(\hat{\mathbf{x}}) \in A_{\boldsymbol{\epsilon}}|u_2\right]. \tag{8}$$

Because the observed distributions are matched for identification, we further have

$$\mathbb{P}\left[\hat{f}_{n+1:}^{-1}(\mathbf{x}) \in A_{\boldsymbol{\epsilon}}|u_1\right] = \mathbb{P}\left[\hat{f}_{n+1:}^{-1}(\mathbf{x}) \in A_{\boldsymbol{\epsilon}}|u_2\right]. \tag{9}$$

Let the function $h := \hat{f}^{-1} \circ f$ denote the map between estimated and ground-truth concepts. Denote $h_{\boldsymbol{\epsilon}} := h_{n+1:} : \mathcal{Z} \times \mathcal{E} \to \mathcal{E}$. It follows that

$$\mathbb{P}[h_{\boldsymbol{\epsilon}}((\mathbf{z}, \boldsymbol{\epsilon})) \in A_{\boldsymbol{\epsilon}}|u_1] = \mathbb{P}[h_{\boldsymbol{\epsilon}}((\mathbf{z}, \boldsymbol{\epsilon})) \in A_{\boldsymbol{\epsilon}}|u_2], \tag{10}$$

which is equivalent to

$$\int_{(\mathbf{z},\boldsymbol{\epsilon}) \in h_{\boldsymbol{\epsilon}}^{-1}(A_{\mathbf{z}_{\boldsymbol{\epsilon}}})} p_{(\mathbf{z},\boldsymbol{\epsilon})|\mathbf{u}}((\mathbf{z}, \boldsymbol{\epsilon})|u_1)\, d\mathbf{z} d\boldsymbol{\epsilon} = \int_{(\mathbf{z},\boldsymbol{\epsilon}) \in h_c^{-1}(A_{\mathbf{z}_c})} p_{(\mathbf{z},\boldsymbol{\epsilon})|\mathbf{u}}((\mathbf{z}, \boldsymbol{\epsilon})|u_2)\, d\mathbf{z} d\boldsymbol{\epsilon}. \tag{11}$$

Since $\mathbf{z}$ and $\boldsymbol{\epsilon}$ are conditionally independent given $\mathbf{u}$, we have

$$\int_{(\mathbf{z},\boldsymbol{\epsilon}) \in h_{\boldsymbol{\epsilon}}^{-1}(A_{\mathbf{z}_{\boldsymbol{\epsilon}}})} p_{\mathbf{z}|u_1}(\mathbf{z}|u_1) p_{\boldsymbol{\epsilon}}(\boldsymbol{\epsilon})\, d\mathbf{z} d\boldsymbol{\epsilon} = \int_{(\mathbf{z},\boldsymbol{\epsilon}) \in h_c^{-1}(A_{\mathbf{z}_{\boldsymbol{\epsilon}}})} p_{\mathbf{z}|u_2}(\mathbf{z}|u_2) p_{\boldsymbol{\epsilon}}(\boldsymbol{\epsilon})\, d\mathbf{z} d\boldsymbol{\epsilon}. \tag{12}$$

We aim to prove that for all $\epsilon \in \mathcal{E}$ and $r \in \mathbb{R}^+$, it follows that $h_\epsilon^{-1}(\mathcal{B}_r(\epsilon)) = \mathcal{Z} \times B_\epsilon^+$, where $\mathcal{B}_r(\epsilon) := \{\epsilon' \in \mathcal{E} : \|\epsilon' - \epsilon\|^2 < r\}$, $B_\epsilon^+ \neq \emptyset$, and $B_\epsilon^+ \subseteq \mathcal{E}$.

First, note that because $\mathcal{B}_r(\epsilon)$ is open and $h_\epsilon(\cdot)$ is continuous, the preimage $h_\epsilon^{-1}(\mathcal{B}_r(\epsilon))$ is open. Additionally, due to the continuity of $h(\cdot)$ and the matched observation distributions (i.e., $\forall u' \in \mathcal{U}$, $\mathbb{P}[\{\mathbf{x} \in A_\mathbf{x}\} \mid u'] = \mathbb{P}[\{\hat{\mathbf{x}} \in A_\mathbf{x}\} \mid u']$), it follows from (Klindt et al., 2020) that $h(\cdot)$ is bijective. This implies that $h_\epsilon^{-1}(\mathcal{B}_r(\epsilon))$ is non-empty. Therefore, $h_\epsilon^{-1}(\mathcal{B}_r(\epsilon))$ is both non-empty and open.

Suppose there exists $A_\epsilon^* := \mathcal{B}_{r^*}(\epsilon^*)$, where $\epsilon^* \in \mathcal{E}$ and $r^* \in \mathbb{R}^+$, such that

$$B^* := \left\{ (\mathbf{z}, \epsilon) \in \mathcal{Z} \times \mathcal{E} : (\mathbf{z}, \epsilon) \in h_\epsilon^{-1}(A_\epsilon^*), \ \mathcal{Z} \times \{\epsilon\} \not\subseteq h_\epsilon^{-1}(A_\epsilon^*) \right\} \neq \emptyset. \tag{13}$$

Intuitively, $B^*$ contains the subset of the preimage $h_\epsilon^{-1}(A_\epsilon^*)$ where $\mathbf{z}$ cannot take all values in $\mathcal{Z}$ for a given $\epsilon$. Only certain values of $\mathbf{z}$ can produce specific outputs of $h_\epsilon(\cdot)$, indicating that $h_\epsilon(\cdot)$ depends on $\mathbf{z}$.

The integral in Eq. (12) with such an $A_\epsilon^*$ is as follows:

$$\int_{(\mathbf{z}, \epsilon) \in h_\epsilon^{-1}(A_\epsilon^*)} \left[ p_{\mathbf{z}|u}(\mathbf{z}|u_1) - p_{\mathbf{z}|u}(\mathbf{z}|u_2) \right] p_\epsilon(\epsilon) \, d\mathbf{z} \, d\epsilon \tag{14}$$

$$= \underbrace{\int_{(\mathbf{z}, \epsilon) \in h_\epsilon^{-1}(A_\epsilon^*) \setminus B^*} \left[ p_{\mathbf{z}|u}(\mathbf{z}|u_1) - p_{\mathbf{z}|u}(\mathbf{z}|u_2) \right] p_\epsilon(\epsilon) \, d\mathbf{z} \, d\epsilon}_{T_1} \tag{15}$$

$$+ \underbrace{\int_{(\mathbf{z}, \epsilon) \in B^*} \left[ p_{\mathbf{z}|u}(\mathbf{z}|u_1) - p_{\mathbf{z}|u}(\mathbf{z}|u_2) \right] p_\epsilon(\epsilon) \, d\mathbf{z} \, d\epsilon}_{T_2}. \tag{16}$$

If $h_\epsilon^{-1}(A_\epsilon^*) \setminus B^* = \emptyset$, then $T_1 = 0$.

Otherwise, by definition, we can rewrite $h_\epsilon^{-1}(A_\epsilon^*) \setminus B^*$ as $\mathcal{Z} \times C_\epsilon^*$, where $C_\epsilon^* \neq \emptyset$ and $C_\epsilon^* \subseteq \mathcal{E}$. With this expression, it follows that

$$T_1 = \int_{(\mathbf{z}, \epsilon) \in \mathcal{Z} \times C_\epsilon^*} \left[ p_{\mathbf{z}|u}(\mathbf{z}|u_1) - p_{\mathbf{z}|u}(\mathbf{z}|u_2) \right] p_\epsilon(\epsilon) \, d\mathbf{z} \, d\epsilon \tag{17}$$

$$= \int_{\epsilon \in C_\epsilon^*} p_\epsilon(\epsilon) \left( \int_{\mathbf{z} \in \mathcal{Z}} \left[ p_{\mathbf{z}|u}(\mathbf{z}|u_1) - p_{\mathbf{z}|u}(\mathbf{z}|u_2) \right] d\mathbf{z} \right) d\epsilon \tag{18}$$

$$= \int_{\epsilon \in C_\epsilon^*} p_\epsilon(\epsilon) \, (1 - 1) \, d\epsilon = 0. \tag{19}$$

Therefore, in both cases, $T_1$ evaluates to zero.

Now, we address $T_2$. As discussed, $h_\epsilon^{-1}(A_\epsilon^*)$ is open and non-empty. Because of the continuity of $h_\epsilon(\cdot)$, for every $(\mathbf{z}, \epsilon) \in B^*$, there exists $r(\epsilon) \in \mathbb{R}^+$ such that $\mathcal{B}_{r(\epsilon)}(\epsilon) \subseteq B^*$.

Since $p_\epsilon(\epsilon) > 0$ over $\mathcal{E}$, we have

$$\mathbb{P}\left[ (\mathbf{z}, \epsilon) \in B^* \mid u' \right] \geq \mathbb{P}\left[ (\mathbf{z}, \epsilon) \in \mathcal{Z} \times \mathcal{B}_{r(\epsilon)}(\epsilon) \mid u' \right] > 0, \quad \forall u' \in \mathcal{U}. \tag{20}$$

The assumption in the lemma indicates that there exist $u_1^*, u_2^* \in \mathcal{U}$ such that

$$T_2 = \int_{(\mathbf{z}, \epsilon) \in B^*} \left[ p_{\mathbf{z}|u}(\mathbf{z}|u_1^*) - p_{\mathbf{z}|u}(\mathbf{z}|u_2^*) \right] p_\epsilon(\epsilon) \, d\mathbf{z} \, d\epsilon \neq 0. \tag{21}$$

This inequality holds because the difference $p_{\mathbf{z}|u}(\mathbf{z}|u_1^*) - p_{\mathbf{z}|u}(\mathbf{z}|u_2^*)$ is not identically zero over $B^*$, and $p_\epsilon(\epsilon) > 0$. Therefore, for such $A_\epsilon^*$, we have $T_1 + T_2 \neq 0$. Therefore, we have

$$\int_{(\mathbf{z}, \epsilon) \in h_\epsilon^{-1}(A_{\mathbf{z}_\epsilon})} p_{\mathbf{z}|u_1}(\mathbf{z}|u_1) p_\epsilon(\epsilon) \, d\mathbf{z} d\epsilon = \int_{(\mathbf{z}, \epsilon) \in h_c^{-1}(A_{\mathbf{z}_\epsilon})} p_{\mathbf{z}|u_2}(\mathbf{z}|u_2) p_\epsilon(\epsilon) \, d\mathbf{z} d\epsilon, \tag{22}$$

which contradicts Eq. (12). This contradiction implies that, for all $\epsilon \in \mathcal{E}$ and $r \in \mathbb{R}^+$, it follows that $h_\epsilon^{-1}(\mathcal{B}_r(\epsilon)) = \mathcal{Z} \times B_\epsilon^+$, where $\mathcal{B}_r(\epsilon) := \{\epsilon' \in \mathcal{E} : \|\epsilon' - \epsilon\|^2 < r\}$, $B_\epsilon^+ \neq \emptyset$, and $B_\epsilon^+ \subseteq \mathcal{E}$.

Suppose there exists $\hat{\epsilon} \in \mathcal{E}$ such that $h_\epsilon^{-1}(\hat{\epsilon})$ cannot be written as $\mathcal{Z} \times B_{\hat{\epsilon}}$ for any $B_{\hat{\epsilon}} \subseteq \mathcal{E}$. Since $h_\epsilon$ is continuous, there exists $\hat{r} \in \mathbb{R}^+$ such that for some $\tilde{\mathbf{z}} \in \mathcal{Z}$ and $\tilde{\epsilon} \in \mathcal{E}$ with $h_\epsilon(\tilde{\mathbf{z}}, \tilde{\epsilon}) = \hat{\epsilon}$, it holds that

$$h_\epsilon(\tilde{\mathbf{z}}, \tilde{\epsilon}) \notin \mathcal{B}_{\hat{r}}(\hat{\epsilon}). \tag{23}$$

This means

$$(\tilde{\mathbf{z}}, \tilde{\epsilon}) \notin h_\epsilon^{-1}\left(\mathcal{B}_{\hat{r}}(\hat{\epsilon})\right). \tag{24}$$

On the other hand, we have

$$h_\epsilon^{-1}\left(\mathcal{B}_{\hat{r}}(\hat{\epsilon})\right) = \mathcal{Z} \times B_{\hat{\epsilon}}^+, \tag{25}$$

where $B_{\hat{\epsilon}}^+ \subseteq \mathcal{E}$ and $B_{\hat{\epsilon}}^+ \neq \emptyset$. By the definition of $\tilde{\epsilon}$, it is clear that $\tilde{\epsilon} \in B_{\hat{\epsilon}}^+$. Therefore,

$$(\tilde{\mathbf{z}}, \tilde{\epsilon}) \in \mathcal{Z} \times B_{\hat{\epsilon}}^+ = h_\epsilon^{-1}\left(\mathcal{B}_{\hat{r}}(\hat{\epsilon})\right), \tag{26}$$

which contradicts our earlier conclusion that $(\tilde{\mathbf{z}}, \tilde{\epsilon}) \notin h_\epsilon^{-1}\left(\mathcal{B}_{\hat{r}}(\hat{\epsilon})\right)$. This implies that there does not exist $\hat{\epsilon} \in \mathcal{E}$ such that $h_\epsilon^{-1}(\hat{\epsilon})$ cannot be written as $\mathcal{Z} \times B_{\hat{\epsilon}}$ for any $B_{\hat{\epsilon}} \subseteq \mathcal{E}$. Therefore, $h_\epsilon^{-1}(\hat{\epsilon}) = \mathcal{Z} \times B_{\hat{\epsilon}}$ for some $B_{\hat{\epsilon}} \subseteq \mathcal{E}$, $B_{\hat{\epsilon}} \neq \emptyset$. This implies that $h_\epsilon(\mathbf{z}, \epsilon)$ does not depend on $\mathbf{z}$, so we can write $\hat{\epsilon} = h_\epsilon(\mathbf{z}, \epsilon) = \tilde{h}_\epsilon(\epsilon)$. Since $h$ is invertible (as both $f$ and $\hat{f}$ are invertible), we have $\epsilon = h_\epsilon^{-1}(\hat{\epsilon})$ Therefore, $\epsilon$ does not depend on $\hat{\mathbf{z}}$. $\square$

Now we are ready for the proof of Theorem 1.

**Theorem 1.** *Let the observed data be generated by a model defined in Eq.* (1). *Together with a $\ell_0$ regularization on $\hat{\mathcal{F}}_{\hat{z}}$ during estimation ($\|\hat{\mathcal{F}}_{\hat{z}}\|_0 \leq \|\mathcal{F}_z\|_0$), suppose the following assumptions:*

    *i. (Nondegeneracy) For all $i \in \{1, \ldots, n\}$, there exist points $\{(\mathbf{z}, \epsilon)^{(\ell)}\}_{\ell=1}^{|(\mathcal{F}_z)_{i,:}|}$ and a matrix $\mathrm{T} \in \mathbf{T}$ s.t. $\mathrm{span}\{D_{\mathbf{z}}f((\mathbf{z}, \epsilon)^{(\ell)})_{i,:}\}_{\ell=1}^{|(\mathcal{F}_z)_{i,:}|} = \mathbb{R}^n_{(\mathcal{F}_z)_{i,:}}$ and $\left[D_{\mathbf{z}}f((\mathbf{z}, \epsilon)^{(\ell)})\mathrm{T}\right]_{i,:} \in \mathbb{R}^n_{(\hat{\mathcal{F}}_{\hat{z}})_{i,:}}$.*

    *ii. (Domain Variability) For any set $A \subseteq \mathcal{Z} \times \mathcal{E}$ with non-zero probability measure that cannot be expressed as $B_\epsilon \times B_{\mathbf{z}}$ for any $B_\epsilon \subseteq \mathcal{E}$ and $B_{\mathbf{z}} = \mathcal{Z}$, there exist two domains $u_1$ and $u_2$ that are independent of $\epsilon$ s.t.*

$$\int_{(\mathbf{z}, \epsilon) \in A} \left[p(\mathbf{z}, \epsilon | u_1) - p(\mathbf{z}, \epsilon | u_2)\right] d\mathbf{z}\, d\epsilon \neq 0.$$

    *iii. (Structural Sparsity) For all $k \in \{1, \ldots, n\}$, there exists a set $\mathcal{C}_k$ s.t. $\bigcap_{i \in \mathcal{C}_k} (\mathcal{F}_z)_{i,:} = \{k\}$.*

*Then latent variables $\mathbf{z}$ are element-wise identifiable (Defn.* 1).

*Proof.* We aim to show that under the given assumptions, the latent variables $\mathbf{z}$ are identifiable up to element-wise invertible transformations and permutations. To this end, we consider the transformation $h : (\mathbf{z}, \epsilon) \to (\hat{\mathbf{z}}, \hat{\epsilon})$, which maps the true latent variables and noise to their estimated counterparts.

First, we apply the chain rule to the composition $\hat{f} \circ h = f$. The derivative of $\hat{f}$ with respect to $(\hat{\mathbf{z}}, \hat{\epsilon})$ can be expressed as:

$$D_{(\hat{\mathbf{z}}, \hat{\epsilon})}\hat{f} = D_{(\mathbf{z}, \epsilon)}f \cdot D_{(\hat{\mathbf{z}}, \hat{\epsilon})}h^{-1}. \tag{27}$$

The Jacobian matrix $D_{(\hat{\mathbf{z}}, \hat{\epsilon})}h^{-1}$ can be partitioned into blocks:

$$D_{(\hat{\mathbf{z}}, \hat{\epsilon})}h^{-1} = \left[\begin{array}{c|c} \frac{\partial \mathbf{z}}{\partial \hat{\mathbf{z}}} & \frac{\partial \mathbf{z}}{\partial \hat{\epsilon}} \\ \hline \frac{\partial \epsilon}{\partial \hat{\mathbf{z}}} & \frac{\partial \epsilon}{\partial \hat{\epsilon}} \end{array}\right]. \tag{28}$$

According to steps 1, 2, and 3 in the proof of Theorem 4.2 in Kong et al. (2022) (Lemma 1), the bottom-left block $\frac{\partial \epsilon}{\partial \hat{\mathbf{z}}}$ is zero. Thus, the Jacobian simplifies to:

$$D_{(\hat{\mathbf{z}}, \hat{\epsilon})}h^{-1} = \left[\begin{array}{c|c} \frac{\partial \mathbf{z}}{\partial \hat{\mathbf{z}}} & \frac{\partial \mathbf{z}}{\partial \hat{\epsilon}} \\ \hline \mathbf{0} & \frac{\partial \epsilon}{\partial \hat{\epsilon}} \end{array}\right]. \tag{29}$$

Since $h$ is invertible, the determinant of $D_{(\hat{\mathbf{z}}, \hat{\boldsymbol{\epsilon}})} h^{-1}$ is non-zero:

$$\det \left( D_{(\hat{\mathbf{z}}, \hat{\boldsymbol{\epsilon}})} h^{-1} \right) = \det \left( \frac{\partial \mathbf{z}}{\partial \hat{\mathbf{z}}} \right) \cdot \det \left( \frac{\partial \boldsymbol{\epsilon}}{\partial \hat{\boldsymbol{\epsilon}}} \right) \neq 0. \tag{30}$$

This implies that both $\dfrac{\partial \mathbf{z}}{\partial \hat{\mathbf{z}}}$ and $\dfrac{\partial \boldsymbol{\epsilon}}{\partial \hat{\boldsymbol{\epsilon}}}$ are invertible matrices:

$$\det \left( \frac{\partial \mathbf{z}}{\partial \hat{\mathbf{z}}} \right) \neq 0, \tag{31}$$

$$\det \left( \frac{\partial \boldsymbol{\epsilon}}{\partial \hat{\boldsymbol{\epsilon}}} \right) \neq 0. \tag{32}$$

Define the map between $\hat{\boldsymbol{\epsilon}}$ and $\boldsymbol{\epsilon}$ as $h_\epsilon : \hat{\boldsymbol{\epsilon}} \to \boldsymbol{\epsilon}$. Since $\det \left( \dfrac{\partial \boldsymbol{\epsilon}}{\partial \hat{\boldsymbol{\epsilon}}} \right) \neq 0$ and $\dfrac{\partial \boldsymbol{\epsilon}}{\partial \hat{\mathbf{z}}} = 0$, it follows that $\boldsymbol{\epsilon}$ depends solely on $\hat{\boldsymbol{\epsilon}}$ and not on $\hat{\mathbf{z}}$. Therefore, there exists an invertible function $h_\epsilon$ such that:

$$\boldsymbol{\epsilon} = h_\epsilon(\hat{\boldsymbol{\epsilon}}). \tag{33}$$

Since $\mathbf{z}$ is independent of $\boldsymbol{\epsilon}$ and $\boldsymbol{\epsilon} = h_\epsilon(\hat{\boldsymbol{\epsilon}})$, it follows that $\mathbf{z}$ is also independent of $\hat{\boldsymbol{\epsilon}}$. Thus

$$\frac{\partial \mathbf{z}}{\partial \hat{\boldsymbol{\epsilon}}} = \mathbf{0}. \tag{34}$$

Thus, the Jacobian further simplifies to:

$$D_{(\hat{\mathbf{z}}, \hat{\boldsymbol{\epsilon}})} h^{-1} = \left[ \begin{array}{c|c} \frac{\partial \mathbf{z}}{\partial \hat{\mathbf{z}}} & \mathbf{0} \\ \hline \mathbf{0} & \frac{\partial \boldsymbol{\epsilon}}{\partial \hat{\boldsymbol{\epsilon}}} \end{array} \right]. \tag{35}$$

Substituting Eq. (35) into the chain rule expression, we focus on the derivatives with respect to $\hat{\mathbf{z}}$:

$$D_{(\hat{\mathbf{z}}, \hat{\boldsymbol{\epsilon}})} \hat{f}_{:,:n} = D_{(\mathbf{z}, \boldsymbol{\epsilon})} f D_{(\hat{\mathbf{z}}, \hat{\boldsymbol{\epsilon}})} h^{-1}_{:,:n} \tag{36}$$

$$= D_{(\mathbf{z}, \boldsymbol{\epsilon})} f_{:,:n} \frac{\partial \mathbf{z}}{\partial \hat{\mathbf{z}}}. \tag{37}$$

Let us define a matrix T as follows:

$$D_{\hat{\mathbf{z}}} \hat{f} = D_{\mathbf{z}} f \mathrm{T}. \tag{38}$$

According to Assumption i, for each $i \in \{1, \ldots, n\}$, there exist points $\{(\mathbf{z}, \boldsymbol{\epsilon})^{(\ell)}\}_{\ell=1}^{|(\mathcal{F}_z)_{i,:}|}$ and a matrix $\mathrm{T} \in \boldsymbol{T}$ such that the set $\{D_{\mathbf{z}} f((\mathbf{z}, \boldsymbol{\epsilon})^{(\ell)})_{i,:}\}_{\ell=1}^{|(\mathcal{F}_z)_{i,:}|}$ spans the subspace $\mathbb{R}^n_{(\mathcal{F}_z)_{i,:}}$. This means that any vector in $\mathbb{R}^n_{(\mathcal{F}_z)_{i,:}}$ can be expressed as a linear combination of these derivative vectors.

Let us consider the standard basis vector $e_{j_0} \in \mathbb{R}^n_{(\mathcal{F}_z)_{i,:}}$ for some $j_0 \in (\mathcal{F}_z)_{i,:}$. Then, there exist coefficients $\beta_\ell$ such that:

$$e_{j_0} = \sum_{\ell=1}^{|(\mathcal{F}_z)_{i,:}|} \beta_\ell \cdot D_{\mathbf{z}} f((\mathbf{z}, \boldsymbol{\epsilon})^{(\ell)})_{i,:}. \tag{39}$$

Multiplying both sides of Eq. (39) by T, we obtain:

$$e_{j_0} \mathrm{T} = \sum_{\ell=1}^{|(\mathcal{F}_z)_{i,:}|} \beta_\ell \cdot D_{\mathbf{z}} f((\mathbf{z}, \boldsymbol{\epsilon})^{(\ell)})_{i,:} \mathrm{T}. \tag{40}$$

By Assumption i, the transformed derivatives $D_{\mathbf{z}} f((\mathbf{z}, \boldsymbol{\epsilon})^{(\ell)}) \mathrm{T}$ have their support within $(\hat{\mathcal{F}}_{\hat{z}})_{i,:}$. Consequently, the vector $e_{j_0} \mathrm{T}$ lies in $\mathbb{R}^n_{(\hat{\mathcal{F}}_{\hat{z}})_{i,:}}$. Therefore, for any $j \in (\mathcal{F}_z)_{i,:}$, it holds that:

$$\mathrm{T}_{j,:} \in \mathbb{R}^n_{(\hat{\mathcal{F}}_{\hat{z}})_{i,:}}. \tag{41}$$

Eq. (41) leads to the following inclusion of supports:

$$\forall (i,j) \in \mathcal{F}_z, \quad \{i\} \times \mathcal{T}_{j,:} \subseteq \hat{\mathcal{F}}_{\hat{z}}. \tag{42}$$

Here, $\mathcal{T}_{j,:}$ denotes the set of indices corresponding to non-zero entries in the $j$-th row of T.

Because both $D_{\hat{z}}\hat{f}$ and $D_z f$ are of full-column rank, T is invertible. Thus, its determinant is non-zero. Expanding the determinant, we have:

$$\det(T) = \sum_{\sigma \in \mathcal{S}_n} \mathrm{sgn}(\sigma) \prod_{i=1}^{n} T_{i,\sigma(i)} \neq 0, \tag{43}$$

where $\mathcal{S}_n$ is the set of all permutations of $\{1, 2, \ldots, n\}$, and $\mathrm{sgn}(\sigma)$ is the sign of the permutation $\sigma$.

The non-zero determinant implies that there exists at least one permutation $\sigma$ such that:

$$\forall i \in \{1, 2, \ldots, n\}, \quad T_{i,\sigma(i)} \neq 0. \tag{44}$$

From Eq. (44), for each $j \in \{1, 2, \ldots, n\}$, we have:

$$\sigma(j) \in \mathcal{T}_{j,:}. \tag{45}$$

Combining this with the support inclusion from Eq. (42), we deduce:

$$\forall (i,j) \in \mathcal{F}_z, \quad (i, \sigma(j)) \in \hat{\mathcal{F}}_{\hat{z}}. \tag{46}$$

Define the set:

$$\sigma(\mathcal{F}_z) = \{(i, \sigma(j)) \mid (i,j) \in \mathcal{F}_z\}. \tag{47}$$

Eq. (46) implies that:

$$\sigma(\mathcal{F}_z) \subseteq \hat{\mathcal{F}}_{\hat{z}}. \tag{48}$$

Since $\hat{\mathcal{F}}_{\hat{z}}$ is estimated under a sparsity constraint, we have:

$$|\hat{\mathcal{F}}_{\hat{z}}| \leq |\mathcal{F}_z|. \tag{49}$$

However, because $\sigma$ is a permutation (hence bijective), it holds that:

$$|\sigma(\mathcal{F}_z)| = |\mathcal{F}_z|. \tag{50}$$

Combining Eqs. (48), (49), and (50), we conclude:

$$|\hat{\mathcal{F}}_{\hat{z}}| = |\mathcal{F}_z| = |\sigma(\mathcal{F}_z)|, \tag{51}$$

which implies that:

$$\hat{\mathcal{F}}_{\hat{z}} = \sigma(\mathcal{F}_z). \tag{52}$$

Assume, for the sake of contradiction, that T is not simply a product of a permutation matrix and a diagonal (invertible scaling) matrix. Then there exist distinct indices $j_1 \neq j_2$ such that:

$$\mathcal{T}_{j_1,:} \cap \mathcal{T}_{j_2,:} \neq \emptyset. \tag{53}$$

Let $j_3$ be an index such that:

$$\sigma(j_3) \in \mathcal{T}_{j_1,:} \cap \mathcal{T}_{j_2,:}. \tag{54}$$

Without loss of generality, assume that $j_3 \neq j_1$.

From Assumption iii (Structural Sparsity), there exists a set $\mathcal{C}_{j_1}$ such that:

$$\bigcap_{i \in \mathcal{C}_{j_1}} (\mathcal{F}_z)_{i,:} = \{j_1\}. \tag{55}$$

Since $j_3 \notin \{j_1\}$, there exists $i_3 \in \mathcal{C}_{j_1}$ such that:

$$j_3 \notin (\mathcal{F}_z)_{i_3,:}. \tag{56}$$

Since $j_1 \in (\mathcal{F}_z)_{i_3,:}$, we have

$$(i_3, j_1) \in \mathcal{F}_z. \tag{57}$$

Thus, according to Eq. (42), we have

$$\{i_3\} \times \mathcal{T}_{j_1,:} \subseteq \hat{\mathcal{F}}_{\hat{z}} \tag{58}$$

Because of Eq. (54), this implies:

$$(i_3, \sigma(j_3)) \in \hat{\mathcal{F}}_{\hat{z}}. \tag{59}$$

Using Eq. (52), it follows that:

$$(i_3, j_3) \in \mathcal{F}_z. \tag{60}$$

This contradicts Eq. (56). Therefore, our assumption must be false, and T must indeed be a product of a permutation matrix and a diagonal matrix.

Thus, $h^{-1}$ in Eq. (38) is a composition of a permutation and an element-wise invertible transformation. Therefore, under the given assumptions, the latent variables $\mathbf{z}$ are identifiable up to element-wise invertible transformations and permutations. $\qquad\square$

## B.2 PROOF OF THEOREM 2

**Theorem 2.** *Let the observed data be generated by a model defined in Eq. (2). Together with a $\ell_0$ regularization on $\hat{\mathcal{F}}_{\hat{z}}$ during estimation ($\|\hat{\mathcal{F}}_{\hat{z}}\|_0 \leq \|\mathcal{F}_z\|_0$), suppose the following assumptions:*

*i. (Nondegeneracy) For all $i \in \{1, \ldots, n\}$, there exist points $\{\mathbf{z}^{(\ell)}\}_{\ell=1}^{|(\mathcal{F}_z)_{i,:}|}$ and a matrix $\mathrm{T} \in \boldsymbol{T}$ s.t. $\mathrm{span}\{D_{\mathbf{z}}f(\mathbf{z}^{(\ell)})_{i,:}\}_{\ell=1}^{|(\mathcal{F}_z)_{i,:}|} = \mathbb{R}^n_{(\mathcal{F}_z)_{i,:}}$ and $[D_{\mathbf{z}}f(\mathbf{z}^{(\ell)})\mathrm{T}]_{i,:} \in \mathbb{R}^n_{(\hat{\mathcal{F}}_{\hat{z}})_{i,:}}$*

*ii. (Structural Sparsity) For all $k \in \{1, \ldots, n\}$, there exists a set $\mathcal{C}_k$ s.t. $\bigcap_{i \in \mathcal{C}_k} (\mathcal{F}_z)_{i,:} = \{k\}$.*

*Then latent variables $\mathbf{z}$ are element-wise identifiable (Defn. 1).*

*Proof.* Under Assumption i, for each $i \in \{1, \ldots, n\}$, the set of vectors $\{D_{\mathbf{z}}f(\mathbf{z}^{(\ell)})_{i,:}\}_{\ell=1}^{|(\mathcal{F}_z)_{i,:}|}$ spans the subspace $\mathbb{R}^n_{(\mathcal{F}_z)_{i,:}}$. This implies that any vector in this subspace can be expressed as a linear combination of these derivative vectors.

Consider a standard basis vector $e_{j_0} \in \mathbb{R}^n_{(\mathcal{F}_z)_{i,:}}$ for some $j_0 \in (\mathcal{F}_z)_{i,:}$. There exist coefficients $\{\beta_\ell\}$ such that:

$$e_{j_0} = \sum_{\ell=1}^{|(\mathcal{F}_z)_{i,:}|} \beta_\ell \, D_{\mathbf{z}}f(\mathbf{z}^{(\ell)})_{i,:}. \tag{61}$$

Let T be the transformation matrix defined by the relationship:

$$D_{\hat{\mathbf{z}}}\hat{f} = D_{\mathbf{z}}f\mathrm{T}. \tag{62}$$

Multiplying both sides by T, we have:

$$e_{j_0}\mathrm{T} = \sum_{\ell=1}^{|(\mathcal{F}_z)_{i,:}|} \beta_\ell \, D_{\mathbf{z}}f(\mathbf{z}^{(\ell)})_{i,:}\mathrm{T}. \tag{63}$$

Since, by assumption, $D_{\mathbf{z}}f(\mathbf{z}^{(\ell)})\mathrm{T}$ has support in $\hat{\mathcal{F}}_{\hat{z}}$, it follows that the vector $e_{j_0}\mathrm{T}$ lies in $\mathbb{R}^n_{(\hat{\mathcal{F}}_{\hat{z}})_{i,:}}$. Therefore, for any $j \in (\mathcal{F}_z)_{i,:}$, we have:

$$\mathrm{T}_{j,:} \in \mathbb{R}^n_{(\hat{\mathcal{F}}_{\hat{z}})_{i,:}}. \tag{64}$$

This establishes the inclusion:

$$\forall (i,j) \in \mathcal{F}_z, \quad \{i\} \times \mathcal{T}_{j,:} \subseteq \hat{\mathcal{F}}_{\hat{z}}. \tag{65}$$

Since both $D_{\hat{\mathbf{z}}}\hat{f}$ and $D_{\mathbf{z}}f$ are invertible, T must also be invertible, implying that its determinant is non-zero:

$$\det(\mathrm{T}) = \sum_{\sigma \in \mathcal{S}_n} \mathrm{sgn}(\sigma) \prod_{i=1}^n \mathrm{T}_{i,\sigma(i)} \neq 0, \tag{66}$$

where $\mathcal{S}_n$ is the set of all permutations of $\{1, \ldots, n\}$, and $\text{sgn}(\sigma)$ denotes the sign of the permutation $\sigma$.

The non-zero determinant ensures that there exists at least one permutation $\sigma \in \mathcal{S}_n$ such that:

$$\forall i \in \{1, \ldots, n\}, \quad \mathrm{T}_{i,\sigma(i)} \neq 0. \tag{67}$$

Therefore, for each $j \in \{1, \ldots, n\}$, we have:

$$\sigma(j) \in \mathcal{T}_{j,:}. \tag{68}$$

Combining the support inclusion from Eq. (65) with Eq. (68), we deduce:

$$\forall (i,j) \in \mathcal{F}_z, \quad (i, \sigma(j)) \in \hat{\mathcal{F}}_{\hat{z}}. \tag{69}$$

Define the set:

$$\sigma(\mathcal{F}_z) = \{(i, \sigma(j)) \mid (i,j) \in \mathcal{F}_z\}. \tag{70}$$

Thus, we have:

$$\sigma(\mathcal{F}_z) \subseteq \hat{\mathcal{F}}_{\hat{z}}. \tag{71}$$

Since the estimated Jacobian $\hat{\mathcal{F}}_{\hat{z}}$ is obtained under a sparsity constraint, it satisfies:

$$|\hat{\mathcal{F}}_{\hat{z}}| \leq |\mathcal{F}_z|. \tag{72}$$

However, because $\sigma$ is a permutation (hence bijective), it holds that:

$$|\sigma(\mathcal{F}_z)| = |\mathcal{F}_z|. \tag{73}$$

Combining these results, we find:

$$|\hat{\mathcal{F}}_{\hat{z}}| = |\mathcal{F}_z| = |\sigma(\mathcal{F}_z)|, \tag{74}$$

which, together with Eq. (71), implies that:

$$\hat{\mathcal{F}}_{\hat{z}} = \sigma(\mathcal{F}_z). \tag{75}$$

Assume, for the sake of contradiction, that $\mathrm{T}$ is not a composition of a permutation matrix and a diagonal (invertible scaling) matrix. Then there exist distinct indices $j_1 \neq j_2$ such that:

$$\mathcal{T}_{j_1,:} \cap \mathcal{T}_{j_2,:} \neq \emptyset. \tag{76}$$

Let $j_3$ be an index such that:

$$\sigma(j_3) \in \mathcal{T}_{j_1,:} \cap \mathcal{T}_{j_2,:}. \tag{77}$$

Without loss of generality, suppose $j_3 \neq j_1$.

From Assumption ii (Structural Sparsity), there exists a set $\mathcal{C}_{j_1}$ such that:

$$\bigcap_{i \in \mathcal{C}_{j_1}} (\mathcal{F}_z)_{i,:} = \{j_1\}. \tag{78}$$

Since $j_3 \notin \{j_1\}$, there exists $i_3 \in \mathcal{C}_{j_1}$ such that:

$$j_3 \notin (\mathcal{F}_z)_{i_3,:}. \tag{79}$$

Since $j_1 \in (\mathcal{F}_z)_{i_3,:}$, we have

$$(i_3, j_1) \in \mathcal{F}_z. \tag{80}$$

Thus, according to Eq. (65), we have

$$\{i_3\} \times \mathcal{T}_{j_1,:} \subseteq \hat{\mathcal{F}}_{\hat{z}} \tag{81}$$

However, from Eq. (77), we have:

$$(i_3, \sigma(j_3)) \in \hat{\mathcal{F}}_{\hat{z}}. \tag{82}$$

Using Eq. (75), it follows that:

$$(i_3, j_3) \in \mathcal{F}_z. \tag{83}$$

This contradicts Eq. (79). Therefore, our assumption must be false, and T must indeed be a composition of a permutation matrix and a diagonal matrix.

Since the noise $\boldsymbol{\epsilon}$ has positive density and thus a non-zero characteristic function, by Step I of the proof of Theorem 1 in (Khemakhem et al., 2020b), the noise-free distributions must be identical for the observational distributions to match. Define the composite function $h = \hat{f}^{-1} \circ f$. Applying the chain rule yields:

$$D_{\hat{\mathbf{z}}}\hat{f} = D_{\mathbf{z}}f \cdot \mathrm{T}. \tag{84}$$

Since T is a composition of a permutation matrix and a diagonal matrix, $h$ must also be of this form. Therefore, the latent variables $\mathbf{z}$ are identifiable up to permutations and element-wise invertible transformations. $\qquad\square$

### B.3 Proof of Corollary 1

**Corollary 1.** *Let the observed data be generated by a model defined in Eqs.* (3) *and* (4). *Together with a $\ell_0$ regularization on $\hat{\mathcal{F}}_{\hat{z}}$ during estimation ($\|\hat{\mathcal{F}}_{\hat{z}}\|_0 \leq \|\mathcal{F}_z\|_0$), suppose the following assumptions:*

  i. *(Nondegeneracy) For all $i \in \{1,\ldots,n\}$, there exist points $\{(\mathbf{z},\boldsymbol{\epsilon},\boldsymbol{\eta})^{(\ell)}\}_{\ell=1}^{|(\mathcal{F}_z)_{i,:}|}$ and a matrix $\mathrm{T} \in \boldsymbol{T}$ s.t. $\mathrm{span}\{D_{\mathbf{z}}f((\mathbf{z},\boldsymbol{\epsilon},\boldsymbol{\eta})^{(\ell)})_{i,:}\}_{\ell=1}^{|(\mathcal{F}_z)_{i,:}|} = \mathbb{R}^n_{(\mathcal{F}_z)_{i,:}}$ and $\left[D_{\mathbf{z}}f((\mathbf{z},\boldsymbol{\epsilon},\boldsymbol{\eta})^{(\ell)})\mathrm{T}\right]_{i,:} \in \mathbb{R}^n_{(\hat{\mathcal{F}}_{\hat{z}})_{i,:}}$.*

  ii. *(Domain Variability) For any set $A \subseteq \mathcal{Z} \times \mathcal{E}$ with non-zero probability measure that cannot be expressed as $B_{\boldsymbol{\epsilon}} \times B_{\mathbf{z}}$ for any $B_{\boldsymbol{\epsilon}} \subseteq \mathcal{E}$ and $B_{\mathbf{z}} = \mathcal{Z}$, there exist two domains $u_1$ and $u_2$ that are independent of $\boldsymbol{\epsilon}$ s.t.*

$$\int_{(\mathbf{z},\boldsymbol{\epsilon})\in A} [p(\mathbf{z},\boldsymbol{\epsilon}|u_1) - p(\mathbf{z},\boldsymbol{\epsilon}|u_2)] \, d\mathbf{z} \, d\boldsymbol{\epsilon} \neq 0.$$

  iii. *(Structural Sparsity) For all $k \in \{1,\ldots,n\}$, there exists a set $\mathcal{C}_k$ s.t. $\bigcap_{i \in \mathcal{C}_k} (\mathcal{F}_{1z})_{i,:} = \{k\}$.*

*Then latent variables $\mathbf{z}$ are element-wise identifiable (Defn. 1).*

*Proof.* Since the noise $\boldsymbol{\eta}$ has a positive density and thus a non-zero characteristic function, by Step I of the proof of Theorem 1 in Khemakhem et al. (2020b), the noise-free distributions must be identical for the observational distributions to match. Denote $h : (\mathbf{z}, \boldsymbol{\epsilon}) \to (\hat{\mathbf{z}}, \hat{\boldsymbol{\epsilon}})$, we have:

$$D_{(\hat{\mathbf{z}},\hat{\boldsymbol{\epsilon}})}\hat{f} = D_{(\mathbf{z},\boldsymbol{\epsilon})}f D_{(\hat{\mathbf{z}},\hat{\boldsymbol{\epsilon}})}h^{-1}. \tag{85}$$

Let us represent the Jacobian $D_{(\hat{\mathbf{z}},\hat{\boldsymbol{\epsilon}})}h^{-1}$ as follows:

$$D_{(\hat{\mathbf{z}},\hat{\boldsymbol{\epsilon}})}h^{-1} = \begin{bmatrix} \frac{\partial \mathbf{z}}{\partial \hat{\mathbf{z}}} & \frac{\partial \mathbf{z}}{\partial \hat{\boldsymbol{\epsilon}}} \\ \hline \frac{\partial \boldsymbol{\epsilon}}{\partial \hat{\mathbf{z}}} & \frac{\partial \boldsymbol{\epsilon}}{\partial \hat{\boldsymbol{\epsilon}}} \end{bmatrix}. \tag{86}$$

According to steps 1, 2, and 3 in the proof of Theorem 4.2 in Kong et al. (2022), the bottom-left block $\frac{\partial \boldsymbol{\epsilon}}{\partial \hat{\mathbf{z}}}$ is zero. Thus, the Jacobian simplifies to:

$$D_{(\hat{\mathbf{z}},\hat{\boldsymbol{\epsilon}})}h^{-1} = \begin{bmatrix} \frac{\partial \mathbf{z}}{\partial \hat{\mathbf{z}}} & \frac{\partial \mathbf{z}}{\partial \hat{\boldsymbol{\epsilon}}} \\ \hline \mathbf{0} & \frac{\partial \boldsymbol{\epsilon}}{\partial \hat{\boldsymbol{\epsilon}}} \end{bmatrix}. \tag{87}$$

Moreover, because $h$ is invertible, the determinant of $D_{(\hat{\mathbf{z}},\hat{\boldsymbol{\epsilon}})}h^{-1}$ is non-zero. Together with the structure of the Jacobian matrix, we have:

$$\det\left(D_{(\hat{\mathbf{z}},\hat{\boldsymbol{\epsilon}})}h^{-1}\right) = \det\left(\frac{\partial \mathbf{z}}{\partial \hat{\mathbf{z}}}\right) \cdot \det\left(\frac{\partial \boldsymbol{\epsilon}}{\partial \hat{\boldsymbol{\epsilon}}}\right) \neq 0. \tag{88}$$

Thus, there must be

$$\det\left(\frac{\partial \mathbf{z}}{\partial \hat{\mathbf{z}}}\right) \neq 0, \tag{89}$$

$$\det\left(\frac{\partial \boldsymbol{\epsilon}}{\partial \hat{\boldsymbol{\epsilon}}}\right) \neq 0. \tag{90}$$

Define the map between $\hat{\boldsymbol{\epsilon}}$ and $\boldsymbol{\epsilon}$ as $h_\epsilon : \hat{\boldsymbol{\epsilon}} \to \boldsymbol{\epsilon}$. Since $\det\left(\frac{\partial \boldsymbol{\epsilon}}{\partial \hat{\boldsymbol{\epsilon}}}\right) \neq 0$ and $\frac{\partial \boldsymbol{\epsilon}}{\partial \hat{\mathbf{z}}} = 0$, it follows that $\boldsymbol{\epsilon}$ depends solely on $\hat{\boldsymbol{\epsilon}}$ and not on $\hat{\mathbf{z}}$. Therefore, there exists an invertible function $h_\epsilon$ such that:

$$\boldsymbol{\epsilon} = h_\epsilon(\hat{\boldsymbol{\epsilon}}). \tag{91}$$

Since $\mathbf{z}$ is independent of $\boldsymbol{\epsilon}$ and $\boldsymbol{\epsilon} = h_\epsilon(\hat{\boldsymbol{\epsilon}})$, it follows that $\mathbf{z}$ is also independent of $\hat{\boldsymbol{\epsilon}}$. Thus

$$\frac{\partial \mathbf{z}}{\partial \hat{\boldsymbol{\epsilon}}} = \mathbf{0}. \tag{92}$$

Thus we have

$$D_{(\hat{\mathbf{z}},\hat{\boldsymbol{\epsilon}})}\hat{f}_{:,:n} = D_{(\mathbf{z},\epsilon)}f_{:,:n}D_{(\hat{\mathbf{z}},\hat{\boldsymbol{\epsilon}})}h^{-1}_{:n,:n}, \tag{93}$$

which is equivalent to

$$D_{\hat{\mathbf{z}}}\hat{f} = D_{\mathbf{z}}fD_{(\hat{\mathbf{z}},\hat{\boldsymbol{\epsilon}})}h^{-1}_{:n,:n}. \tag{94}$$

We need to prove that $D_{(\hat{\mathbf{z}},\hat{\boldsymbol{\epsilon}})}h^{-1}_{:n,:n}$ is a generalized permutation matrix.

Since both $D_{\hat{\mathbf{z}}}\hat{f}$ and $D_{\mathbf{z}}f$ are of full column rank, we have

$$D_{\hat{\mathbf{z}}}\hat{f} = D_{\mathbf{z}}fD_{(\hat{\mathbf{z}},\hat{\boldsymbol{\epsilon}})}\mathbf{T}, \tag{95}$$

where $\mathbf{T}$ has a non-zero determinant. Expanding the determinant, we have:

$$\det(\mathrm{T}) = \sum_{\sigma \in \mathcal{S}_n} \mathrm{sgn}(\sigma) \prod_{i=1}^{n} \mathrm{T}_{i,\sigma(i)} \neq 0, \tag{96}$$

where $\mathcal{S}_n$ is the set of all permutations of $\{1, 2, \ldots, n\}$, and $\mathrm{sgn}(\sigma)$ denotes the sign of the permutation $\sigma$.

The non-zero determinant ensures that there exists at least one permutation $\sigma \in \mathcal{S}_n$ such that:

$$\forall i \in \{1, 2, \ldots, n\}, \quad \mathrm{T}_{i,\sigma(i)} \neq 0. \tag{97}$$

Therefore, for each $j \in \{1, 2, \ldots, n\}$, we have:

$$\sigma(j) \in \mathcal{T}_{j,:}. \tag{98}$$

According to Assumption i, for each $i \in \{1, \ldots, n\}$, there exist points $\{(\mathbf{z}, \boldsymbol{\epsilon}, \boldsymbol{\eta})^{(\ell)}\}_{\ell=1}^{|(\mathcal{F}_z)_{i,:}|}$ such that the set $\{D_{\mathbf{z}}f((\mathbf{z}, \boldsymbol{\epsilon}, \boldsymbol{\eta})^{(\ell)})_{i,:}\}_{\ell=1}^{|(\mathcal{F}_z)_{i,:}|}$ spans the subspace $\mathbb{R}^n_{(\mathcal{F}_z)_{i,:}}$. This means that any vector in $\mathbb{R}^n_{(\mathcal{F}_z)_{i,:}}$ can be expressed as a linear combination of these derivative vectors.

Let us consider the standard basis vector $e_{j_0} \in \mathbb{R}^n_{(\mathcal{F}_z)_{i,:}}$ for some $j_0 \in (\mathcal{F}_z)_{i,:}$. Then, there exist coefficients $\{\beta_\ell\}$ such that:

$$e_{j_0} = \sum_{\ell=1}^{|(\mathcal{F}_z)_{i,:}|} \beta_\ell \cdot D_{\mathbf{z}}f((\mathbf{z}, \boldsymbol{\epsilon}, \boldsymbol{\eta})^{(\ell)})_{i,:}. \tag{99}$$

Multiplying both sides of Eq. (99) by T, we obtain:

$$e_{j_0} \cdot \mathrm{T} = \sum_{\ell=1}^{|(\mathcal{F}_z)_{i,:}|} \beta_\ell \cdot D_{\mathbf{z}}f((\mathbf{z}, \boldsymbol{\epsilon}, \boldsymbol{\eta})^{(\ell)})_{i,:}\mathrm{T}. \tag{100}$$

By Assumption i, the transformed derivatives $D_{\mathbf{z}} f((\mathbf{z}, \boldsymbol{\epsilon}, \boldsymbol{\eta})^{(\ell)}) \mathrm{T}$ have their support within $(\hat{\mathcal{F}}_{\hat{z}})_{i,:}$. Consequently, the vector $e_{j_0} \cdot \mathrm{T}$ lies in $\mathbb{R}^n_{(\hat{\mathcal{F}}_{\hat{z}})_{i,:}}$. Therefore, for any $j \in (\mathcal{F}_z)_{i,:}$, it holds that:

$$\mathrm{T}_{j,:} \in \mathbb{R}^n_{(\hat{\mathcal{F}}_{\hat{z}})_{i,:}}. \tag{101}$$

Eq. (101) leads to the following inclusion of supports:

$$\forall (i,j) \in \mathcal{F}_z, \quad \{i\} \times \mathcal{T}_{j,:} \subseteq \hat{\mathcal{F}}_{\hat{z}}. \tag{102}$$

Here, $\mathcal{T}_{j,:}$ denotes the set of indices corresponding to non-zero entries in the $j$-th row of matrix $\mathrm{T}$.

Combining the support inclusion from Eq. (102) with Eq. (98), we deduce:

$$\forall (i,j) \in \mathcal{F}_z, \quad (i, \sigma(j)) \in \hat{\mathcal{F}}_{\hat{z}}. \tag{103}$$

Define the set:

$$\sigma(\mathcal{F}_z) = \{(i, \sigma(j)) \mid (i,j) \in \mathcal{F}_z\}. \tag{104}$$

Thus, we have:

$$\sigma(\mathcal{F}_z) \subseteq \hat{\mathcal{F}}_{\hat{z}}. \tag{105}$$

Since $\hat{\mathcal{F}}_{\hat{z}}$ is estimated under a sparsity constraint, we have:

$$|\hat{\mathcal{F}}_{\hat{z}}| \leq |\mathcal{F}_z|. \tag{106}$$

However, because $\sigma$ is a permutation (hence bijective), it holds that:

$$|\sigma(\mathcal{F}_z)| = |\mathcal{F}_z|. \tag{107}$$

Combining these results, we conclude:

$$|\hat{\mathcal{F}}_{\hat{z}}| = |\mathcal{F}_z| = |\sigma(\mathcal{F}_z)|, \tag{108}$$

which implies that:

$$\hat{\mathcal{F}}_{\hat{z}} = \sigma(\mathcal{F}_z). \tag{109}$$

Assume, for the sake of contradiction, that $\mathrm{T}$ is not a composition of a permutation matrix and a diagonal (invertible scaling) matrix. Then there exist distinct indices $j_1 \neq j_2$ such that:

$$\mathcal{T}_{j_1,:} \cap \mathcal{T}_{j_2,:} \neq \emptyset. \tag{110}$$

Let $j_3$ be an index such that:

$$\sigma(j_3) \in \mathcal{T}_{j_1,:} \cap \mathcal{T}_{j_2,:}. \tag{111}$$

Without loss of generality, assume that $j_3 \neq j_1$.

From Assumption iii (Structural Sparsity), there exists a set $\mathcal{C}_{j_1}$ such that:

$$\bigcap_{i \in \mathcal{C}_{j_1}} (\mathcal{F}_z)_{i,:} = \{j_1\}. \tag{112}$$

Since $j_3 \notin \{j_1\}$, there exists $i_3 \in \mathcal{C}_{j_1}$ such that:

$$j_3 \notin (\mathcal{F}_z)_{i_3,:}. \tag{113}$$

Note that

$$j_1 \in (\mathcal{F}_z)_{i_3,:}, \tag{114}$$

which indicates $(i_3, j_1) \in \mathcal{F}_z$. Therefore, we have the following relation according to Eq. (102):

$$\{i_3\} \times \mathcal{T}_{j_1,:} \subseteq \hat{\mathcal{F}}_{\hat{z}} \tag{115}$$

From Eq. (111), we have

$$(i_3, \sigma(j_3)) \in \hat{\mathcal{F}}_{\hat{z}}. \tag{116}$$

Using Eq. (109), it follows that

$$(i_3, j_3) \in \mathcal{F}_z. \tag{117}$$

This contradicts Eq. (113). Therefore, our assumption must be false, and $\mathrm{T}$ must indeed be a composition of a permutation matrix and a diagonal matrix. This ensures that $D_{(\hat{\mathbf{z}}, \hat{\epsilon})} h^{-1}_{:n,:n}$ is a generalized permutation matrix.

Thus, under the given assumptions, the latent variables $\mathbf{z}$ are identifiable up to permutations and element-wise invertible transformations. $\qquad \square$

### B.4 PROOF OF THEOREM 3

**Theorem 3.** *Let the observed data be generated by a model defined in Eqs.* (5) *and* (6). *Suppose for each* $i \in \{1, \ldots, m\}$, *there exist* $\{(\boldsymbol{\xi}, \boldsymbol{\eta})^{(\ell)}\}_{\ell=1}^{|\mathcal{F}_{\xi_{i,:}}|}$ *and a matrix* $\mathrm{T}_\xi \in \boldsymbol{T}_\xi$ *s.t.* $\mathrm{span}\{D_{\boldsymbol{\xi}} f((\boldsymbol{\xi}, \boldsymbol{\eta})^{(\ell)})_{i,:}\}_{\ell=1}^{|(\mathcal{F}_\xi)_{i,:}|} = \mathbb{R}^m_{(\mathcal{F}_\xi)_{i,:}}$ *and* $\left[ D_{\boldsymbol{\xi}} f((\boldsymbol{\xi}, \boldsymbol{\eta})^{(\ell)}) \mathrm{T}_\xi \right]_{i,:} \in \mathbb{R}^m_{(\hat{\mathcal{F}}_{\hat{\xi}})_{i,:}}$. *Then* $\mathrm{G}_{\hat{f}^{-1}} = P\mathrm{G}_{f_1^{-1}}$ *for a permutation matrix* $P$ together with a $\ell_0$ regularization on $\hat{\mathcal{F}}_{\hat{z}}$ during estimation $(\|\hat{\mathcal{F}}_{\hat{z}}\|_0 \leq \|\mathcal{F}_z\|_0)$.

*Proof.* Since the noise $\boldsymbol{\eta}$ has positive density and thus a non-zero characteristic function. Thus, by Step I of the proof of Theorem 1 in (Khemakhem et al., 2020b), the noise-free distributions must be identical for the observational distributions to match. Denote $h : \boldsymbol{\xi} \to \hat{\boldsymbol{\xi}}$, we have:

$$D_{\hat{\boldsymbol{\xi}}} \hat{f} = D_{\boldsymbol{\xi}} f \cdot D_{\hat{\boldsymbol{\xi}}} h^{-1}, \tag{118}$$

where $D_{\boldsymbol{\xi}} f$ is the Jacobian of $f$ with respect to $\boldsymbol{\xi}$, $D_{\hat{\boldsymbol{\xi}}} \hat{f}$ is the Jacobian of $\hat{f}$ with respect to $\hat{\boldsymbol{\xi}}$, and $D_{\hat{\boldsymbol{\xi}}} h^{-1}$ is the Jacobian of $h^{-1}$ with respect to $\hat{\boldsymbol{\xi}}$.

Remember that we have the following notations:

$$\begin{aligned} \mathcal{F}_\xi &:= \mathrm{supp}(D_{\boldsymbol{\xi}} f), \\ \hat{\mathcal{F}}_{\hat{\xi}} &:= \mathrm{supp}(D_{\hat{\boldsymbol{\xi}}} \hat{f}). \end{aligned} \tag{119}$$

Furthermore, $\boldsymbol{T}_\xi$ refers to a set of matrices with the same support as $D_{\hat{\boldsymbol{\xi}}} h^{-1}$, and $\mathrm{T} \in \boldsymbol{T}_\xi$. Based on the assumption, we have:

$$\mathrm{span}\{D_{\boldsymbol{\xi}} f((\boldsymbol{\xi}, \boldsymbol{\eta})^{(\ell)})_{i,:}\}_{\ell=1}^{|\mathcal{F}_{\xi_{i,:}}|} = \mathbb{R}^{|\mathcal{F}_{\xi_{i,:}}|}. \tag{120}$$

Given that the set $\{D_{\boldsymbol{\xi}} f((\boldsymbol{\xi}, \boldsymbol{\eta})^{(\ell)})_{i,:}\}_{\ell=1}^{|\mathcal{F}_{\xi_{i,:}}|}$ forms a basis of $\mathbb{R}^{|\mathcal{F}_{\xi_{i,:}}|}$, we can express any vector in this space as a linear combination of these basis vectors. In particular, for any $j_0 \in \mathcal{F}_{\xi_{i,:}}$, the one-hot vector $e_{j_0} \in \mathbb{R}^{|\mathcal{F}_{\xi_{i,:}}|}$ can be written as

$$e_{j_0} = \sum_\ell \alpha_\ell D_{\boldsymbol{\xi}} f((\boldsymbol{\xi}, \boldsymbol{\eta})^{(\ell)})_{i,:}, \tag{121}$$

where $\alpha_\ell$ denotes the respective coefficient.

With this in mind, we can find the transformation of $e_{j_0}$ under $\mathrm{T}$ as

$$\mathrm{T}_{j_0,:} = e_{j_0} \mathrm{T} = \sum_\ell \alpha_\ell D_{\boldsymbol{\xi}} f((\boldsymbol{\xi}, \boldsymbol{\eta})^{(\ell)})_{i,:} \mathrm{T}. \tag{122}$$

According to the assumption, each term in the above summation belongs to the space $\mathbb{R}^{|(\hat{\mathcal{F}}_{\hat{\xi}})_{i,:}|}$. Therefore, $\mathrm{T}_{j_0,:}$ itself resides in $\mathbb{R}^{|(\hat{\mathcal{F}}_{\hat{\xi}})_{i,:}|}$, i.e., $\mathrm{T}_{j_0,:} \in \mathbb{R}^{|(\hat{\mathcal{F}}_{\hat{\xi}})_{i,:}|}$. Thus,

$$\forall j \in \mathcal{F}_{\xi_{i,:}}, \quad \mathrm{T}_{j,:} \in \mathbb{R}^{|(\hat{\mathcal{F}}_{\hat{\xi}})_{i,:}|}. \tag{123}$$

Then the connections between these supports can be established:

$$\forall (i, j) \in \mathcal{F}_\xi, \quad \{i\} \times \mathrm{supp}(\mathrm{T}_{j,:}) \subseteq \hat{\mathcal{F}}_{\hat{\xi}}. \tag{124}$$

Since $D_{\boldsymbol{\xi}} f$ and $D_{\hat{\boldsymbol{\xi}}} \hat{f}$ have full rank $n$, $\mathrm{T}$ must have a non-zero determinant. Otherwise, it would follow that the rank of $\mathrm{T}$ is less than $n$, which would imply a contradiction that $D_{\hat{\boldsymbol{\xi}}} \hat{f} = D_{\boldsymbol{\xi}} f \cdot \mathrm{T}$ has rank less than $n$. Representing the determinant of the matrix $\mathrm{T}$ as its Leibniz formula yields

$$\det(\mathrm{T}) = \sum_{\sigma \in S_n} \mathrm{sgn}(\sigma) \prod_{i=1}^n \mathrm{T}_{i,\sigma(i)} \neq 0, \tag{125}$$

where $S_n$ is the set of all permutations of $\{1, \ldots, n\}$. Thus, there is at least one permutation $\sigma \in S_n$ such that:

$$\forall i \in \{1, \ldots, n\}, \quad \mathrm{T}_{i,\sigma(i)} \neq 0. \tag{126}$$

Then we can conclude that this $\sigma$ is in the support of T. Therefore, it follows that

$$\forall j \in \{1, \ldots, n\}, \quad \sigma(j) \in \mathrm{supp}(\mathrm{T}_{j,:}). \tag{127}$$

Together with Eq. (124), we have

$$\forall (i, j) \in \mathcal{F}_\xi, \quad (i, \sigma(j)) \in \hat{\mathcal{F}}_{\hat{\xi}}. \tag{128}$$

Denote

$$\sigma(\mathcal{F}_\xi) = \{(i, \sigma(j)) \mid (i, j) \in \mathcal{F}_\xi\}. \tag{129}$$

Then we have

$$\sigma(\mathcal{F}_\xi) \subseteq \hat{\mathcal{F}}_{\hat{\xi}}. \tag{130}$$

Because of the sparsity regularization on the estimated Jacobian, we further have

$$|\hat{\mathcal{F}}_{\hat{\xi}}| \leq |\mathcal{F}_\xi| = |\sigma(\mathcal{F}_\xi)|. \tag{131}$$

Combining this with Eq. (130), we derive

$$\sigma(\mathcal{F}_\xi) = \hat{\mathcal{F}}_{\hat{\xi}}. \tag{132}$$

This implies that

$$D_{\hat{\xi}}\hat{f} = D_1 D_\xi f D_2 P, \tag{133}$$

where $D_1$ and $D_2$ are diagonal matrices, and $P$ is a permutation matrix corresponding to $\sigma$.

According to the chain rule, we have

$$\begin{aligned} D_\xi f &= D_{\mathbf{z}} f_2 \cdot D_\xi f_1, \\ D_{\hat{\xi}}\hat{f} &= D_{\hat{\mathbf{z}}}\hat{f}_2 \cdot D_{\hat{\xi}}\hat{f}_1. \end{aligned} \tag{134}$$

Since both $D_{\mathbf{z}} f_2$ and $D_{\hat{\mathbf{z}}}\hat{f}_2$ are diagonal matrices (because $f_2$ and $\hat{f}_2$ are element-wise functions), Eq. (134) further yields

$$\begin{aligned} \mathrm{supp}(D_\xi f) &= \mathrm{supp}(D_\xi f_1), \\ \mathrm{supp}(D_{\hat{\xi}}\hat{f}) &= \mathrm{supp}(D_{\hat{\xi}}\hat{f}_1). \end{aligned} \tag{135}$$

Because

$$\mathrm{supp}(D_\xi f) = \mathrm{supp}(D_1 D_\xi f D_2), \tag{136}$$

we have the following equation together with the previous result:

$$\mathrm{supp}(D_{\hat{\xi}}\hat{f}) = \mathrm{supp}(D_\xi f P). \tag{137}$$

This implies

$$\mathrm{supp}(D_{\hat{\xi}}\hat{f}_1) = \mathrm{supp}(D_\xi f_1 P). \tag{138}$$

Now, let us consider the inverses of these matrices:

$$\begin{aligned} (D_\xi f_1 P)^{-1} &= \frac{1}{\det(D_\xi f_1 P)} \cdot \mathrm{adj}(D_\xi f_1 P), \\ (D_{\hat{\xi}}\hat{f}_1)^{-1} &= \frac{1}{\det(D_{\hat{\xi}}\hat{f}_1)} \cdot \mathrm{adj}(D_{\hat{\xi}}\hat{f}_1). \end{aligned} \tag{139}$$

Since $D_{\hat{\xi}}\hat{f}_1$ and $D_\xi f_1 P$ have the same support, the submatrices $(D_{\hat{\xi}}\hat{f}_1)_{[n]\backslash i, [n]\backslash j}$ and $(D_\xi f_1 P)_{[n]\backslash i, [n]\backslash j}$ also have the same support. That is,

$$\mathrm{supp}((D_{\hat{\xi}}\hat{f}_1)_{[n]\backslash i, [n]\backslash j}) = \mathrm{supp}((D_\xi f_1 P)_{[n]\backslash i, [n]\backslash j}). \tag{140}$$

This means for any position $(k, l)$ in $(D_{\hat{\xi}}\hat{f}_1)_{[n]\backslash i, [n]\backslash j}$ and $(D_\xi f_1 P)_{[n]\backslash i, [n]\backslash j}$,

$$[(D_{\hat{\xi}}\hat{f}_1)_{[n]\backslash i, [n]\backslash j}]_{k,l} \neq 0 \iff [(D_\xi f_1 P)_{[n]\backslash i, [n]\backslash j}]_{k,l} \neq 0. \tag{141}$$

The determinant of an $(n-1) \times (n-1)$ matrix is a sum of products of its elements, each product corresponding to a permutation of the row and column indices, with a sign given by the parity of the permutation. Specifically,

$$\det((D_{\hat{\boldsymbol{\xi}}}\hat{f}_1)_{[n]\backslash i,[n]\backslash j}) = \sum_{\sigma \in S_{n-1}} \text{sgn}(\sigma) \prod_{k=1}^{n-1} [(D_{\hat{\boldsymbol{\xi}}}\hat{f}_1)_{[n]\backslash i,[n]\backslash j}]_{k,\sigma(k)}, \tag{142}$$

where $S_{n-1}$ is the set of all permutations of $\{1, \ldots, n-1\}$. Similarly,

$$\det((D_{\boldsymbol{\xi}}f_1 P)_{[n]\backslash i,[n]\backslash j}) = \sum_{\sigma \in S_{n-1}} \text{sgn}(\sigma) \prod_{k=1}^{n-1} [(D_{\boldsymbol{\xi}}f_1 P)_{[n]\backslash i,[n]\backslash j}]_{k,\sigma(k)}. \tag{143}$$

Clearly, the determinant is non-zero if there exists at least one term in the sum that is non-zero. For such a term to be non-zero, all elements in the corresponding product must be non-zero.

Given Eq. (140), any product of elements in $\det((D_{\hat{\boldsymbol{\xi}}}\hat{f}_1)_{[n]\backslash i,[n]\backslash j})$ that is non-zero will correspond to a product of elements in $\det((D_{\boldsymbol{\xi}}f_1 P)_{[n]\backslash i,[n]\backslash j})$ that is also non-zero, and vice versa. This is because the positions of non-zero elements in the two submatrices are identical.

Therefore,

$$\det((D_{\hat{\boldsymbol{\xi}}}\hat{f}_1)_{[n]\backslash i,[n]\backslash j}) \neq 0 \iff \det((D_{\boldsymbol{\xi}}f_1 P)_{[n]\backslash i,[n]\backslash j}) \neq 0. \tag{144}$$

This implies that for each position $(i, j)$, the cofactor $C_{ij}$ will be non-zero for $D_{\hat{\boldsymbol{\xi}}}\hat{f}_1$ if and only if it is non-zero for $D_{\boldsymbol{\xi}}f_1 P$:

$$C_{ij}(D_{\hat{\boldsymbol{\xi}}}\hat{f}_1) \neq 0 \iff C_{ij}(D_{\boldsymbol{\xi}}f_1 P) \neq 0. \tag{145}$$

According to the definition of the adjugate matrix, we have

$$\text{adj}(D_{\hat{\boldsymbol{\xi}}}\hat{f}_1)_{ij} = C_{ji}(D_{\hat{\boldsymbol{\xi}}}\hat{f}_1), \tag{146}$$

and similarly,

$$\text{adj}(D_{\boldsymbol{\xi}}f_1 P)_{ij} = C_{ji}(D_{\boldsymbol{\xi}}f_1 P). \tag{147}$$

Since $C_{ij}(D_{\hat{\boldsymbol{\xi}}}\hat{f}_1) \neq 0$ if and only if $C_{ij}(D_{\boldsymbol{\xi}}f_1 P) \neq 0$, it follows that:

$$\text{adj}(D_{\hat{\boldsymbol{\xi}}}\hat{f}_1)_{ij} \neq 0 \iff \text{adj}(D_{\boldsymbol{\xi}}f_1 P)_{ij} \neq 0. \tag{148}$$

Thus, the supports of the adjugate matrices are the same:

$$\text{supp}(\text{adj}(D_{\hat{\boldsymbol{\xi}}}\hat{f}_1)) = \text{supp}(\text{adj}(D_{\boldsymbol{\xi}}f_1 P)). \tag{149}$$

Therefore, their inverses also have supports related by the permutation $P$ according to Eq. (139):

$$\text{supp}((D_{\boldsymbol{\xi}}f_1 P)^{-1}) = \text{supp}((D_{\hat{\boldsymbol{\xi}}}\hat{f}_1)^{-1}). \tag{150}$$

Which implies

$$\text{supp}((D_{\hat{\boldsymbol{\xi}}}\hat{f}_1)^{-1}) = P^{-1}\,\text{supp}((D_{\boldsymbol{\xi}}f_1)^{-1}). \tag{151}$$

Since $\hat{f}_2$ is an element-wise transformation, we have

$$\mathrm{G}_{\hat{f}^{-1}} = P\mathrm{G}_{f_1^{-1}}. \tag{152}$$

$\square$

### B.5 PROOF OF PROPOSITION 2

**Proposition 2.** *Suppose the assumptions in Theorem 3 and Proposition 1 hold, then $\mathcal{A}$ in Eq. 7 is identifiable.*

*Proof.* Under the assumptions of Theorem 3, we have established that

$$G_{f_1^{-1}} = P G_{\hat{f}^{-1}}. \tag{153}$$

From Proposition 1, the matrix $G_{f_1^{-1}}$ is structurally equivalent to $\mathbf{I}_n - \mathcal{A}$, where $\mathcal{A}$ is the adjacency matrix of the directed acyclic graph (DAG) defined by the structural causal model in Eq. (7). That is,

$$G_{f_1^{-1}} \sim \mathbf{I}_n - \mathcal{A}, \tag{154}$$

where "$\sim$" denotes structural equivalence (i.e., they have the same pattern of zeros and non-zeros).

In addition, with the right causal ordering(s), $\mathcal{A}$ can be arranged to be strictly lower-triangular, i.e., $P_\pi^\top \mathcal{A} P_\pi$ is lower-triangular, where $P_\pi$ is a permutation matrix representing the unknown causal ordering. Clearly, $P_\pi^\top (\mathbf{I}_n - \mathcal{A}) P_\pi$, as well as $P_\pi^\top G_{f_1^{-1}} P_\pi$, are also lower-triangular.

According to Eq. (153), we have

$$P_\pi^\top G_{f_1^{-1}} P_\pi = P_\pi^\top P G_{\hat{f}^{-1}} P_\pi, \tag{155}$$

where both sides are lower-triangular. For brevity, we denote $P_\pi^\top P$ as $P_{\tilde{\pi}}$ so that

$$P_\pi^\top G_{f_1^{-1}} P_\pi = P_{\tilde{\pi}} G_{\hat{f}^{-1}} P_\pi, \tag{156}$$

where both sides are also lower-triangular. Because the diagonal elements of $G_{f_1^{-1}}$ are non-zero, the diagonal elements of $P_\pi^\top G_{f_1^{-1}} P_\pi$ are also non-zero.

Then we aim to find the permutation matrices $P_{\tilde{\pi}}$ and $P_\pi$ to make the estimated inverse of the Jacobian lower-triangular. We then need to show: (1) if the causal ordering $P_\pi$ is unique (and unknown), $P_{\tilde{\pi}}$ is also unique; (2) if the causal ordering $P_\pi$ is not unique (but unknown), each of them corresponds to a unique $P_{\tilde{\pi}}$. Similar techniques have been used in (Shimizu et al., 2006; Reizinger et al., 2022) to bridge ICA to causal discovery. The two cases are considered as follows:

- If $P_\pi$ is unique (and unknown), we need to show that $P_{\tilde{\pi}}$ is also unique. Suppose we have two row-permutations $P_{\tilde{\pi}_1}$ and $P_{\tilde{\pi}_2}$ such that $P_{\tilde{\pi}_1} \neq P_{\tilde{\pi}_2}$ and both $P_{\tilde{\pi}_1} J_{(\hat{g} \circ \hat{f})^{-1}} P_\pi$ and $P_{\tilde{\pi}_2} J_{(\hat{g} \circ \hat{f})^{-1}} P_\pi$ are lower-triangular with no zero entries on the diagonal. Equivalently, $P_{\tilde{\pi}_1} J_{\hat{f}^{-1}} P_\pi$ and $P_{\tilde{\pi}_2} J_{\hat{f}^{-1}} P_\pi$ are lower-triangular since $J_{\hat{g}^{-1}}$ is a diagonal matrix. Suppose that $P_{\tilde{\pi}_1} J_{\hat{f}^{-1}} P_\pi$ is lower-triangular; it is impossible for $P_{\tilde{\pi}_2} J_{\hat{f}^{-1}} P_\pi$ to be also lower-triangular since $P_{\tilde{\pi}_1} \neq P_{\tilde{\pi}_2}$. Thus, $P_{\tilde{\pi}}$ must be unique given a unique $P_\pi$.

- If $P_\pi$ is not unique (but unknown), we can apply a similar argument above to identify a set of row-permutation matrices $\mathcal{P}_{\tilde{\pi}}$, each of which ensures the lower-triangularity (with no zero entries on the diagonal).

Therefore, we can always resolve the indeterminacy $P_{\tilde{\pi}}$ in Eq. (153) by the lower-triangularity of $P_{\tilde{\pi}} G_{\hat{f}^{-1}} P_\pi$, even though the causal ordering $P_\pi$ is unknown. As a result, we can identify $G_{f_1^{-1}}$, which leads to the identifiability of $\mathbf{I}_n - \mathcal{A}$ and clearly also that of $\mathcal{A}$. $\quad\square$

## C  ADDITIONAL DISCUSSIONS

In this section, we discuss the challenge of modeling general noise, emphasizing the distinctions between noise and content variables as explored in previous works.

Content variables are typically semantically meaningful and are often explicitly contrasted with style variables. This enables existing techniques to disentangle content from style through structured variability. For example, contrastive learning frameworks (Von Kügelgen et al., 2021) use paired observations differing only in style (e.g., images with the same object but different backgrounds) to disentangle content. In multi-domain settings (Kong et al., 2022), content remains invariant across $O(n)$ distinct domains characterized by different styles. Similarly, in intervention-based settings (Lachapelle et al., 2024), agents or environments serve as auxiliary variables that induce changes in the conditional distributions of latent variables. These structured variations provide the foundation for effective disentanglement.

In contrast, noise variables often lack semantic meaning and cannot be explicitly manipulated across multiple domains or paired in observations. This makes it infeasible to assume the existence of $O(n)$ conditional distributions or define contrastive objectives. As a result, existing frameworks designed for content-style disentanglement cannot be directly applied to general noise modeling.

To address this, we propose to only leverage the existence of variability in the latent distribution, requiring only two distinct distributions as a minimal degree of change. This relaxation reduces the need for $O(n)$ distinct distributions, which are common in existing frameworks, and broadens applicability to scenarios where the distribution is not completely invariant. This shift, from explicitly controlling different types of variables to achieve a required degree of change or transition, to accommodating general variability in scenarios where the distribution is not completely invariant, represents a significant technical contribution that is essential to address the unique contribution of modeling general noise.

## D EXPERIMENTAL DETAILS AND ADDITIONAL RESULTS

In this section, we present additional details regarding the experimental settings (Sec. D.1) as well as supplementary empirical results (Sec. D.2).

### D.1 EXPERIMENTAL DETAILS

We provide supplementary details of the experimental configurations as follows:

**Evaluation Metric.** We assess the correspondence between ground-truth and recovered latent variables using the Mean Correlation Coefficient (MCC). To compute MCC, we first apply an element-wise transformation learned through regression, then calculate the pairwise correlation coefficients between the true and recovered latent variables. An assignment problem is then solved to match each recovered variable with the ground-truth one showing the highest correlation. MCC, commonly used in the literature for measuring identifiability under element-wise transformations (Hyvärinen & Morioka, 2016), serves as our evaluation metric.

**Implementaion Details.** In our experiments with synthetic datasets, we use a sample size of $10,000$, with a learning rate of $0.01$ and a batch size of $200$. The flow-based models are trained using 10 coupling layers. All experiments are performed using the official GIN implementation[1](Sorrenson et al., 2020), incorporating an additional $\ell_1$ regularization term on the Jacobian of the estimated generating function, with a regularization coefficient of $0.1$.

### D.2 ADDITIONAL RESULTS

**Additional Synthetic Experiments.** In this section, we present additional experiments under different specific settings to validate our theoretical results. We generate datasets for two scenarios: (1) a setting with only additive noise (*Model 1*) and (2) a setting with both general noise and distortion (*Model 2*). For *Model 1*, we use additive Gaussian noise. For *Model 2*, we apply an element-wise nonlinear transformation and add Gaussian noise, as described in Eq. (4) and the corresponding mixture model in Eq. (3). All other experimental details follow those of the main simulations. We construct datasets with

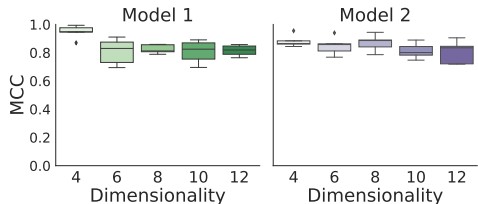

Figure 8: Identification of latent variables w.r.t. different $m$ for *Model 1* and *Model 1*, where $n = m/2$.

varying numbers of variables and conduct five random trials for each configuration. The results, presented in Fig. 8, demonstrate that all latent variables are identifiable across different settings, providing further validation of our theoretical findings.

**Additional Real-world Experiments.** As outlined in Sec. 5, here we include the results on the real-world image datasets. we present the results from experiments conducted on real-world image datasets. Specifically, we evaluated our model on the Fashion-MNIST (Xiao et al., 2017) and

---

[1]https://github.com/VLL-HD/GIN

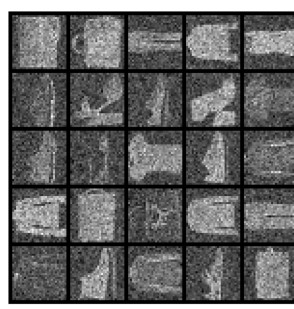

(a) Fashion-MNIST with noise        (b) EMNIST with noise

Figure 9: Samples of images with noise from (a) Fashion-MNIST and (b) EMNIST datasets.

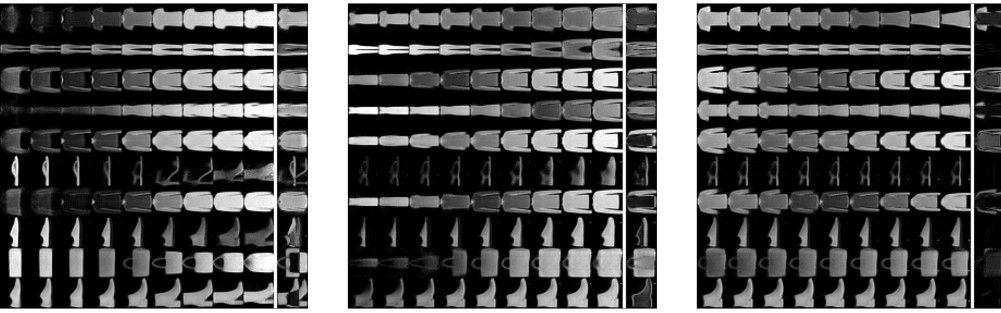

Figure 10: Identified variables from Fashion-MNIST.

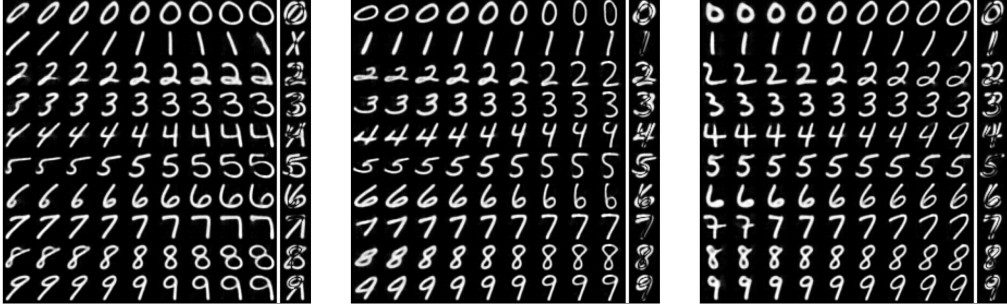

Figure 11: Identified variables from EMNIST.

EMNIST (Cohen et al., 2017) datasets. Fashion-MNIST contains $60,000$ images of clothing items, each with a resolution of $28 \times 28$ pixels, while EMNIST consists of $240,000$ images of handwritten digits, also sized $28 \times 28$. We added noises (mixtures of multiplicative and additive Gaussians) to the images to evaluate whether we could identify latent variables from observed pixels with noises. Figure 9 shows the samples of images with a rather complicated form of noise.

Figures 10 and 11 show the identified components, highlighting the top three concepts with the largest standard deviations (SDs) from our analysis. Each sub-figure presents reconstructed images where the corresponding latent variable varies from $-4$ to $+4$ SDs, demonstrating its effect on the image. The rightmost column displays a heat map of absolute pixel differences between $-1$ and $+1$ SDs, further visualizing these changes in the reconstruction. The identified latent variables clearly capture meaningful semantics. For instance, variables identified from EMNIST represent left-leaning, height, and right-leaning characteristics. This confirms that semantically meaningful latent variables can be identified from real-world images even in the presence of general noise, illustrating the practical viability of the proposed nonparametric identifiability in real-world scenarios.

