# OpenReview forum: "On the Identifiability of Nonlinear Representation Learning with General Noise"
_ICLR.cc/2025/Conference — ICLR 2025 Conference Withdrawn Submission_

### Official Review · Reviewer_KLTp · 2024-10-28

**Soundness:** 2
**Presentation:** 2
**Contribution:** 2
**Rating:** 5
**Confidence:** 4

**Summary:**

In recent years, there has been an active development on theoretical and practical research towards learning identifiable representations.
The most common family of data generation assumptions take the following form -- z ~Pz, x <--f(z), where z is some latent variable sampled from a distribution Pz, and f is a diffeomorphism that mixes the latents to generate the observation x. For varying assumptions either on Pz or mixing maps or other forms of assumptions on weak supervision, there has been a large body of work that establishes varying forms of identification guarantees. In contrast to existing works, this work places a special emphasis on the role of noise in generation of x. The authors propose a family of results under different assumption.

In Theorem 1, the authors study x <--f(z,e), where z is latent, e is noise, f is diffeomorphism.
In Theorem 2, the authors study x <-- f(z) +e, and show that some of the requirements on multiple domains in Theorem 1 can be relaxed.
In Theorem 3 and 4, the authors study, a model with non-linear distortions and measurement error and arrive at identification guarantees (representation identification and structure identification).

Towards, the end the authors conduct experiments to reflect the theoretical claims.

**Strengths:**

The current frameworks on identifiability have mostly studied additive noise models and here the authors mean to go beyond that family of models. This is an important problem, which has not been well studied and it is good to see authors taking a stab at it.    The paper tries does a good job of dividing the analysis based on family of assumptions -- i) with non-additive noise models x <--f(z,e), ii) additive noise models x <--f(z) +e, iii) non-linear distortion models with noise, and finally to iv)models with measurement error.

**Weaknesses:**

1. **On noise modeling**:  The authors seem to overclaim that they are the first to arrive at identifcation guarantees with general noise. I believe they are referring to the first theorem. In the first theorem the authors use the following model x<-- f(z,e), where f is a diffeomorphism. I would argue that existing frameworks on partial identification already cover these types of problems even though they do not state it like this.  One can always state existing works on partial identification in the above light. Suppose I divide my latents z into two parts z1 and z2, and x <--f(z1,z2). In this case, one can seek partial identification guarantees, where one estimates \hat{z}1 that only identifies z1 element-wise and rest of the z2 terms correspond to noise that we don't care about identifying.
Let us take the work from Kugelgen et al. https://proceedings.neurips.cc/paper_files/paper/2021/file/8929c70f8d710e412d38da624b21c3c8-Paper.pdf. One could re-interpret their result and say the following. In their result, the authors show that there are two types of latents, content latent and style latent. If one observes a pair of observations where style latent changes but the content latent stays the same, then one can separate the content latents. At this point the problem reduces to standard noise free representation identification.
One could also use frameworks https://arxiv.org/pdf/2401.04890 and divide latents into two blocks. The actions of an agent may impact only the latents u want to identify.
Then there is the framework from https://arxiv.org/pdf/2306.06510, which I believe the authors are already using in their results. I would like to better understand the take of authors on the above perspective and why do they not acknowledge this properly in their work. By overclaiming about noise, the authors are not highlighting the specific challenges that they face beyond separation of content and style.



2. **On Theorem 1**:

   a) The structure and assumptions in Theorem 1 are reminiscent of the assumptions in Theorem 1 in https://proceedings.neurips.cc/paper_files/paper/2022/file/6801fa3fd290229efc490ee0cf1c5687-Paper-Conference.pdf and Theorem 4.1 in https://proceedings.neurips.cc/paper_files/paper/2023/file/2aebc17b683792a17dd4a24fcb038ba6-Paper-Conference.pdf.
     Can the authors contrast their proof technique from the above papers and give some intutions on what differs?

   b) In existing works on structural sparsity, e.g., https://proceedings.neurips.cc/paper_files/paper/2022/file/6801fa3fd290229efc490ee0cf1c5687-Paper-Conference.pdf, the authors relied on independence assumption on the latents. In this work, the authors do not seem to require that assumption. What is the assumption that the authors make that allows them to get rid of independence assumption? Also, this seems a bit odd because it seems to imply having noise in the setting makes the problem easier than https://proceedings.neurips.cc/paper_files/paper/2022/file/6801fa3fd290229efc490ee0cf1c5687-Paper-Conference.pdf, which is fairly counterintuitive.

   c) As I stated above, the problem of identification under noise has been reduced in the authors framework to that of block identification. For block identification, the authors proof of Theorem 1 is a direct combination of result from Kong et al. and structrual sparsity work https://proceedings.neurips.cc/paper_files/paper/2022/file/6801fa3fd290229efc490ee0cf1c5687-Paper-Conference.pdf. Can the authors better tell if there is really new proof techniques that were needed to solve the problem? I again emphasize that authors seemed to highlight noise as a very difficult problem but reduce it to two well studied cases -- block identification (Kugelgen et al., Kong et al.) or deconvolution via additive noise models (Khemakhem et al.).

3. **On Theorem 2**: In this Theorem, the authors rely on additive noise assumption. Here I would like to understand why is this result any different from a generalization of Theorem 1 in https://proceedings.neurips.cc/paper_files/paper/2022/file/6801fa3fd290229efc490ee0cf1c5687-Paper-Conference.pdf to additive noise models? I would argue that due to additive noise models this generalization is trivial because you rely on the deconvolution argument from Khemakhem et al.. In Khemakhem's argument, one requires to know the noise distribution to cancel both sides out in the Fourier transform (See equation 20-27 in https://arxiv.org/pdf/1907.04809). If you resort to the same sort of assumptions as Khemakhem et al., then I don't think you are going beyond what already exists in identification literature in your results (Theorem 2 to 4 all rely on additive noise models).

4. **On Theorem 3**:  The setup of Theorem 3, i.e., equations 3 and 4 seem to combine the models used in Theorem 1 and 2. Can the authors clarify this? For equation 4, we can use Theorem 2 to obtain elementwise transform of x*. After that one could use the inferred x* and use Theorem 1 for obtaining z. Why is presented as a special thing? This could just be a corollary.

5.  **On experiments:** It would have been nice if there was one experiment reflecting the unique settings of the different theorems. I do not see that is the case. For instance, synthetic expmts for Theorem 2-4 would have been nice, both representation identification ones and causal discovery ones.

6.  **On the writing**: I found the writing quite verbose in many places. The authors seem to give examples without giving citations. For instance,  see lines 165-188, lines 298-308, lines 357-369.

**Questions:**

In the weaknesses section itself I have mentioned my concerns and questions. Please refer to that section.

---

> ### Author Response · Authors · 2024-11-21
> **We deeply value your detailed review and the insightful feedback you have shared (1/4)**
>
> We deeply value your detailed review and the insightful feedback you have shared. All of these constructive suggestions have further improved our manuscript. In light of these, we have incorporated **several new discussions** and a **new section** in the updated manuscript, particularly focusing on **clarifying theoretical results** within the context of broader literature. Moreover, we have also conducted **additional experiments** as you suggested. Please kindly find our detailed response to each of your comments below. If you have any additional feedback, we would be truly grateful to hear it and would be delighted to provide further clarifications.
>
>
> **Q1:** Difference on the noise modeling and the separation of content and style.
>
> **A1:** This is an excellent point, which inspired us to add a new section for detailed discussion and clarification. These works are highly interesting and contribute significant theoretical advances in identifying latent variable models. Regarding the two considered settings, we believe the key distinction lies in the nature of the noise and content variables.
>
> For content, they are usually semantically meaningful and are often explicitly compared with style variables. This makes it natural for existing techniques to disentangle content from style. For instance, in contrastive learning (Von Kügelgen et al., 2021), one can find pairs of observations differing only in their styles and use contrastive learning to disentangle the content. Similarly, in multi-domain settings with $O(n)$ domains (Kong et al., 2022), the content remains invariant while there are $O(n)$ (e.g., $2n+1$) distinct domains characterized by different styles. Similarly, in scenarios involving actions or interventions (Lachapelle et al., 2024), agents/domains/environments act as auxiliary variables, inducing multiple changes in the conditional distribution by intervening/impacting on latent variables.
>
> However, for noise, they often lack semantic meaning and cannot be explicitly changed across multiple domains like actions or styles, nor can they be paired in observations as in contrastive learning. Thus, it is challenging to assume the existence of $O(n)$ conditional distributions based on $O(n)$ auxiliary variables or to define contrastive objectives for noise modeling. This is a specific challenge that we face and cannot be directly addressed by previous frameworks. Moreover, for learning SEMs with measurement error, we have more noises compared to observed ones, which cannot be solved by previous methods.
>
> Therefore, for the modeling of general noise, we propose assuming only the existence of variability in the noise distribution, specifically requiring two distinct distributions as a minimal degree of change. This relaxation, from $O(n)$ to two distributions, not only quantitatively weakens the assumption but also qualitatively broadens its applicability. It shifts from explicitly controlling different types of variables to achieve a required degree of change or transition, to accommodating any scenarios where the distribution exhibits variability and is not completely invariant. This is clearly a significant technical contribution that addresses the unique challenges in modeling general noise.
>
> In light of your great question, we have also added a new section for detailed discussion, with a particular focus on the difference with previous work in content-style separation:
>
> > “In this section, we discuss the challenge of modeling general noise, emphasizing the distinctions between noise and content variables as explored in previous works.”
>
> > “Content variables are typically semantically meaningful and are often explicitly contrasted with style variables. This enables existing techniques to disentangle content from style through structured variability. For example, contrastive learning frameworks (Kugelgen et al., 2021) use paired observations differing only in style (e.g., images with the same object but different backgrounds) to disentangle content. In multi-domain settings (Kong et al., 2022), content remains invariant across $O(n)$ distinct domains characterized by different styles. Similarly, in intervention-based settings (Lachapelle et al., 2024)}, agents or environments serve as auxiliary variables that induce changes in the conditional distributions of latent variables. These structured variations provide the foundation for effective disentanglement.”
>
> > “In contrast, noise variables often lack semantic meaning and cannot be explicitly manipulated across multiple domains or paired in observations. This makes it infeasible to assume the existence of $O(n)$ conditional distributions or define contrastive objectives. As a result, existing frameworks designed for content-style disentanglement cannot be directly applied to general noise modeling.”

---

> > ### Author Response · Authors · 2024-11-21
> > **We deeply value your detailed review and the insightful feedback you have shared (2/4)**
> >
> > > “To address this, we propose to only leverage the existence of variability in the latent distribution, requiring only two distinct distributions as a minimal degree of change. This relaxation reduces the need for $O(n)$ distinct distributions, which are common in existing frameworks, and broadens applicability to scenarios where the distribution is not completely invariant. This shift, from explicitly controlling different types of variables to achieve a required degree of change or transition, to accommodating general variability in scenarios where the distribution is not completely invariant, represents a significant technical contribution that is essential to address the unique contribution of modeling general noise.”
> >
> > Thanks again for your valuable insight. Please kindly let us know if there are any further questions.
> >
> >
> > **Q2:** Questions on Theorem 1.
> >
> > **A2:** We sincerely appreciate all these questions and the time you spent on reviewing our manuscript. After carefully reviewing the mentioned works, we believe that there may exist some potential misunderstanding, which could be clarified by the point-by-point responses below.
> >
> > - **Q2.1:** Can the authors contrast their proof technique from (Zheng et al., 2022, Zheng & Zhang, 2023)?
> >
> > - **A2.1:** Thank you for the question. Compared to Theorem 1 in (Zheng et al., 2022), our setting involves general noise variables, and thus need additional assumption of variability to make them distinguishable. In contrast, (Zheng et al., 2022) does not address this challenge as it does not involve disentangling different types of variables in the proof.
> >
> >   Compared to Theorem 4.1 (and Theorem 4.2) in (Zheng & Zhang, 2023), our results significantly generalize their conditions by requiring only two conditional distributions to disentangle the latent variables dependent on domains, whereas (Zheng & Zhang, 2023) relies on $n+1$ domains to construct a full-rank linear system for disentangling the changing part. While Theorem 4.1 in (Zheng & Zhang, 2023) includes a variability condition with two domains, it is only used to disentangle the invariant part. They then require $n+1$ domains to disentangle the changing part, whereas we identify both invariant and changing latent variables with only two domains.
> >
> >   In summary, our proof fully leverages the variability assumption, demonstrating that latent variables can be disentangled from invariant noises with just two domains. This contrasts with (Zheng et al., 2022), which does not require disentanglement of variable types, and (Zheng & Zhang, 2023), which requires $n+1$ domains to disentangle the changing latent variables.
> >
> >   We have also added the following proof sketch for further clarification:
> >
> >   > “*Proof Sketch.* We leverage distributional variability across two domains of the latent variables $\mathbf{z}$ to disentangle $\mathbf{z}$ and $\boldsymbol{\epsilon}$ into independent subspaces. To distinguish general noise from latent variables, we use the conditional independence between $\mathbf{z}$ and $\boldsymbol{\epsilon}$ alongside the variability within $\mathbf{z}$. Following (Zheng et al., 2022), a structural sparsity condition then identifies individual components of $\mathbf{z}$ in the nonlinear case. Specifically, for each latent variable, the intersection of parental sets from a subset of observed variables uniquely specifies it. Since we only achieve relations among supports due to the nonparametric setting, there exists an unresolved element-wise transformation. Thus, we achieve element-wise identifiability for the latent variables $\mathbf{z}$ (Defn. 1.)”
> >
> >   We hope these discussions and updates could clarify any potential confusion. Thanks again for raising this point. Please kindly let us know if you have any further feedback.

---

> > > ### Author Response · Authors · 2024-11-21
> > > **We deeply value your detailed review and the insightful feedback you have shared (3/4)**
> > >
> > > - **Q2.2:** In existing works on structural sparsity, e.g., (Zheng et al., 2022). In this work, the authors do not seem to require that assumption. What is the assumption that the authors make that allows them to get rid of the independence assumption?
> > >
> > > - **A2.2:** Thanks for raising this interesting point. In fact, the structural sparsity assumption in both (Zheng et al., 2022) and our work directly implies independence. We conjecture that this might be the reason that, although independence is mentioned in (Zheng et al., 2022), the proof in (Zheng et al., 2022) never utilizes the most common property of it, i.e., the factorization of the joint density.
> > >
> > >   Specifically, if two latent variables $\mathbf{z}_k$ and $\mathbf{z}_v$ are dependent, then it is either ${(\mathcal{F}\_{z})}\_{:, q} \subseteq {(\mathcal{F}\_{z})}\_{:, v}$ or ${(\mathcal{F}\_{z})}\_{:, v} \subseteq {(\mathcal{F}\_{z})}\_{:, q}$. Without loss of generality, suppose ${(\mathcal{F}\_{z})}\_{:, q} \subseteq {(\mathcal{F}\_{z})}\_{:, v}$. In this case, it becomes impossible to find a set $\mathcal{C}\_{q}$ s.t. $\bigcap\_{i \in \mathcal{C}\_{q}} {(\mathcal{F}\_{z})}\_{i, :}=\{q\}$. This is because for every row in the Jacobian where the $q$-th column is nonzero, the $v$-th column is also nonzero, making it impossible to uniquely disentangle the $q$-th column by the intersection of rows.
> > >
> > >   To highlight this point, we have also added the following note in the updated manuscript:
> > >
> > >   > “It might be worth noting that the structural sparsity implies the independence among latent variables $\mathbf{z}$. Specifically, if two latent variables are dependent, it becomes impossible to disentangle one of them by the intersection of a set of observed variables that are influenced by these latent variables.”
> > >
> > >
> > > - **Q2.3:** Can the authors better tell if there are really new proof techniques that were needed to solve the problem in Theorem 1, other than a direct combination of results from (Kong et al., 2022) and (Zheng et al. 2022)?
> > >
> > >
> > > - **A2.3:** Thanks a lot for the question, which provides an opportunity for us to further clarify our results. As discussed in detail in A2.1 and A2.2, there is no previous work that can disentangle the changing latent part with only two domains. Thus, for block identification, our proof of Theorem 1 is not a direct combination of results from (Kong et al., 2022) and (Zheng et al., 2022).
> > >
> > >   Specifically, in (Kong et al., 2022), given only two domains, their results show that the estimated invariant part does not depend on the changing part. However, the changing part itself is not disentangled; its estimation may still be mixed with the invariant variables. In contrast, our proof demonstrates block identifiability for both the invariant and changing parts with only two domains (see Eqs. 27–36). This distinction is critical, as without disentangling the changing latent variables $\mathbf{z}$, we could not achieve the component-wise identifiability results for $\mathbf{z}$ presented in Theorem 1.
> > >
> > > **Q3:** I would like to understand why is Theorem 2 any different from a generalization of the identifiability based on structural sparsity to additive noise models?
> > >
> > > **A3:** Thank you for raising this point. Indeed, Theorem 2 can be seen as a generalization of identifiability based on structural sparsity to additive noise models, which can be addressed using the deconvolution strategy presented in (Khemakhem et al., 2020). At the same time, as highlighted in the remark (L293-297), additive noise becomes even simpler to remove when adopting a structural perspective. Specifically, the derivative of the observed variables with respect to the latent variables is unaffected by additive noise, making it easier to disentangle. Thus, Theorem 2 is introduced primarily to establish a connection between our work and existing methods for handling noise. This connection has been emphasized in the manuscript (L300-304) to align our contributions with the broader context of identifiability research.
> > >
> > >
> > > **Q4:** It seems that Theorem 3 can be derived based on Theorems 1 and 2, and thus it could just be a corollary.
> > >
> > > **A4:** Thank you for your valuable suggestion. In response, we have updated the manuscript to present Theorem 3 as a corollary (Corollary 1 in the updated manuscript) to avoid potential confusion. The result was intended to formally introduce the concept of distortion, which serves as a natural transition to exploring a new setting—structure learning with nonlinear measurement error.

---

> ### Author Response · Authors · 2024-11-21
> **We deeply value your detailed review and the insightful feedback you have shared (4/4)**
>
> **Q5:** It would have been nice if there were experiments for the other specific settings.
>
> **A5:** We sincerely appreciate your constructive feedback. In light of it, we have conducted **additional experiments** and included the results in Appx. D.2. As shown in Figure 11, latent variables are identifiable in the presence of noise and distortion in the considered settings. Additionally, we can recover the structure of the Jacobian; however, since we have not proposed a practical causal discovery algorithm, we focus here on the identification of latent variables.
>
> **Q6:** I found the writing quite verbose in many places. The authors seem to give examples without giving citations.
>
> **A6:** Thanks a lot for your suggestions on the writing. In light of your feedback, we believe that removing some examples would provide more space for including proof sketches in the main paper, which will highlight the unique technical contributions in the proof as well as making the discussion more concise. We have made the changes in the updated manuscript accordingly (e.g., L168-175). We sincerely appreciate your time devoted to improving the manuscript, and looking forward to any further feedback.
>
>
> ---
>
> **References**
>
>
> [1] Von Kügelgen et al., Self-supervised learning with data augmentations provably isolates content from style, NeurIPS 2021
>
> [2] Kong et al., Partial identifiability for domain adaptation, ICML 2022
>
> [3] Lachapelle et al., Nonparametric partial disentanglement via mechanism sparsity: Sparse actions, interventions and sparse temporal dependencies. arXiv 2024
>
> [4] Zheng et al., On the identifiability of nonlinear ICA: Sparsity and beyond, NeurIPS 2022
>
> [5] Zheng and Zhang, Generalizing nonlinear ICA beyond structural sparsity, NeurIPS 2023
>
> [6] Khemakhem et al., Variational autoencoders and nonlinear ICA: A unifying framework, AISTATS 2020

---

> > ### Author Response · Authors · 2024-11-25
> > **We greatly appreciate the time and effort you’ve dedicated to reviewing our work**
> >
> > Dear Reviewer KLTp,
> >
> > We greatly appreciate the time and effort you’ve dedicated to reviewing our work. As the discussion period draws to a close, we would be truly grateful if you could let us know whether our responses have sufficiently addressed your concerns.
> >
> > With gratitude,
> >
> > Authors of Submission7585

---

> > > ### Comment · Reviewer_KLTp · 2024-11-26
> > >
> > > I thank the reviewers for their time and efforts in their responses. Based on the responses, I am still quite convinced that the framework is not as different from existing works as it is made out to be.
> > >
> > > 1. I still stand by the remark that existing works (Kugelgen et al.) to disentangle content from style are quite relevant to this setting as well. You could say that they use style variables in their story and style variables are semantically meaningful. But those style variables mathematically speaking could as well be noise variables and then you will be able to isolate content from noise.
> > >
> > > 2. I find the proofs to be a combination of the techniques already introduced in Kong et al. and Zheng et al. It is not bad to combine existing proofs but if that is the main contribution then it feels somewhat incremental and not very surprising.  The results with additive noise can be done with standard strategies of deconvolution.
> > >
> > > I believe that authors should present a more convincing case for their contributions. Here is a concrete theoretical suggestion I have for authors. Based on existing frameworks of identifiability, it is quite clear one can tackle additive noise or one can tackle noise via treating it as latent variable in the diffeomorphism. But can we go beyond these two settings in a very clear way, especially when it is not a diffeomorphism. Consider x <-- f(z,e), where e is noise, x could be used to invert back z but not e. In such a framework developing identification guarantees would be a novel departure from existing works.
> > >
> > > In view of the above, I maintain my rating.

---

> > > > ### Author Response · Authors · 2024-11-26
> > > > **Thank you very much for your thoughtful suggestion**
> > > >
> > > > Dear Reviewer KLTp,
> > > >
> > > > Thank you very much for your thoughtful suggestion. Indeed, extending the current framework to the non-invertible case would be a highly interesting direction for future research. At the same time, given the nonparametric nature of the setting, it remains unclear whether disentangling noise from a non-invertible mapping—where information is fundamentally lost—is feasible. By significantly relaxing the existing conditions for block-wise disentanglement (e.g., from $O(n)$ domains to just $2$ domains), our results represent an essential step toward that ultimate goal.
> > > >
> > > > Meanwhile, as highlighted throughout the paper, some of our results use the structural conditions from Zheng et al. and the variability condition from Kong et al. At the same time, we would like to emphasize the unique contributions of our work, which, to the best of our knowledge, **have not been achieved in any prior research**, even from a purely technical perspective:
> > > >
> > > > - **Representation Identification:** Previous theories and techniques typically require **$O(n)$ domains** to disentangle the changing latent components. In contrast, our theory demonstrates that as few as **$2$ domains** may suffice.
> > > >
> > > > - **Structure Identification:** We establish the identifiability of the hidden causal graph for **nonparametric structural equation models** (with general nonlinear functional relations and non-additive exogenous noise) under **nonlinear measurement error**. To the best of our knowledge, no prior work has introduced identifiability results in such a general setting.
> > > >
> > > > Lastly, we are sincerely grateful for your time and insightful suggestions. We genuinely hope that you might reconsider your assessment and give our work another chance. Of course, we deeply respect your perspective and appreciate the thoughtful effort you have put into reviewing our work.
> > > >
> > > > Best regards,
> > > >
> > > > Authors of Submission7585

---

### Official Review · Reviewer_LShV · 2024-10-29

**Soundness:** 3
**Presentation:** 4
**Contribution:** 4
**Rating:** 8
**Confidence:** 4

**Summary:**

This paper clearly sets up a set of nonparametric assumptions that allow for identifiability of latent factors, a fundamental problem in theoretical machine learning.

**Strengths:**

This paper clearly sets up a set of nonparametric assumptions that allow for identifiability of latent factors, a fundamental problem in theoretical machine learning. The assumptions are somewhat strong but are nicely set up and create a framework where proofs seem relatively straightforward.

The framework is applied to a wide variety of special cases that have arisen in the literature, yielding strong results in each case.

If the questions below are addressed, I believe this result would be a strong contribution to the space.

**Weaknesses:**

While notation is thoroughly defined and overall there is a lot of discussion, still there are many key places where details are either missing or difficult to ascertain (see questions).

The structural sparsity assumption is extremely strong, albeit somewhat understandable. That said, the paper could be somewhat more realistic when discussing the assumptions - many references are made to real-world data types that do not immediately seem to necessarily follow this framework exactly.

The assumptions on nondegeneracy preclude linear functions, which is also understandable.

Technically, the variability assumption is not satisfiable, since A = Z x E is not disallowed (since Z, E are not subsets of themselves). Hence p(A|d_1) = p(A|d_2) = 1 identically, violating the assumption. I think this is possibly a typo.

**Questions:**

Several parts of the assumptions are not clear to me. For instance, is it assumed that the 2 domains are labeled? Also, in Eq (30), it is stated that \hat{F} is estimated under a sparsity constraint - what is this constraint and where is it listed in the assumptions?

Eq (35) - why is this without loss of generality?

In the experiment section, can more details be provided on this split between latent factors and noise factors in the synthetic data, or is this as extremely simple as it sounds? What is the "base" experiment in this case?

For the variability assumption - why are rectangular sets not included? Where is this assumption used in the proof of Thm 1, for instance?

---

> ### Author Response · Authors · 2024-11-21
> **We are very grateful for the time and effort you dedicated and constructive feedback (1/2)**
>
> We are very grateful for the time and effort you dedicated to thoroughly reviewing our manuscript and providing constructive feedback, which has been invaluable in improving its quality. In particular, your thoughtful suggestions have prompted us to include several **clarifications** and **new discussions** to enhance the clarity of our messages. **New experiments** have been also been conducted to further support our theoretical findings. We kindly invite you to review our detailed responses below and would sincerely appreciate any additional feedback you may have.
>
>
> **Q1:** While notation is thoroughly defined and overall there is a lot of discussion, still there are many key places where details are either missing or difficult to ascertain.
>
> **A1:** Thank you very much for taking the time to thoroughly review our manuscript. We sincerely appreciate your comments. Here we provide point-to-point responses to the mentioned details as follows:
>
> - **Q1.1:** Is it assumed that the $2$ domains are labeled?
>
> - **A1.1:** Thank you for the question. Yes, as mentioned in L469 and L475, the two domains are labeled. It might be worth noting that, most previous works (see e.g., a recent survey (Hyvärinen et al., 2024)) require $2n+1$ labeled domains, where $n$ is the number of latent variables; while we only necessitate $2$ domains for arbitrary number of latent variables, which is essential a minimal degree of variability. To avoid potential confusion, in addition to the current notes, we have further highlighted it earlier in the updated manuscript:
>
>   > “These two domains are realizations of a domain variable $\mathbf{u}$, which are observed and labeled.”
>
> - **Q1.2:** What is the sparsity constraint and where is it listed in the assumptions?
>
> - **A1.2:** Thanks for your question. The sparsity constraint refers to an $\ell_0$ regularization on $\hat{\mathcal{F}}_\hat{z}$ during estimation, approximated using $\ell_1$ for gradient-based optimization (L468). Since the identifiability studies what conditions make the data generating process identifiable, and the sparsity constraint is a regularization used during estimation, we previously did not list it in the assumptions. However, we agree with you that introducing it in the theorem could help avoid potential confusion. Therefore, we have updated the manuscript and added the following clarification to the theorems:
>
>   > “Together with a $\ell_0$ regularization on $\hat{\mathcal{F}}\_\hat{z}$ during estimation (${\\|\hat{\mathcal{F}}\_{\hat{z}}\\|}_0 \leq {\\|\mathcal{F}_z\\|}_0$), suppose the following assumptions …”
>
>   Thank you again for your constructive feedback. If you have any additional suggestions, please feel free to share them, and we would be more than happy to incorporate further changes to improve clarity.
>
> - **Q1.3:** Why is Eq. 35 (now Eq. 54) without loss of generality?
>
> - **A1.3:** This is because $j_1 \neq j_2$, and thus it is either $j_3 \neq j_1$ or $j_3 \neq j_2$. Since $j_1$ and $j_2$ are symmetric in the proof, we can assume $j_3 \neq j_1$ without loss of generality, making Eq. 35 (now Eq. 54) valid.
>
> - **Q1.4:** Can more details be provided on this split between latent factors and noise factors in the synthetic data? What is the “base” experiment in this case?
>
> - **A1.4:** Thanks so much for your suggestions. As mentioned in the experimental setup, the main difference between latent and noise variables in generating synthetic dataset is that, for latent variables, we sample them from two distinct multivariate Gaussian distributions conditioning on the corresponding domain index, while the noise is sampled from a single multivariate Gaussian. The corresponding details of the distributions are included in the appendix, which we now moved to the main paper in the updated manuscript to make details more noticeable:
>
>   > “During training, latent variables are drawn from two multivariate Gaussian distributions to satisfy the variability condition, while noise is also sampled from a separate multivariate Gaussian, with means sampled uniformly from the range $[-5, 5]$ and variances sampled uniformly from $[0.5, 2.5]$.”
>
>   For the baseline model, we use a fully connected structure to violate the structural sparsity condition, and use a single domain to violate the variability condition. We have modified the current description (L479-481) as follows:
>
>   > “In contrast, the baseline model (Base) violates key assumptions, particularly those related to structural sparsity (by a fully connected structure) and variability (by sampling from a single domain).”

---

> > ### Author Response · Authors · 2024-11-21
> > **We are very grateful for the time and effort you dedicated and constructive feedback (2/2)**
> >
> > - **Q1.5:** For the variability assumption - why are rectangular sets not included? Where is this assumption used in the proof of Thm. 1, for instance?
> >
> > - **A1.5:** Thank you for your question. The variability assumption is used to prove that the bottom-left block of the Jacobian, i.e., $\dfrac{\partial \boldsymbol{\epsilon}}{\partial \hat{\mathbf{z}}}$, is zero. As mentioned there, this part of the proof is directly based on steps 1, 2, and 3 in the proof of Theorem 4.2 in (Kong et al. 2022), where the variability assumption is used. For the ease of reference, we have now included the related result in (Kong et al. 2022) as Lemma 1 in Appx. B.1, as well as the full proof with notations being transferred to our setting. Technically, the rectangular sets are not included to build the contradiction needed in the proof.
> >
> > **Q2:** The structural sparsity assumption is extremely strong, albeit somewhat understandable. That said, the paper could be somewhat more realistic when discussing the assumptions - many references are made to real-world data types that do not immediately seem to necessarily follow this framework exactly.
> >
> > **A2:** Thanks so much for your suggestion. Indeed, given the complexity and randomness of the real-world hidden generating process, some scenarios mentioned in our discussion may not perfectly align with the conditions exactly. In light of your constructive feedback, we have removed these examples, and added the following highlight on the limitations:
> >
> > > “Moreover, for real-world scenarios, it is extremely challenging to make sure that all conditions on the latent data generating process are perfectly satisfied and the distributions are perfectly matched after estimation. Bridging the gap requires a thorough study of the finite sample error and the robustness of the identification, which remains an open challenge in the literature.”
> >
> > At the same time, we believe the structural sparsity is more likely to be satisfied when the number of observed variables exceeds the number of latent variables. While we fully acknowledge that this assumption is quite strong when the numbers of latent and observed variables are equal—a common setting in previous work—our approach allows for additional observed variables, making the structural sparsity condition more natural in our context. As mentioned in L238-245, the structural sparsity condition only requires a subset of the observed variables—potentially as few as one or two—to satisfy the conditions. Therefore, in cases where additional observed variables are available or can be introduced (e.g., by adding more microphones), the assumption becomes much more plausible. Empirical evidence from Zheng & Zhang (2023) further supports this, showing that when the number of observed variables is three times the number of latent variables, approximately 80% of random structures satisfy the condition, and this percentage approaches 100% when the ratio increases to five or more.
> >
> > **Q3:** The assumptions on nondegeneracy preclude linear functions, which is also understandable.
> >
> > **A3:** Yes, you are absolutely right—this assumption precludes purely linear functions where the Jacobian remains invariant. When the Jacobian is invariant, it cannot span the support space and, as a result, fails to capture the structural information necessary for identifiability.
> >
> >
> > **Q4:** A typo in the variability assumption.
> >
> > **A4:** We sincerely appreciate you for catching this typo. Since the assumption was proposed by previous work (Kong et al. 2022), we have also confirmed your findings with them. Thanks so much for your careful attention. We have corrected it in the updated manuscript accordingly.
> >
> >
> > ---
> >
> > **References**
> >
> > [1] Hyvärinen et al., Identifiability of latent-variable and structural-equation models: from linear to nonlinear, Annals of the Institute of Statistical Mathematics, 2024
> >
> > [2] Kong et al., Partial identifiability for domain adaptation, ICML 2022
> >
> > [3] Zheng and Zhang, Generalizing nonlinear ICA beyond structural sparsity, NeurIPS 2023

---

> > > ### Author Response · Authors · 2024-11-21
> > > **Thanks so much for your encouragement**
> > >
> > > Thank you so much for your updated rating—it means a great deal to us and provides significant encouragement! We deeply appreciate your support and thoughtful feedback.

---

### Official Review · Reviewer_QWn8 · 2024-11-01

**Soundness:** 3
**Presentation:** 3
**Contribution:** 2
**Rating:** 5
**Confidence:** 2

**Summary:**

This paper explores learning the latent structure of data in the presence of non-parametric noise. The author derived conditions for model generation under which the latent variables can be identified. The analysis was then extended to various data generation models, establishing identifiable conditions for each.

**Strengths:**

The motivations and results are clearly presented, despite the theoretical focus and extensive mathematical derivations. The assumptions and extensions of the results are explained in detail. The main result (Theorem 1) is effectively extended to several commonly encountered models in Theorems 2, 3, and 4.

**Weaknesses:**

1. It is unclear how Definition 1 (element-wise identifiability) is commonly used in the field of latent variable identification, as only two papers (by the same author) are cited below this definition.
2. In all the theorems, the author notes that the dataset should be sufficiently large. However, it is unclear what qualifies as 'large' and how this requirement is applied in the proofs.
3. The main challenge of proving Theorem 1, as compared to previous works, is not clearly explained.

**Questions:**

See the weaknesses above.

---

> ### Author Response · Authors · 2024-11-21
> **We sincerely appreciate your insightful feedback and valuable comments (1/3)**
>
> We sincerely appreciate your insightful feedback and valuable comments, which have been immensely helpful in improving our manuscript. In response, we have included **additional clarifications**, **expanded discussions**, and a **new section** in the updated manuscript. Moreover, we have conducted **additional experiments** to further validate our theoretical results in more diverse settings. Please find our detailed, point-by-point responses below. If you have any further feedback, please do not hesitate to share, and we would be more than happy to address it.
>
> **Q1:** It is unclear how Definition 1 (element-wise identifiability) is commonly used in the field of latent variable identification, as only two papers (by the same author) are cited below this definition.
>
> **A1:** Thanks for raising this point. Element-wise identifiability is the most common objective in the related field, and arguably the best we can achieve without additional information. In the original submission, we only listed two papers since (Hyvärinen and Pajunen, 1999) was one of the first to explore the problem of identifying nonlinear latent variable models with a focus on existence and uniqueness, while (Hyvärinen et al., 2024) provides a comprehensive survey on the development in the last two decades.
>
> At the same time, we agree that introducing more works would further highlight the importance of Definition 1 in the literature, and we have updated the manuscript as follows:
>
> > “Element-wise identifiability guarantees that the estimated factors correspond to the true generating factors without any mixture or entanglement. Standard ambiguities such as permutations and rescaling may remain after identification, which are fundamental indeterminacies commonly noted in the literature (Hyvärinen & Pajunen, 1999; Khemakhem et al., 2020a; Sorrenson et al., 2020; Hälvä et al., 2021; Yao et al., 2021; Lachapelle et al., 2022; Buchholz et al., 2022; Zheng et al., 2022; Lachapelle et al., 2024; Hyvärinen et al., 2024) and represent the best achievable outcome without imposing further restrictive assumptions.”
>
> We hope that our explanation and the update addresses your concern and clarifies the relevance of Definition 1 within the context of latent variable identification. Thanks again for your comment.
>
> **Q2:** In all the theorems, the author notes that the dataset should be sufficiently large. However, it is unclear what qualifies as 'large' and how this requirement is applied in the proofs.
>
> **A2:** Thank you for your great question. Following standard practice in the identifiability literature (see e.g., a recent survey (Hyvärinen et al., 2024)), our theoretical results are derived under the asymptotic setting, which is why we note ‘sufficiently large’ in the theorem. This ensures that finite sample errors do not interfere with the results, allowing the proofs to focus on the asymptotic case.
>
> To clarify and avoid potential confusion, we have removed this phrasing from the theorems and instead highlighted the asymptotic setting in the preliminaries as follows:
>
> > “Following the previous works (see e.g., a recent survey (Hyvärinen et al., 2024)), all of our results are in the asymptotic setting.”
>
> We hope this could help to avoid potential misunderstanding, and we sincerely appreciate your constructive feedback.

---

> > ### Author Response · Authors · 2024-11-21
> > **We sincerely appreciate your insightful feedback and valuable comments (2/3)**
> >
> > **Q3:** The main challenge of proving Theorem 1, as compared to previous works, is not clearly explained.
> >
> > **A3:** Thanks a lot for your feedback. In light of it, we have added a new section to discuss the technical challenge compared to previous works in detail. Since we have already highlighted the unique contributions of our results in introduction and discussion, we mainly focus on the technical challenges in our proof compared to others. Intuitively, since we do not assume specific types of noise (e.g., additivity), there exists minimal difference between noise and latent variables. Therefore, we cannot utilize some existing assumptions such as $2n+1$ domains or contrastive learning, to disentangle noise and latent variables. Instead, we leverage the existence of variability as a minimal difference on the distributions. The details are included in the new section added in the updated manuscript as follows:
> >
> > > “In this section, we discuss the challenge of modeling general noise, emphasizing the distinctions between noise and content variables as explored in previous works.”
> >
> > > “Content variables are typically semantically meaningful and are often explicitly contrasted with style variables. This enables existing techniques to disentangle content from style through structured variability. For example, contrastive learning frameworks (Von Kügelgen et al., 2021) use paired observations differing only in style (e.g., images with the same object but different backgrounds) to disentangle content. In multi-domain settings (Kong et al., 2022), content remains invariant across $O(n)$ distinct domains characterized by different styles. Similarly, in intervention-based settings (Lachapelle et al., 2024)}, agents or environments serve as auxiliary variables that induce changes in the conditional distributions of latent variables. These structured variations provide the foundation for effective disentanglement.”
> >
> > > “In contrast, noise variables often lack semantic meaning and cannot be explicitly manipulated across multiple domains or paired in observations. This makes it infeasible to assume the existence of $O(n)$ conditional distributions or define contrastive objectives. As a result, existing frameworks designed for content-style disentanglement cannot be directly applied to general noise modeling.”
> >
> > > “To address this, we propose to only leverage the existence of variability in the latent distribution, requiring only two distinct distributions as a minimal degree of change. This relaxation reduces the need for $O(n)$ distinct distributions, which are common in existing frameworks, and broadens applicability to scenarios where the distribution is not completely invariant. This shift, from explicitly controlling different types of variables to achieve a required degree of change or transition, to accommodating general variability in scenarios where the distribution is not completely invariant, represents a significant technical contribution that is essential to address the unique contribution of modeling general noise.”
> >
> > Thanks again for your effort and time reviewing our paper. If you have any additional feedback, please do not hesitate to let us know—we would be more than grateful for your insights.

---

> > > ### Author Response · Authors · 2024-11-21
> > > **We sincerely appreciate your insightful feedback and valuable comments (3/3)**
> > >
> > > **References**
> > >
> > > [1] Hyvärinen and Pajunen, Nonlinear independent component analysis: Existence and uniqueness results. Neural networks, 1999
> > >
> > > [2] Hyvärinen et al., Identifiability of latent-variable and structural-equation models: from linear to nonlinear, Annals of the Institute of Statistical Mathematics, 2024
> > >
> > > [3] Khemakhem et al., Variational autoencoders and nonlinear ICA: A unifying framework, AISTATS 2020
> > >
> > > [4] Sorrenson et al., Disentanglement by nonlinear ICA with general incompressible-flow networks (GIN), ICLR 2020
> > >
> > > [5] Hälvä et al., Disentangling identifiable features from noisy data with structured nonlinear ICA, NeurIPS 2021
> > >
> > > [6] Yao et al., Learning temporally causal latent processes from general temporal data, ICLR 2022
> > >
> > > [7] Lachapelle et al., Disentanglement via mechanism sparsity regularization: A new principle for nonlinear ICA, CLeaR 2022
> > >
> > > [8] Buchholz et al., Function classes for identifiable nonlinear independent component analysis, NeurIPS 2022
> > >
> > > [9] Zheng et al., On the identifiability of nonlinear ICA: Sparsity and beyond, NeurIPS 2022
> > >
> > > [10] Lachapelle et al., Nonparametric partial disentanglement via mechanism sparsity: Sparse actions, interventions and sparse temporal dependencies. arXiv 2024
> > >
> > > [11] Von Kügelgen et al., Self-supervised learning with data augmentations provably isolates content from style, NeurIPS 2021

---

> > > > ### Author Response · Authors · 2024-11-25
> > > > **We sincerely thank you for the time you’ve dedicated to reviewing our work**
> > > >
> > > > Dear Reviewer QWn8,
> > > >
> > > > We sincerely thank you for the time you’ve dedicated to reviewing our work. As the discussion period comes to a close, we would be truly grateful if you could let us know whether our responses have adequately addressed your concerns.
> > > >
> > > > With appreciation,
> > > >
> > > > Authors of Submission7585

---

> > > > > ### Comment · Reviewer_QWn8 · 2024-11-26
> > > > > **Response to author's comments**
> > > > >
> > > > > Thank you for addressing my questions. After considering the other reviewers' comments and the discussion, I will maintain my score, particularly in light of the concerns about the novelty of the proof for Theorem 1 raised by other reviewers.

---

> > > > > > ### Author Response · Authors · 2024-11-26
> > > > > > **Thanks a lot for your response**
> > > > > >
> > > > > > Thank you for your response. We believe the technical contribution of Theorem 1 is both significant and unique, as it establishes identifiability requiring only two domains, compared to the $O(n)$ domains necessary in previous techniques. We sincerely hope you might reconsider your assessment, though we fully respect and value your opinion. Thanks again for your time.

---

### Official Review · Reviewer_88C8 · 2024-11-04

**Soundness:** 2
**Presentation:** 2
**Contribution:** 2
**Rating:** 5
**Confidence:** 3

**Summary:**

In this work, the authors consider the problem of Element-wise Identifiability of the latent variables $\mathbf{z}$, i.e. up to the permutation and element-wise (coordinate-wise) invertible transformation of these latent variables. As the main contribution authors propose sufficiency conditions under which the latent variables $\amthbf{z}$ are element-wise identifiable for the general case of generative models. Additionally, they consider two subcases of this generative model one of which is generative models with additive noise, and prove similar results. Also, the authors provide the result of causal structure learning for the specific subcase of the causal model with additive noise.

**Strengths:**

As the main result, authors propose the sufficiency conditions for the element-wise identifiability of the latent variables under general assumptions, i.e:
- no restrictions on the noise (e.g. as non-Gaussian, etc);
- no restrictions on the generative mechanism f() (e.g. as additive noise model, etc).
This generality of the result is very interesting from a theoretical point of view that the latent space can be learned uniquely up to some permutation and element-wise invertible transformation.

**Weaknesses:**

- The paper is missing the citation of some important theoretical results from the field. For example:
  -  Peters, Jonas, et al. "Causal discovery with continuous additive noise models." (2014). This work solves the causal discovery problem for the general SEM with additive noise, while the authors in section 4.3 consider a subcase of the general SEM with additive noise. It is important to specify the limitations of both works and the importance of the proposed results compared to already existing ones.
- The paper is hard to read due to the lack of sufficient explanatory details of the theoretical constructs and manipulations. More details in the questions below.
- The paper proposes only identifiability guarantees, but there is no method proposed to identify the latent variable $\mathbf{z}$. Moreover, guarantees of identifiability of the latent variables up to element-wise invertible transformation are quite restrictive since the invertible transformation can be very complex and infeasible to learn.
- The generative model in Section 4.2 is just a specific case of the model considered in theorem 1, therefore it is not clear what is new contribution of the results presented in section 4.2 with respect to theorem 1.
- It is not clear how restrictive the assumption "Variability exists". More specifically, it is not obvious for me why Example 1 satisfy this assumption. More details in the question part.

**Questions:**

- The assumption "Variability exists" is an analogy for the assumption of "Domain Variability" (Kong et al. 2022). However, (Kong et al. 2022) require that for any set A there exist two realizations of unobserved variables $\mathbf{u}$ (domains) such that two integrals are not equal. However, in this work authors require that there exist two domains $d_1$ and $d_2$ such that for any set A the two specific integrals are not equal, that is much much stronger assumption. Additionally, it is not clear what these domains $d_1$ and $d_2$ are in the given model. Finally, the authors provide example 1 in which they propose a model that should satisfy the assumption "Variability exists". However, it is not clear from the given explanations why the considered integral would not be equal to 0. Moreover, the statement is only given for the set A of a specific structure, but it has to be true for arbitrary set A that satisfies the conditions from the assumption "Variability exists".
- In the proof of theorem 1 the authors refer to steps 1, 2, and 3 in the proof of Theorem 4.2 Kong et al. (2022) from which they conclude that some part of Jacobian is zero. However, it is not clear how the assumptions of (Kong et al. (2022)) correspond to the assumptions considered in this work and how the setting of the problems maps to each other. Therefore it's not clear and hard to follow how the results of steps 1, 2, and 3 (which take a few pages in Kong et al. (2022)) adjust to the proof considered in this work. I think all of these steps should be rewritten explicitly in the proof so anyone can follow it.
- Also, the latent variables $\mathbf{z}$ and noise $\epsilon$ are symmetric with respect to the generating mechanism $\mathbf{x}=f(\mathbf{z}, \mathbf{\epsilon})$. Therefore I would like to ask the authors if there is anything that can stop noise $\mathbf{\epsilon}$ to follow the same assumptions considered for $\mathbf{z}$. And if not, then how we can make sure that our algorithm recovers latent variables $\mathbf{z}$ instead of noise $\mathbf{\epsilon}$?

---

> ### Author Response · Authors · 2024-11-21
> **We deeply appreciate the time and effort you invested and your insightful comments (1/3)**
>
> We deeply appreciate the time and effort you invested and your insightful comments, which have greatly enhanced the clarity of the manuscript. Accordingly, we have added **several new discussions**, **full details of previous results**, and **additional experiments** in the updated manuscript, with a special focus on the **relation with previous works** and **clarification on conditions**. Please kindly find our response to each point detailed below. If you would like to offer any further suggestions, please feel free to let us know; we would be more than happy to make further adjustments as needed.
>
> **Q1:** The paper is missing the citation of some important theoretical results from the field, e.g., causal discovery with continuous additive noise models.
>
> **A1:** Thank you for raising this point. Upon carefully reviewing the mentioned results—which are indeed fundamental to causal discovery—we believe there may be some misunderstanding regarding the distinction between our work and existing causal discovery methods, such as those based on additive noise models (e.g., Peters et al., 2014). The key differences are as follows:
>
> - In existing works, they only consider the general SEM with *additive* noise, *without* any measurement error. That is, $\mathbf{x}_i = f\_{1,i}(\textbf{Pa}(\mathbf{x}_i)) + \boldsymbol{\xi}_i$.
>
> - In our work, we consider the general SEM with *general* noise, *with* nonlinear measurement error. That is, $\mathbf{z}_i = f\_{1,i}(\textbf{Pa}(\mathbf{z}_i), \boldsymbol{\xi}_i)$ and $\mathbf{x}_i = f\_{2,i}(\mathbf{z}_i) + \boldsymbol{\eta}_i$.
>
> In summary, the "additive noise" in existing works refers to noise strictly within the SEM, whereas we allow for more general, nonlinear, and non-additive noise in the SEM while also addressing scenarios with additional nonlinear measurement error. Thus, we are actually dealing with a more general setting compared to existing works.
>
>
> **Q2:** Lack of sufficient explanatory details of the theoretical constructs and manipulations.
>
>
> **A2:** Thank you very much for your time and effort in reviewing our manuscript. We greatly appreciate your thoughtful feedback. After carefully considering your questions, we believe there may be potential misunderstandings, some of which likely stem from a typo that we can immediately clarify. Your constructive comment has been invaluable in helping us identify and correct this typo to improve the clarity of our work. We have addressed these concerns in the revised manuscript to ensure better understanding and avoid further confusion. Please find our detailed response below:
>
>
> - **Q2.1:** The assumption of “Variability exists” seems to be much stronger than the assumption of “Domain Variability” (Kong et al. 2022), and it is not clear how restrictive it is.
>
> - **A2.1:** Thanks a lot for raising this point. We sincerely appreciate you bringing this to our attention. We conjecture that the confusion stems from a typo in our original submission. Our intention was for the assumption of “Variability exists” to be equivalent to the “Domain Variability” assumption as defined in Kong et al. (2022). To address and rectify this, we have implemented the following revisions in our manuscript:
>
>   - Notation:
> 	- Our domain variables are exactly the same as these in (Kong et al., 2022). To avoid potential misunderstanding, we have changed the notation from $\mathbf{d}$ to $\mathbf{u}$.
>
>   - Corrected Typo:
> 	- Identical to (Kong et al., 2022), the assumption requires that for any set $A$ there exists two domains, and these domains can change across different sets of $A$. The original manuscript contained a typo in the assumption statement by switching ‘for any’ and ‘there exists’, which we have corrected in the updated manuscript:
>
>   	> “ Suppose for any set $A \subseteq \mathcal{Z} \times \mathcal{E}$ …, there exist two domains $u_1$ and $u_2$ that are independent of $\boldsymbol{\epsilon}$ s.t….”
>
>   - Enhanced Explanation:
> 	- To further prevent confusion, we have included the following statement next to the assumption:
>
>   	> “The same as in (Kong et al., 2022), these two domains can differ for different values of $A$, providing great flexibility.”
>
>   	As a result, it is clearer that the assumption holds as long as the conditional probabilities do not cancel with each other over all pairs of domains. We have also highlighted it in the example, hoping that could also address your concerns:
>
>   	>“... Let us consider the two domains where the integrals do not cancel with each other (the domains can change across different sets $A$):...”
>
>   These updates enhance the overall clarity of our presentation and make it straightforward that the assumption of variability is natural, which is consistent with the conclusions in (Kong et al. 2022). We sincerely apologize for the oversight and any confusion it may have caused. Your feedback was instrumental, and we are grateful for your diligence in improving our manuscript.

---

> > ### Author Response · Authors · 2024-11-21
> > **We deeply appreciate the time and effort you invested and your insightful comments (2/3)**
> >
> > - **Q2.2:** In the proof of theorem 1 the authors refer to steps 1, 2, and 3 in the proof of Theorem 4.2 in (Kong et al., 2022) from which they conclude that some part of the Jacobian is zero. However, it is not clear how the assumptions of (Kong et al., 2022) correspond to the assumptions considered in this work and how the setting of the problems maps to each other.
> >
> >
> > - **A2.2:** Thanks so much for your comment. As detailed in A2.1, the related assumption is identical to that in (Kong et al. 2022) after correcting the typo. In light of your suggestion, we have included the proof from (Kong et al.) in our appendix for ease of reference (Lemma 1) with notational changes.
> >
> >
> > - **Q2.3:** The latent variables $\mathbf{z}$ and noise $\boldsymbol{\epsilon}$ are symmetric with respect to the generating mechanism. Therefore I would like to ask the authors if there is anything that can stop noise to follow the same assumptions considered for $\mathbf{z}$.
> >
> >
> > - **A2.3:** Thank you for your question. We would like to clarify that the latent variables $\mathbf{z}$ and the noise $\boldsymbol{\epsilon}$ are, in fact, *not symmetric* in the generating process. As mentioned in the assumption, $\mathbf{z}$ changes across different domains $\mathbf{u}$ while $\boldsymbol{\epsilon}$ is independent of $\mathbf{u}$. In other words, in our setting, noise stays invariant across different domains while latent variables do not.
> >
> >
> > **Q3:** The paper proposes only identifiability guarantees, but there is no method proposed to identify the latent variable. Moreover, guarantees of identifiability of the latent variables up to element-wise invertible transformation are quite restrictive since the invertible transformation can be very complex and infeasible to learn.
> >
> > **A3:** Thank you for raising this important point. Since our work focuses on identifiability theory, the results are intentionally estimator-agnostic. The theory addresses the conditions under which latent concepts can be identified from observations but does not prescribe a specific algorithm. Therefore, the specific estimation procedure is deferred in the experimental setup.
> >
> > Moreover, we fully agree with you that we cannot recover the exact value of each latent variable without any indeterminacy. At the same time, it might be worth noting that, in the identifiability literature, identifiability up to element-wise invertible transformation is the most common objective, and, arguably, the *strongest achievable* result without additional constraints (Hyvärinen and Pajunen, 1999; Taleb and Jutten 1999; Hyvärinen & Morioka, 2016; Khemakhem et al., 2020; Sorrenson et al., 2020; Hälvä et al., 2021;  Buchholz et al., 2022; Lachapelle et al., 2022; Zheng et al., 2022, Hyvärinen et al., 2024). This objective ensures full disentanglement of the latent variables, which has addressed a longstanding challenge in unsupervised disentangled representation learning (Locatello et al., 2019). Such guarantees have direct applications in various fields, including computer vision (Xie et al., 2023), extrapolation (Lachapelle et al., 2023), and dynamical systems (Yao et al., 2022). Therefore, we believe our identifiability result is both theoretically significant and practically impactful.
> >
> >
> > **Q4:** It is not clear what is the new contribution of the results presented in Section 4.2 with respect to Theorem 1.
> >
> > **A4:** Thank you for your question. Compared to Theorem 1, the result in Section 4.2 introduces additional contributions by demonstrating that component-wise identifiability can still be achieved even in the presence of an unknown nonlinear distortion and corresponding noise. Notably, in this extended setting, the new noise does not need to depend on the domains, and the same structural condition on the original mapping (prior to distortion) remains sufficient.
> >
> > This result serves to formally introduce the concept of distortion, which provides a natural transition to exploring a new setting—structure learning with nonlinear measurement error. To make this distinction clearer and avoid potential confusion, we have updated the manuscript by presenting the result in Section 4.2 as a corollary. We hope this clarification highlights their role in extending the scope of our framework and connecting different settings. We have also conducted additional experiments to validate the results in different settings (Appx. D.2). Please let us know if there are additional aspects we can further clarify.

---

> > > ### Author Response · Authors · 2024-11-21
> > > **We deeply appreciate the time and effort you invested and your insightful comments (3/3)**
> > >
> > > **References**
> > >
> > > [1] Peters et al., Causal discovery with continuous additive noise models, JMLR, 2014
> > >
> > > [2] Kong et al., Partial identifiability for domain adaptation, ICML 2022
> > >
> > > [3] Hyvärinen and Pajunen, Nonlinear independent component analysis: Existence and uniqueness results. Neural networks, 1999
> > >
> > > [4] Taleb and Jutten, Source separation in post-nonlinear mixtures,  IEEE Transactions on signal Processing, 1999
> > >
> > > [5] Hyvärinen and Morioka, Unsupervised feature extraction by time-contrastive learning and
> > > nonlinear ICA, NeurIPS 2016
> > >
> > > [6] Khemakhem et al., Variational autoencoders and nonlinear ICA: A unifying framework, AISTATS 2020
> > >
> > > [7] Sorrenson et al., Disentanglement by nonlinear ICA with general incompressible-flow networks (GIN), ICLR 2020
> > >
> > > [8] Hälvä et al., Disentangling identifiable features from noisy data with structured nonlinear ICA, NeurIPS 2021
> > >
> > > [9] Buchholz et al., Function classes for identifiable nonlinear independent component analysis, NeurIPS 2022
> > >
> > > [10] Lachapelle et al., Disentanglement via mechanism sparsity regularization: A new principle for nonlinear ICA, CLeaR 2022
> > >
> > > [11] Zheng et al., On the identifiability of nonlinear ICA: Sparsity and beyond, NeurIPS 2022
> > >
> > > [12] Hyvärinen et al., Identifiability of latent-variable and structural-equation models: from linear to nonlinear, Annals of the Institute of Statistical Mathematics, 2024
> > >
> > > [13] Locatello et al., Challenging common assumptions in the unsupervised learning of disentangled representations, ICML 2019
> > >
> > > [14] Xie et al., Unpaired image-to-image translation with shortest path regularization, CVPR 2023
> > >
> > > [15] Lachapelle et al., Additive decoders for latent variables identification and cartesian-product extrapolation, NeurIPS 2023
> > >
> > > [16] Yao et al.,Temporally Disentangled Representation Learning, NeurIPS 2022

---

> > ### Comment · Reviewer_88C8 · 2024-11-22
> > **Response to the authors**
> >
> > Thank you for the deliberate response and for giving deliberate details on my questions!
> > With the additional details on the environment properties my concerns related to the generating function and distinguishability of latent variables and noise resolved.
> > However, I still have a few questions:
> > - In the manuscript (Kong et al. 2022) there is an assumption that each domain $u$ is the component-wise monotonic transformation of the latent variables $z$. In (Kong et al. 2022) this is an important assumption, so they conclude some specific properties for the estimations, which makes sense. There is no such assumption in this manuscript that feels concerning.
> > - I could not find in this manuscript how the function $\hat{f}$ is defined except that it is an estimation of function f. How one can obtain such an estimation and what properties this estimation has?
> > - "In contrast, our theory only necessitates two domains, regardless of the number of variables, representing what could be considered a minimal level of required variability." This should be equivalent to asking the existence of just one set $A$ such that there are two domains $u_1$ and $u_2$ exist in the assumption (Variability exists). However (Variability exists) requires that for any set $A$ there exist two domains $u_1$ and $u_2$ that could be different for different sets $A$. Does it mean that the assumption (Variability exists) can be simplified?
> > - If the assumption (Variability exists) is equivalent to the assumption (Domain Variability) (Kong et al. 2022), why do they have different names? It may be confusing.
> > - I was not able to find example 1 from the old version of the manuscript in which they propose a model that should satisfy the assumption "Variability exists". I still don't understand why the conditional probabilities should cancel out each other to make the integral equal to zero. If I remember correctly the integral was something like $\int_{A} p(\mathbf{\epsilon})(p(..|u_1) - p(..|u_2)) d \mathbf{z} d \mathbf{\epsilon}$. Where the difference in conditional probabilities can be as negative as positive. So the overall integral in general may be equal to zero even if $p(..|.., u_1) - p(..|..,u_2) \neq 0$.

---

> ### Author Response · Authors · 2024-11-22
> **Thanks so much for your prompt and insightful feedback (1/2)**
>
> Dear Reviewer 88C8,
>
> Thanks so much for your prompt and insightful feedback. These new questions, again, helped us to further improve the manuscript, especially in presentation and clarification. We sincerely appreciate your time and effort. Please kindly find our point-by-point responses:
>
> **Q5:** In the manuscript (Kong et al. 2022) there is an assumption that each domain $\mathbf{u}$ is the component-wise monotonic transformation of the latent variables $\mathbf{z}$. In (Kong et al. 2022) this is an important assumption, so they conclude some specific properties for the estimations, which makes sense. There is no such assumption in this manuscript that feels concerning.
>
> **A5:** Thanks for raising this point. By “each domain $\mathbf{u}$ is the component-wise monotonic transformation of the latent variables $\mathbf{z}$”, if we understand correctly, perhaps you mean style variables $\mathbf{z}_s$ are generated from a high-level invariant part $\tilde{\mathbf{z}}_s$ by a component-wise monotonic function $f\_\mathbf{u}$, i.e., $\mathbf{z}\_s = f\_{\mathbf{u}} (\tilde{\mathbf{z}}\_s)$. The assumption is only used for the application in domain adaptation but not in the proof of identifiability in (Kong et al. 2022). Since our study does not address domain adaptation, this assumption is unnecessary for our theoretical results.
>
> Please let us give a bit more details to also describe our understanding of the role of assumption in (Kong et al. 2022). In that work, for the purpose of domain adaptation, they need to find a high-level invariant representation across domains to learn an optimal classifier over domains. Thus, although $p(\mathbf{z}_s)$ change but $p(\tilde{\mathbf{z}}_s)$, $p(y|\tilde{\mathbf{z}}_s)$, and $p(y|\tilde{\mathbf{z}}_c)$ stay the same ($\mathbf{z}_s$ denotes changing variables while $\mathbf{z}_c$ denotes the invariant part). Specifically, if they assume that transformation is monotonic, then they can directly find the information of $\mathbf{z}_s$ in each domain, and thus domain adaptation is achieved. In our work, since the focus is not on domain adaptation, this treatment is not relevant to our setting or results.
>
> **Q6:** I could not find in this manuscript how the function is defined except that it is an estimation of function f. How one can obtain such an estimation and what properties this estimation has?
>
> **A6:** Thanks so much for your question. In light of it, in addition to the details in the experimental setup (L465-476), we have also added the following sentence earlier in the preliminary to avoid any potential confusion (L128-130):
>
> > “The estimated model $(\hat{f}, \hat{\mathbf{z}}, \hat{\boldsymbol{\epsilon}})$ follows the data-generating process and matches the observed distributions, i.e., $p(\hat{\mathbf{x}}) = p(\mathbf{x})$ ($p(\hat{\mathbf{x}}|\mathbf{u}) = p(\mathbf{x}|\mathbf{u})$ if there exists a domain variable $\mathbf{u}$).”
>
> Please kindly let us know if you have any further suggestions. Thanks again for your valuable feedback.
>
> **Q7:** Certain aspects of the discussion regarding the (Variability exists) assumption are unclear or potentially misleading.
>
> **A7:** We sincerely appreciate your detailed reading and constructive suggestion, which helped us to further improve the manuscript to avoid potential confusion. You are totally right, that sentence could be misleading. Therefore, in light of your insightful suggestion, we have changed it as follows (L219-222) to highlight the specific condition that needs to be satisfied (note that we changed the name of the assumption):
>
> > “Differently, our theory does not put a hard constraint on requiring $O(n)$ domains, as long as the specific assumption of domain variability holds. However, since the conditions are different, the assumption of domain variability is not strictly weaker than the previous assumptions.”
>
> We hope this clarification could address your concern and ensure the discussion is more precise. If you have further additional feedback, please feel free to let us know, and we would be more than happy to make corresponding adjustments.
>
>
> **Q8:** If the assumption (Variability exists) is equivalent to the assumption (Domain Variability) (Kong et al. 2022), why do they have different names? It may be confusing.
>
> **A8:** Thank you for your excellent suggestion. Accordingly, we have updated the terminology throughout the paper to use "Domain Variability" for consistency. While we initially chose the previous name to emphasize that the problem is not domain adaptation, we fully agree that having different names could be confusing. We sincerely appreciate your constructive feedback, which has helped us improve the clarity of our manuscript.

---

> > ### Author Response · Authors · 2024-11-22
> > **Thanks so much for your prompt and insightful feedback (2/2)**
> >
> > **Q9:** I was not able to find the example of the variability exists assumption from the old version of the manuscript. The old example does not seem to work.
> >
> > **A9:** Thanks so much for your feedback. We fully agree with you that some parts were missing in the previous example. Specifically, in the example, an additional constraint should have been included regarding the choice of domains:
> >
> > >“...Let us consider the two domains where the integrals do not cancel with each other (the domains can change across different sets $A$):”
> >
> > That said, even with this clarification, we feel the example remains uninformative. The condition essentially serves as a generic faithfulness assumption, ruling out specific cases where specific combinations of parameters make these two integrals cancel out each other. As such, the example does not add significant insight. We have therefore decided to remove the example in the updated manuscript to avoid unnecessary confusion, as well as compensating the space for other discussions and clarifications. If possible, please kindly let us know if you think this would be a reasonable choice. We sincerely appreciate your thoughtful suggestions, which have been invaluable in improving the manuscript from various perspectives.
> >
> > ---
> >
> > To summarize, we are sincerely grateful for your thoughtful and constructive questions, suggestions, and comments. It is truly a privilege to have you dedicate your time and effort to helping us improve our manuscript. We deeply appreciate your insights and sincerely look forward to any additional feedback you may have.
> >
> >
> >
> > Many thanks,
> >
> > Authors of Submission7585

---

> > > ### Author Response · Authors · 2024-12-01
> > > **Sincerely appreciate your insightful questions; Looking forward to your further feedback**
> > >
> > > Dear Reviewer 88C8,
> > >
> > > We sincerely appreciate your insightful questions and suggestions. We have provided additional responses above to address these in detail. As the discussion period is nearing its conclusion, we would greatly appreciate it if you could kindly let us know whether our responses have adequately addressed your questions.
> > >
> > > Thank you again for your time and thoughtful feedback.
> > >
> > > With gratitude,
> > >
> > > Authors of Submission7585

---

> > > > ### Author Response · Authors · 2024-12-02
> > > > **Gentle Reminder: Feedback Appreciated Before Discussion Ends in 12 Hours**
> > > >
> > > > Dear Reviewer 88C8,
> > > >
> > > > We hope this message finds you well. As the discussion period is nearing its conclusion in approximately **12 hours**, we wanted to kindly follow up on our earlier responses. We would greatly appreciate your feedback on whether our clarifications have addressed your additional questions. If they have, we would be most grateful if you could consider updating your rating accordingly.
> > > >
> > > > Thank you once again for your time and thoughtful input throughout the review process.
> > > >
> > > > Sincerely,
> > > >
> > > > Authors of Submission7585

---

> > > > > ### Comment · Reviewer_88C8 · 2024-12-02
> > > > > **Would increase the score from 3 to 5**
> > > > >
> > > > > Thanks for the deliberate answer to all my questions. Therefore I am willing to increase my score to 5. Finally, the main reasons for such a score are the following:
> > > > > - although the identifiability result is established, however, it is not clear how it can be recovered algorithmically
> > > > > - I agree with a reviewer KLTp concerns regarding the novelty of the proofs
> > > > > - additionally, the authors emphasize in their responses that 2 domains may suffice for identifiability. However, it would happen only under very specific conditions in the assumption (Domain Variability). But it is not clear how this would be restrictive. Moreover, it is not explicitly clear if even O(n) domains would be enough for the general case, and whether any bound for it exists.

---

> > > > > > ### Author Response · Authors · 2024-12-03
> > > > > > **Thanks so much for your follow-up response**
> > > > > >
> > > > > > Dear Reviewer 88C8,
> > > > > >
> > > > > > Thanks so much for your follow-up response and the update on the score. We fully respect your opinions on the assumption. At the same time, we would like to highlight the following points that may provide additional context:
> > > > > >
> > > > > > - From a theoretical perspective, we believe the condition is **not overly restrictive**, as it only excludes probability measures that cancel each other out within specific rectangular sets. This is analogous to the generic faithfulness assumption in the literature, which avoids specific parameter combinations that cancel each other. While we fully agree that further exploration (e.g., providing bounds) for such generic conditions would be highly valuable, we humbly feel it is beyond the current scope of our asymptotic identifiability setup.
> > > > > >
> > > > > > - In all of our experiments, we use **only two domains**, and the results consistently indicate that all latent variables can be identified in the considered setting. Empirically, this suggests that two domains are sufficient for identification in our framework.
> > > > > >
> > > > > > Moreover, we believe it is worth noting that we have proposed structural identifiability results for **general nonlinear SEMs with nonlinear measurement error**, addressing a long-standing open problem in causal discovery. As such, we believe our contributions to both representation and structure identification are novel and valuable.
> > > > > >
> > > > > > Lastly, we deeply appreciate the time and effort you have dedicated to reviewing our manuscript. While we may hold differing perspectives on certain aspects, we have the utmost respect for your feedback. We sincerely hope you might reconsider your assessment in light of our clarifications and contributions, and we would be more than happy to address any remaining concerns you may have.
> > > > > >
> > > > > > With gratitude,
> > > > > >
> > > > > > Authors of Submission7585

---

### Official Review · Reviewer_Vhc1 · 2024-11-04

**Soundness:** 4
**Presentation:** 2
**Contribution:** 3
**Rating:** 6
**Confidence:** 3

**Summary:**

This paper gives conditions for identifying latent variables up to an element-wise transformation under general noise (i.e. there's no assumption of additive noise in the mixing / generative function). The show results under general noise with three main assumptions (Nondegeneracy, variability and structural sparsity), and then show how assuming additive noise allows them to drop the variability assumption. Finally in theorem 3 they give a more general noise condition, which can be extended to learning a causal DAG.

**Strengths:**

The theory gives strong results: they have a more general noise condition (in that they do not need to assume additivity), while reducing the number environments needed to identify the latents from n (where n is the number of latents) to 2.

They are also able to identify causal DAGs of the form given in figure 5, by leveraging the equivalence between the mixing function and the DAG for problems of that form (there are many problems in causal representation learning where that won't be true because there isn't a fixed DAG that connects latents & observations; but it is still a useful observation).

**Weaknesses:**

My complaints are mostly around presentation. I felt that the paper does a poor job of explaining its results & it tends to oversell the practical relevance.

* **Explanation** I am very familiar with this literature and I found it hard to see why the assumptions leveraged in this paper lead to just strong identifiability results. For example, whether or not *(Variability exists)* is assumed, is the key difference between Theorem 1 and 2 (assuming additive noise allow one to drop the assumption that variability exists), but I couldn't see where in the proof the assumption that variability exists is used (it isn't referenced anywhere).

  Lines 226 - 230 explain (correctly) that most identifiability results need at least O(n) (where n is the number of latents) to disentangle latents, but it doesn't explain *how* this approach avoids that requirement? What are you leveraging to avoid that? The example doesn't give any insight because it is a case where there's only two parameters & two environments, so it's not surprising that its sufficient. Is it that you only require two environments to separate the noise from the latents or something else?

  Finally, I think giving a proof sketch in the main paper is a far better use of space than some of the examples that were given. I think the reader should be able to understand the main steps of the proofs in the main text without having to go through all the details in the appendix.
* **Overselling applications** the paper lists many applications from medical imaging to finance where general noise models are necessary, and makes it sound like extending identifiability results to these settings is all that stands in the way of applications. E.g. in the statement *"the capacity to handle general, nonparametric noise extends the applicability of the proposed theory across a wide range of real-world scenarios, regardless of the complexity of the noise"*. Or *"This is crucial for applications such as dependable patient monitoring systems. [line 302]"*.

  While it's good to know that these settings are identifiable, we need to be far more upfront about the limitations of causal representation learning methods. There is a big gap between theory and practice. These methods all assume that we perfectly fit the data distribution with no constraints on model capacity & it's not at all clear how they perform in finite samples with estimation error. That is likely to be a far bigger block to practical application than identifiability. Or put differently, identifiability is necessary, but far from sufficient for practical application, and making that clear is important (or alternatively, provide convincing experiments in real world settings, not toy problems like MNIST / fashion MNIST).
* **Unconvincing experiments** Simple simulations are fine as a sanity check, but we have to move past these synthetic problems in evaluation. The simulation results don't give enough detail to properly evaluate, but the data generating process appears to be just a simple diffeomorphism defined by a normalizing flow model (I'm guessing here). If that's the case, you're in the well-specified case, so it's far more simple than any real world scenario.
* I liked the MNIST and FashionMNIST examples a little more - but they also expose just how unreliable and hard-to-work-with these methods are in practice. For example, in the MNIST example, panel 2 and 3 are interpreted as capturing hight and "right slope" respectively, but they're also clearly entangled with stroke width (the right most columns uses thinner pen strokes than the left most columns in both images). And I couldn't understand the latents in the FashionMNIST examples. I think its better to be upfront about these limitations in the main text rather than bury it in the appendix.

**Questions:**

What makes the general noise setting fundamentally different from the noiseless setting? If I have method that can identify latents in the noiseless setting with n variables, can I not just treat the noise variable as the n+1'th latent and apply an existing approach?

Assuming the above is true, why didn't you empirically compare to any existing methods?

---

> ### Author Response · Authors · 2024-11-21
> **We greatly respect the depth of your review and the meaningful perspectives you offered (1/3)**
>
> We greatly respect the depth of your review and the meaningful perspectives you offered. All of these constructive comments have not only improved the manuscript but also inspired us to think more deeply about the future challenges of the field. In light of these, we have incorporated **new experiments**, **new discussions**, and **additional clarification** in the updated manuscript. Our detailed responses are outlined below for your review. Please feel free to share any additional suggestions; we would be more than happy to make further adjustments as needed.
>
>
> **Q1:** Explanation of several details.
>
> **A1:** We sincerely appreciate all these insightful questions, which have helped us to further improve the presentation. Please kindly find our point-by-point responses below.
>
> - **Q1.1:** Where in the proof the assumption that variability exists is used?
>
> - **A1.1:** Thank you for your question. The variability assumption is utilized to demonstrate that the bottom-left block of the Jacobian, $\dfrac{\partial \boldsymbol{\epsilon}}{\partial \hat{\mathbf{z}}}$, is zero. As noted in the manuscript, this portion of the proof directly follows steps 1, 2, and 3 of the proof of Theorem 4.2 in (Kong et al., 2022), where the variability assumption is applied. In light of your great suggestion, we have now included the relevant result from (Kong et al., 2022) as Lemma 1 in Appendix B.1, along with the complete proof, adapted to align with our notation.
>
>
> - **Q1.2:** How does the approach avoid the requirement of at least $O(n)$ environments? What are you leveraging to avoid that? Is it that you only require two environments to separate the noise from the latents or something else?
>
> - **A1.2:** Exactly, we only require two environments to disentangle the noise and latent variables, where the dimensions of both can be significantly larger than two. As you pointed out, this approach is fundamentally more general than prior work that assumes at least $O(n)$ environments. Requiring only two environments essentially reduces to ensuring the presence of variability. This is not only practically significant but also theoretically intriguing, as it avoids introducing any additional assumptions. The relevant part of the proof can be found in L888-1057. Essentially, we found that the variability assumption, previously used to disentangle invariant variables from changing ones, can also imply the disentanglement of the changing part $\mathbf{z}$ from the invariant part $\boldsymbol{\epsilon}$ by leveraging the generative process where $\mathbf{z}$ is independent of $\boldsymbol{\epsilon}$, and thus establish block-wise identifiability.
>
> - **Q1.3:** I think giving a proof sketch in the main paper is a far better use of space than some of the examples that were given.
>
> - **A1.3:** Thanks so much for your great suggestion. In light of this, we have removed some real-world examples and added a proof sketch for the main result as follows:
>
>   > “Proof Sketch. We leverage distributional variability across two domains of the latent variables $\mathbf{z}$ to disentangle $\mathbf{z}$ and $\boldsymbol{\epsilon}$ into independent subspaces. To separate general noise from latent variables, we use the independence between $\mathbf{z}$ and $\boldsymbol{\epsilon}$ alongside the variability within $\mathbf{z}$. The structural sparsity condition is then employed to identify individual components of $\mathbf{z}$ in the nonlinear setting. Specifically, for each latent variable, the intersection of parental sets from a subset of observed variables uniquely specifies it. Since we only achieve relations among supports due to the nonparametric nature of the problem, an unresolved element-wise transformation remains. Consequently, we achieve element-wise identifiability for the latent variables $\mathbf{z}$ (Defn. 1).”

---

> > ### Author Response · Authors · 2024-11-21
> > **We greatly respect the depth of your review and the meaningful perspectives you offered (2/3)**
> >
> > **Q2:** It tends to oversell the applications of causal representation learning methods, since these methods all assume that we perfectly fit the data distribution with no constraints on model capacity and it is not at all clear how they perform in finite samples with estimation error.
> >
> > **A2:** Thank you for your thoughtful comment. We agree that causal representation learning (CRL) methods face significant challenges before they can fully benefit a wide range of real-world applications. In complex real-world scenarios, it is difficult to ensure that assumptions are perfectly satisfied, likelihoods are accurately matched, scalability is not a concern, or that finite sample errors do not introduce bias.
> >
> > We share your view that substantial work remains in the field to bridge this gap. To highlight this, we have removed the discussion on applications and instead emphasized the limitations in the updated manuscript. For instance, we have added the following clarification:
> >
> > > “Moreover, for real-world scenarios, it is extremely challenging to make sure that all conditions on the latent data generating process are perfectly satisfied and the distributions are perfectly matched after estimation. Bridging the gap requires a thorough study of the finite sample error and the robustness of the identification, which remains an open challenge in the literature.”
> >
> > Moreover, we have also emphasized this point in the introduction, specifically in the paragraphs outlining our contributions, to ensure the message is as clear as possible:
> >
> > > “... but addressing practical challenges like finite sample errors remains a key open problem for future work to enable broader deployment of identifiability theory.”
> >
> > **Q3:** Synthetic experimental setting is far more simple than real-world scenarios since simulation is well-specified, although simulations are fine as a sanity check. I liked the MNIST and FashionMNIST examples a little more, but the results are not easy to interpret and also expose practical challenges.
> >
> > **A3:** Thank you for your insightful opinion. As discussed in detail in Q2&A2, we fully agree that there remain significant challenges in completely bridging the gap between causal representation learning (CRL) methods and large-scale real-world applications. Your suggestion highlights one of the key obstacles the CRL community must address, and we sincerely appreciate your valuable perspective. To work towards that goal, we have further conducted **additional experiments** on more synthetic settings with different noise and distortion. According to the new results (Appx. D.2), we can still identify the latent variables in these settings. Although the discussed gaps such as finite sample error are still there, we hope to try our best to empirically support the theory in more diverse settings.
> >
> > Regarding the image examples (e.g., MNIST and FashionMNIST), we acknowledge the difficulty of ensuring fully interpretable recovered semantics. While humans can often infer meaning based on their understanding, there is no guarantee that the underlying generative process aligns with our interpretations. Some latent factors may be inherently "entangled" or lack clear semantic meaning in human terms, yet still represent statistically independent components of the underlying generative process. At the same time, practical issues such as finite sample error also hinder the perfect recovery of the hidden factors. In light of your great comment, to emphasize this point, we have explicitly listed these challenges in the main text of the updated manuscript:
> >
> > > “Importantly, several practical challenges persist. For example, human interpretations of latent factors are often guided by intuition, yet there is no guarantee that the true generative process aligns with these interpretations. Certain latent factors may inherently appear entangled or lack clear semantic meaning from a human perspective, even if they represent statistically independent components of the generative mechanism. Furthermore, practical constraints, such as finite sample errors, pose additional challenges to achieving perfect recovery of the hidden factors.”

---

> > > ### Author Response · Authors · 2024-11-21
> > > **We greatly respect the depth of your review and the meaningful perspectives you offered (3/3)**
> > >
> > > **Q4:** What makes the general noise setting fundamentally different from the noiseless setting? If I have a method that can identify latents in the noiseless setting with $n$ variables, can I not just treat the noise variable as the $(n+1)$-th latent and apply an existing approach?
> > >
> > > **A4:** Thank you so much for your question. The fundamental difference in the general noise setting lies in the lack of specific constraints on the noise. Thus it is challenging to separate them from the latent variables. That being said, in our general setting, we cannot assume existing assumptions such as additivity, $O(n)$ domains or structural conditions w.r.t. to noise. Existing conditions can indeed help to identify latent variables in the noiseless setting, but if we have additional general noise variables without these conditions, the previous theory cannot be applied to disentangle the influence of these non-negligible noises.
> > >
> > >
> > > ---
> > >
> > > **References**
> > >
> > > [1] Kong et al., Partial identifiability for domain adaptation, ICML 2022

---

> > > > ### Author Response · Authors · 2024-11-25
> > > > **Thank you so much for taking the time to provide your valuable feedback**
> > > >
> > > > Dear Reviewer Vhc1,
> > > >
> > > > Thank you so much for taking the time to provide your valuable feedback. As the discussion period draws to a close, we would greatly appreciate it if you could kindly let us know whether our responses have adequately addressed your concerns.
> > > >
> > > > Many thanks,
> > > >
> > > > Authors of Submission7585

---

### Note · Authors · 2025-01-24

I have read and agree with the venue's withdrawal policy on behalf of myself and my co-authors.